# Achieving Sub-linear Regret in Infinite Horizon Average Reward Constrained MDP with Linear Function Approximation

**Arnob Ghosh**
Electrical and Computer Engineering
The Ohio State University
ghosh.244@osu.edu

**Xingyu Zhou**
Electrical and Computer Engineering
Wayne State University
xingyu.zhou@wayne.edu

**Ness Shroff**
Electrical and Computer Engineering
The Ohio State University
shroff.11@osu.edu

## Abstract

We study the infinite horizon average reward constrained Markov Decision Process (CMDP). In contrast to existing works on model-based, finite state space, we consider the model-free linear CMDP setup. We first propose a computationally inefficient algorithm and show that $\tilde{\mathcal{O}}(\sqrt{d^3 T})$ regret and constraint violation can be achieved, in which $T$ is the number of interactions, and $d$ is the dimension of the feature mapping. We also propose an efficient variant based on the primal-dual adaptation of the LSVI-UCB algorithm and show that $\tilde{\mathcal{O}}((dT)^{3/4})$ regret and constraint violation can be achieved. This improves the known regret bound of $\tilde{\mathcal{O}}(T^{5/6})$ for the finite state-space model-free constrained RL which was obtained under a stronger assumption compared to ours. We also develop an efficient policy-based algorithm via novel adaptation of the MDP-EXP2 algorithm to the primal-dual set up with $\tilde{\mathcal{O}}(\sqrt{T})$ regret and even zero constraint violation bound under a stronger set of assumptions.

## 1 Introduction

In many standard applications of Reinforcement learning (RL) (e.g., autonomous vehicles), the agent needs to satisfy certain constraints (e.g., safety constraint, fairness). These problems can be formulated as constrained Markov Decision process (CMDP) such that the agent needs to ensure that average utility (cost, resp.) exceeds a certain threshold (is below a threshold, resp.).

While CMDP in the finite state-space has been studied, those studies can not be extended to the large state space. RL with value function approximation has demonstrated empirical success for large-scale RL application using the deep neural networks. However, theoretical understandings of constrained RL with value function approximation is quite limited. Recently, Ghosh et al. (2022) has made some progress towards the understanding of the constrained RL for linear MDP under episodic setting. In particular, Ghosh et al. (2022) developed a primal-dual adaptation of the LSVI-UCB and showed $\tilde{\mathcal{O}}(\sqrt{d^3 T})$ regret and violation where $d$ is the dimension of the feature space and $T$ is the number of interactions. Importantly, the above bounds are independent of the cardinality of state-space.

However, infinite-horizon model fits well compared to the finite horizon setting for many real-world applications (e.g., stock-market investment, routing decisions). Compared to the discounted-reward model, maximizing the long-term average reward under the long-term average utility also has its advantage in the sense that the transient behavior of the learner does not really matter for the latter case (Wei et al., 2020). Recently, model-based RL algorithm for infinite-horizon average reward CMDP has been proposed (Chen et al., 2022). However, it considers tabular setup. Further, the model-based approach requires large memory to store the model parameters. It is also hard (computationally) to extend model-based approaches for infinite state-space such as linear MDP (Wei et al., 2020).

Model-free RL algorithm is more popular because of the ease of implementation, and being computational, and storage efficient particularly for the large state space. However, model-free learning for the infinite horizon average reward setup is even more challenging. For example, it is still unknown whether it is possible to achieve a computationally efficient model-free algorithm with $\tilde{\mathcal{O}}(\sqrt{T})$ regret even under the unconstrained tabular setup (Wei et al., 2020) for a weakly communicating MDP. To the best of our knowledge, Wei et al. (2022) is the only paper to study model-free algorithms for CMDPs in the infinite horizon average reward setup. In particular, they consider the finite-state tabular setting and the regret scales polynomially with the number of states. Thus, the result *would not be useful* for large-scale RL applications where the number of states could even be infinite. To summarize, little is known for the performance guarantee of *model-free algorithms* in CMDPs beyond tabular settings in the infinite-horizon average reward setting, even in the case of linear CMDP. Motivated by this, we are interested in the following question:

*Can we achieve provably sample-efficient and model-free exploration for CMDPs beyond tabular settings for the infinite horizon average reward setting?*

**Contribution.** To answer the above question, we consider CMDP with linear function approximation, where the transition dynamics, the utility function, and the reward function can be represented as a linear function of some known feature mapping. Our main contributions are as follows.

- We propose an algorithm (Algorithm 1) which achieves $\tilde{\mathcal{O}}(\sqrt{d^3T})$ regret and constraint violation bounds with a high probability when the optimal policy belongs to a smooth function class (Definition 1). *This is the first result that shows that $\tilde{\mathcal{O}}(\sqrt{T})$ regret and violation are achievable for linear CMDP in the infinite-horizon regime using model-free RL.* Achieving uniform concentration bound for individual value function turns out to be challenging and we need to rely on the smoothness of the policy, unlike the unconstrained case. The algorithm relies on an optimizer that returns the parameters corresponding to the state-action bias function by solving a contained optimization problem.

- We also propose an efficient variant and show that $\tilde{\mathcal{O}}((dT)^{3/4})$ regret and violation bound can be achieved. *This is the first result that provides sub-linear regret and violation guarantee under only Assumption 1 for linear CMDP using a computationally efficient algorithm.* The idea is to consider a finite-horizon episodic setup by dividing the entire time-horizon $T$ into multiple episodes $T/H$ where each episode consists of $H$ steps. We then invoke the primal-dual adaptation of the LSVI-UCB algorithm proposed in Ghosh et al. (2022) to learn the good policy for the finite-horizon setting by carefully crafting the constraint for the episodic case. Finally, we bound the gap between the infinite-horizon average and the finite-horizon result to obtain the final result.

- We also propose an algorithm which can be implemented efficiently and achieves $\tilde{\mathcal{O}}(\sqrt{T})$ regret and $\tilde{\mathcal{O}}(\sqrt{T})$ constraint violation under a stronger set of assumptions (similar to the ones made in Wei et al. (2021a) for the unconstrained setup). We, further, show that one can achieve *zero* constraint violation for large enough (still, finite) $T$ while maintaining the same order on regret.

- We attain our bounds without estimating the unknown transition model or *requiring a simulator*, and they depend on the state-space only through the dimension of the feature mapping. *To the best of our knowledge, these sub-linear bounds are the first results for model-free (or, model-based) online RL algorithms for infinite-horizon average reward CMDPs with function approximations.* Wei et al. (2022) proposes a model-free algorithm in the tabular setting which achieves $\tilde{\mathcal{O}}(T^{5/6})$ regret. Since linear MDP contains tabular setting our result improves the existing results. Further, we show that we can achieve zero constraint violation by maintaining the same order on regret under the same set of Assumptions of Wei et al. (2022).

We relegate Related Work to Appendix A.

## 2 PRELIMINARIES

We consider an infinite horizon constrained MDP, denoted by $(\mathcal{S}, \mathcal{A}, \mathbb{P}, r, g)$ where $\mathcal{S}$ is the state space, $\mathcal{A}$ is the action space, $\mathbb{P}$ is transition probability measures, $r$, and $g$ are reward and utility functions respectively. We assume that $\mathcal{S}$ is a measurable space with possibly infinite number of elements, and $\mathcal{A}$ is a finite action set. $\mathbb{P}(\cdot|x, a)$ is the transition probability kernel which denotes the probability to reach a state when action $a$ is taken at state $x$. We also denote $\mathbb{P}$ as $p$ to simplify the notation. $p$ satisfies $\int_{\mathcal{X}} p(dx'|x, a) = 1$ (following integral notation from Hernández-Lerma (2012))

$r : \mathcal{S} \times \mathcal{A} \to [0, 1]$, and $g : \mathcal{S} \times \mathcal{A} \to [0, 1]$ and are assumed to be deterministic. We can readily extend to settings when $r$ and $g$ are random.

The process starts with the state $x_1$. Then at each step $t \in [T]$, the agent observes state $x_t \in \mathcal{S}$, picks an action $a_t \in \mathcal{A}$, receives a reward $r(x_t, a_t)$, and a utility $g(x_t, a_t)$. The MDP evolves to $x_{t+1}$ that is drawn from $\mathbb{P}(\cdot|x_t, a_t)$. In this paper, we consider the *challenging* scenario where the agent only observes the bandit information $r(x_t, a_t)$ and $g(x_t, a_t)$ at the visited state-action pair $(x_t, a_t)$. The policy-space of an agent is $\Delta(\mathcal{A}|\mathcal{S}); \{\{\pi_t(\cdot|\cdot)\} : \pi_t(\cdot|x) \in \Delta(\mathcal{A}), \forall x \in \mathcal{S}, \text{ and } t \in T\}$. Here $\Delta(\mathcal{A})$ is the probability simplex over the action space. For any $x_t \in \mathcal{S}$, $\pi_t(a_t|x_t)$ denotes the probability that the action $a_t \in \mathcal{A}$ is taken at step $t$ when the state is $x_t$. We also denote the space of stationary policy as $\Delta^s(\mathcal{A}|\mathcal{S})$ where the policy $\pi(\cdot|x_t)$ is independent of time $t$.

Let $J_r^\pi(x)$ and $J_g^\pi(x)$ denote the expected value of the average reward and average utility respectively starting from state $x$ when the agent selects action using the stationary policy $\pi \in \Delta^s(\mathcal{A}|\mathcal{S})$

$$J_\diamond^\pi(x) = \lim_{T \to \infty} \frac{1}{T} \mathbb{E}_\pi \left[ \sum_{i=1}^T \diamond(x_i, a_i)|x_1 = x \right], \tag{1}$$

for $\diamond = r, g$, where $\mathbb{E}$ is taken with respect to the policy $\pi$ and the transition probability kernel $\mathbb{P}$.

From Puterman (2014), there exists a state-action bias function $q$ satisfying the Bellman's equation for $\diamond = r, g$: $q_\diamond^\pi(x, a) + J_\diamond^\pi(x) = \diamond(x, a) + \mathbb{E}_{x' \sim p(\cdot|x, a)}[v_\diamond^\pi(x')]$ where $v_\diamond^\pi(x) = \sum_a \pi(a|x) q_\diamond(x, a) = \langle \pi(\cdot|x), q_\diamond(x, \cdot) \rangle_\mathcal{A}$. $q^\pi$ and $v^\pi$ are analogue to the $Q$-function and value function respectively for finite-horizon and discounted infinite horizon scenario.

**The Problem**: We are interested in solving the following problem

$$\text{maximize}_\pi J_r^\pi(x) \quad \text{subject to } J_g^\pi(x) \geq b \tag{2}$$

Let $\pi^*$ be an optimal stationary solution of the above problem. Note that both $J_r^\pi$ and $J_g^\pi$ depend on the initial state $x$. Also note that our approach can be extended to multiple constraints, and the constraints of the form $J_g^\pi(x) \leq b$.

Unlike the episodic set-up where the sub-linear regret is possible, it is known that even in the *unconstrained* set-up, the necessary condition for sub-linear regret is that the optimal policy has a long-term average reward and utility that are independent of the initial state (Bartlett and Tewari, 2012). Hence, we assume the following:

**Assumption 1.** *There exists $J_r^*$, $J_g^*$ and a stationary $\pi^*$ such that they are solution of (2) and the followings hold for all $\diamond = r, g$, $x \in \mathcal{S}$, and $a \in \mathcal{A}$*

$$J_\diamond^* + q_\diamond^*(x, a) = \diamond(x, a) + \mathbb{E}_{x' \sim \mathbb{P}(\cdot|x, a)}[v_\diamond^*(x')], v_\diamond^*(x) = \sum_a \pi^*(a|x) q_\diamond^*(x, a) \tag{3}$$

We denote $J_r^*$ as the optimal gain. We also denote the span of $v_\diamond^*$ as $\text{sp}(v_\diamond^*) = \sup_s v_\diamond^*(s) - \inf_s v_\diamond^*(s)$ for $\diamond = r, g$. For a finite state and action space (a.k.a. tabular case), the weakly communicating MDP is the broadest class to study the regret minimization in the literature, and is known to satisfy Assumption 1. Assumption 1 is adaptation of the Bellman's optimality equation for unconstrained version (Wei et al., 2021a) in the CMDP. *Note that unlike the unconstrained case, here the optimal policy might not be greedy with respect to $q_r$ as the greedy policy might not be feasible.*

**Bellman's equation with state-independent average reward and average utility**: A stationary policy $\pi$ satisfies the Bellman's equation with state-independent average reward and average utility if there exists measurable functions $v_\diamond^\pi : \mathcal{S} \to \mathbb{R}$, $q_\diamond^\pi : \mathcal{S} \times \mathcal{A} \to \mathbb{R}$ such that the following holds for all $x \in \mathcal{S}, \diamond = r, g$, and $a \in \mathcal{A}$

$$J_\diamond^\pi + q_\diamond^\pi(x, a) = \diamond(x, a) + \mathbb{E}_{x' \sim \mathbb{P}(\cdot|x, a)}[v_\diamond^\pi(x')], v_\diamond^\pi(x) = \sum_a \pi(a|x) q_\diamond^\pi(x, a), \tag{4}$$

Wei et al. (2022) assumes that *any* stationary policy (not only the optimal policy) satisfies the Bellman's equation with state-independent average reward and average utility. In Section 3 we consider the setup where Assumption 1 is satisfied which is weaker compared to Wei et al. (2022). In Section 4 we consider a set of assumptions which entails that every stationary policy satisfies (4).

**Learning Metric**: The agent is unaware of the underlying environment, and seeks to learn the optimal policy. Hence, the agent can take action according to a non-stationary policy $\pi_t$ at a state $x_t$.

We are interested in minimizing the following metrics

$$\text{Regret}(T) = \sum_{t=1}^{T}(J_r^* - r(x_t, a_t)), \quad \text{Violation}(T) = (Tb - \sum_{t=1}^{T}g(x_t, a_t)). \tag{5}$$

The regret denotes the difference between the total reward obtained by best stationary policy and the reward obtained by the learner. The violation denotes how far the total utility achieved by the learner compared to the $Tb$. Similar metrics are employed in model-based infinite horizon average reward (Chen et al., 2022) and in the finite horizon setup (Ding et al., 2021).

**Linear Function Approximation.** To handle a possible large number of states, we consider the following linear MDPs.

**Assumption 2.** *The CMDP is a linear MDP with feature map $\phi : \mathcal{S} \times \mathcal{A} \to \mathbb{R}^d$, if there exists $d$ unknown signed measures $\mu = \{\mu^1, \ldots, \mu^d\}$ over $\mathcal{S}$ such that for any $(x, a, x') \in \mathcal{S} \times \mathcal{A} \times \mathcal{S}$,*

$$\mathbb{P}(x'|x, a) = \langle \phi(x, a), \mu(x') \rangle \tag{6}$$

*and there exists vector $\theta_r, \theta_g \in \mathbb{R}^d$ such that for any $(x, a) \in \mathcal{S} \times \mathcal{A}$,*

$$r(x, a) = \langle \phi(x, a), \theta_r \rangle \quad g(x, a) = \langle \phi(x, a), \theta_g \rangle$$

Assumption 2 adapts the definition of linear MDP in Jin et al. (2020) to the constrained case. By the above definition, the transition model, the reward, and the utility functions are linear in terms of feature map $\phi$. We remark that despite being linear, $\mathbb{P}(\cdot|x, a)$ can still have infinite degrees of freedom since $\mu(\cdot)$ is unknown. Note that tabular MDP is part of linear MDP (Jin et al., 2020). Similar Constrained Linear MDP model is also considered in Ghosh et al. (2022) (for episodic setup).

Note that Ding et al. (2021); Zhou et al. (2021) studied another related concept known as linear kernel MDP in episodic setup. In the linear kernel MDP, the transition probability is given by $\mathbb{P}(x'|x, a) = \langle \psi(x', x, a), \theta \rangle$. In general, linear MDP and linear kernel MDPs are two different classes of MDP (Zhou et al., 2021).

Under linear CMDP, we can represent the $q_\diamond^*$ function as a linear in the feature space.

**Lemma 1** (Wei et al. (2021a)). *Under Assumptions 1 and 2, there exists a fixed $w_\diamond^* \in \mathbb{R}^d$ such that $q_\diamond^*(x, a) = \phi(x, a)^T w_\diamond^*$ for $\diamond = r, g$ and all $(x, a) \in \mathcal{S} \times \mathcal{A}$ with $||w_\diamond^*||\sqrt{d}(2 + \text{sp}(v_\diamond^*))$.*

## 3   OPTIMISM-BASED ALGORITHMS

### 3.1   FIXED-POINT OPTIMIZATION WITH OPTIMISM

We present our first algorithm (Algorithm 1) which achieves $\tilde{\mathcal{O}}(\text{sp}(v_r^*)\sqrt{d^3T})$ regret and $\mathcal{O}(\text{sp}(v_g^*)\sqrt{d^3T})$ constraint violation. Note that for the unconstrained tabular version, $\Omega(\sqrt{T})$ is the lower bound for regret (Auer et al., 2008). Hence, our obtained bound is optimal in terms of regret (nearly). Further, we show that we can attain zero violation while maintaining the same order in regret with an additional assumption (Remark 1). In the following we describe our Algorithm and point out the key modifications we have made over the unconstrained version.

From Lemma 1 and (3), we have

$$\phi(x, a)^T w_\diamond^* = \diamond(x, a) - J_\diamond^* + \int_{\mathcal{X}} v_\diamond^*(x')p(dx'|x, a)$$

Hence, the natural idea is to build an estimator $w_{\diamond,t}$ for $w_\diamond^*$ from the data. However, we do not know $J_\diamond^*$, $p$, $r$, and $g$. Further, we also need to find $\pi$ in order to obtain $v_\diamond^*$ (cf.(3)). In the following, we describe how we address those challenges. First, in order to handle unknown $p$, and $\diamond$, we use the sample points till time $t$ to estimate $w_{\diamond,t}$. In particular, if for a moment, if we assume $J_\diamond^*$ and $\pi$ are known then we can fit $w_{\diamond,t}$ such that $\phi(x, a)^T w_{\diamond,t} \approx \diamond(x_\tau, a_\tau) - J_\diamond^* + v_{\diamond,t}(x_{\tau+1})$. Then solving a regularized least square problem with regularization term $\lambda||w_{\diamond,t}||^2$ gives a natural estimate of $w_{\diamond,t}-$

$$w_{\diamond,t} = (\Lambda_t)^{-1}\sum_{\tau=1}^{t-1}(\phi(x_\tau, a_\tau)(\diamond(x_\tau, a_\tau) - J_\diamond^* + v_{\diamond,t}(x_{\tau+1})) \tag{7}$$

where $\Lambda_t = \lambda \mathbf{I} + \sum_{\tau < t} \phi(x_\tau, a_\tau) \phi(x_\tau, a_\tau)$ is the empirical co-variance matrix.

Now, we replace $v_{\diamond,t}(x_{\tau+1})$ with the expression $v_{\diamond,t}(x_{\tau+1}) = \langle \pi(\cdot|x_{\tau+1}), q_{\diamond,t}(x_{\tau+1}, \cdot) \rangle_{\mathcal{A}}$ which gives rise a fixed-point equation as $q_{\diamond,t}(\cdot, \cdot) = \phi(\cdot, \cdot)^T w_{\diamond,t}$. In order to incorporate uncertainty, we introduce a slack variable $b_{\diamond,t}$ with a bounded quadratic form $||b_{\diamond,t}||_{\Lambda_t} \leq \beta_\diamond$ ($\beta_\diamond$ is a parameter for $\diamond = r, g$). It will ensure that $w_\diamond^*$ is contained within $w_{\diamond,t} + b_{\diamond,t}$ with high-probability. Now, we discuss how we handle the unknown $J_\diamond^*$ and $\pi$.

We introduce $J_{\diamond,t}$ and $\pi_t$ as variables and design an optimization problem $\mathcal{P}_1$ which maximizes $J_{r,t}$ with the constraint $J_{g,t} \geq b$. We also restrict $\pi$ to a class of soft-max policy $\Pi$ (Definition 1) and search for $\pi_t$ over $\Pi$.

**Definition 1.** *We define the policy class as $\Pi$ as the collection of soft-max policies $\Pi = \{\pi(\cdot|\cdot) \propto \exp(\phi(\cdot, \cdot)^T \zeta), \zeta \in \mathbb{R}^d, ||\zeta|| \leq L\}$.*

We replace $J_\diamond^*$ in (7) with $J_{\diamond,t}$. Further, we add the slack variable $b_{\diamond,t}$ to $w_{\diamond,t}$ in (7) to obtain (8) in Algorithm 1. This has similarity with the "optimism under uncertainty" as the optimal $(w_\diamond^*, b_\diamond^*, J_\diamond^*, \pi^*)$ for $\diamond = r, g$ are feasible for the optimization problem with high probability (Lemma 2) if $\pi^* \in \Pi$. Hence, it naturally entails that $J_{r,t} \geq J_r^*$. Wei et al. (2021a) also proposed an optimization problem with fixed-point equation (Algorithm 1) to obtain $w_r^*$ in the unconstrained setup. In the following, we point out the main differences with the Algorithm 1 proposed in Wei et al. (2021a) and describe the necessity of the introduction of policy space $\Pi$.

First, we have an additional constraint where $J_{g,t} \geq b$ since we consider CMDP. Second and more importantly, in the unconstrained case (Wei et al., 2021a) the policy was greedy with respect to $q_r$. Since the greedy policy may not be feasible, we need to search over policy in the CMDP. However, we restrict the policy to function class $\Pi$. The main reason for introduction to $\Pi$ is that the uniform concentration bound (a key step for proving both regret and violation for model-free approach) for value function class $v_\diamond$ can not be achieved unless the policy has smooth properties such as Lipschitz continuous. In particular, we need to show that $\log \epsilon$ covering number for the class of $v_\diamond$ scales at most $\mathcal{O}(\log(T))$. For the unconstrained case, we only need to obtain $\epsilon$-covering number for $v_r$ and with greedy policy, $\epsilon$-covering number for the class of $q$-function was enough. However, we need to obtain $\epsilon$-covering number for both $v_r$ and $v_g$, hence, such a trick would not hold. In order to obtain $\epsilon$-covering number for $v_\diamond$, we use the smoothness of soft-max to show that $\log \epsilon$-covering number for the class of value function scales at most $\mathcal{O}(\log(T))$ (Lemma 7). In Appendix F.6 our analysis can be extended to the class of policies which satisfy Lemma 8.

Also note that we update the parameters when the determinant of $\Lambda_t$ doubles compared to $\Lambda_{s_{t-1}}$ where $s_{t-1}$ is the time step with the most recent update before the time $t$. This can happen only at most $\mathcal{O}(d \log T)$ times. Similar to Wei et al. (2021a), our algorithm also does not allow any efficient implementation due to the complicated nature of the optimization problem. However, we can show that with high probability, the optimal solution and parameters are feasible of $\mathcal{P}_1$ by the careful choice of the parameter $\beta_\diamond$, and using the fact that $||w_\diamond^*|| \leq \text{sp}(v_\diamond^*)\sqrt{d}$.

**Lemma 2.** *With probability $1 - 2\delta$, Algorithm 1 ensures that $J_{r,t} \geq J_r^*$ if $\pi^* \in \Pi$.*

Using Lemma 2 we obtain the regret and violation bound for Algorithm 1 in the following

**Theorem 1.** *With probability $1 - 5\delta$, the regret and violation bounds are*

$$\text{Regret}(T) \leq \mathcal{O}(\text{sp}(v_r^*) \log(T/\delta)\sqrt{d^3 T}), \quad \text{Violation}(T) \leq \mathcal{O}(\text{sp}(v_g^*) \log(T/\delta)\sqrt{d^3 T})$$

*if the optimal policy $\pi^* \in \Pi$.*

This is the first result which shows that $\tilde{\mathcal{O}}(\sqrt{T})$ regret and violation is achievable for infinite horizon average reward linear CMDP under Assumption 1 *using model-free approach*. Recently, Chen et al. (2022) obtained $\tilde{\mathcal{O}}(\sqrt{T})$ regret and violation for model-based tabular setup for weakly communicating MDP. The proposed algorithm (Algorithm 4) in Chen et al. (2022) is also computationally inefficient. Chen et al. (2022) also requires the knowledge of $\text{sp}(v_\diamond^*)$ for $\diamond = r, g$.

For the unconstrained MDP, the lower bound of regret is $\Omega(\sqrt{T})$ (proved for tabular setup, (Auer et al., 2008)). Hence, the regret order is optimal with respect to $T$. We also show that it is possible to zero violation for large $T$ under an additional set of Assumption (Remark 1).

---

**Algorithm 1** Model Free Fixed-point Algorithm for Long-term Average Reward and Constraint

1: **Initialization:** $\beta_\diamond = 2d(2 + \mathrm{sp}(v_\diamond^*))\sqrt{\log(32Ld(2 + \mathrm{sp}(v_\diamond^*))T/\delta)}$ for $\diamond = r, g$.
2: **for** $t = 1, \ldots, T$ **do**
3:     **if** $t = 1$ or $\det(\Lambda_t) \geq 2\Lambda_{s_{t-1}}$ **then**
4:         Set $s_t = t$.
5:         Obtain $w_{\diamond,t}$ for $\diamond = r, g$ as the solution of the following optimization problem $\mathcal{P}_1$:

$$\max_{w_{r,t}, w_{g,t}, b_{r,t}, b_{g,t}, \pi_t, J_{r,t}, J_{g,t}} J_{r,t}$$

$$\text{s.t. } w_{\diamond,t} = (\Lambda_t)^{-1} \sum_{\tau=1}^{t-1} (\phi(x_\tau, a_\tau)(\diamond(x_\tau, a_\tau) - J_{\diamond,t} + v_{\diamond,t}(x_{\tau+1})) + b_{j,t}), \diamond = r, g \quad (8)$$

$$J_{g,t} \geq b, \quad q_{\diamond,t}(x,a) = \phi(x,a)^T w_{\diamond,t}, v_{\diamond,t}(x) = \langle \pi_t, q_{\diamond,t} \rangle, \quad \pi_t \in \Pi$$

$$||b_{\diamond,t}||_{\Lambda_t^{-1}} \leq \beta_\diamond, \quad ||w_{\diamond,t}|| \leq (2 + \mathrm{sp}(v_\diamond^*))\sqrt{d}$$

6:     Take action $a_t \sim \pi_t(\cdot|x_t)$ and observe $x_{t+1}$.
7:     Update $\Lambda_{t+1} = \Lambda_t + \phi(x_t, a_t)\phi(x_t, a_t)^T$

---

### 3.2 Algorithm based on Episodic Primal-Dual Algorithm

We now present another optimism-based algorithm which can be implemented efficiently albeit with a sub-optimal regret and violation guarantee under the Slater's condition (Assumption 3).

**Assumption 3.** *We assume the following, there exists $\bar{\pi}$, $v^{\bar{\pi}}$, $q^{\bar{\pi}}$, $J_r^{\bar{\pi}}$, and $J_g^{\bar{\pi}}$ such that (4) is satisfied; and $J_g^{\bar{\pi}} \geq b + \gamma$ with $\gamma > 0$.*

The above assumption states that there exists a stationary policy with strict feasibility. Strict feasibility assumption is common both in the infinite horizon average reward (Wei et al., 2022; Chen et al., 2022) and finite horizon episodic CMDP set up (Efroni et al., 2020; Ding et al., 2021). We only need to know $\gamma$ rather than the strictly feasible policy.

In algorithm 1 the inefficiency arises since we do not know how to efficiently solve a fixed-point equation and find the policy to solve the optimization problem. We show that we eliminate the above inefficiencies by considering a finite-horizon problem. In particular, we divide the entire horizon $T$ into $T/H$ episodes with $H$ rounds and employ a finite-horizon primal dual optimistic LSVI-based algorithm proposed in Ghosh et al. (2022). However, since the original problem is still infinite horizon we need to bound the gap because of the finite-horizon truncation which we describe in the following.

#### 3.2.1 Our Approach

We replace the time index $t$ with the combination of the episode index $k$ and the step index $h$, $t = (k-1)H + h$. The $k$-th episode starts at time $(k-1)H + 1$. The agent chooses the policy $\pi_h^k$ for the step $h \in [H]$ at episode $k \in [K]$. Note that for finite-horizon setting, the policy needs not to be stationary even though the optimal policy for the infinite horizon problem is stationary. As we mentioned, the idea is here to obtain policy with good performance guarantee in the episodic setup and then bound gap between the infinite horizon and episodic setup.

We now introduce the notations specific to the episodic setting.

$$V_{\diamond,h}^\pi(x) = \mathbb{E}_\pi\left[\sum_{i=h}^{H} \diamond(x_i, a_i)|x_h = x\right], \quad Q_{\diamond,h}^\pi(x,a) = \mathbb{E}_\pi\left[\sum_{i=h}^{H} \diamond(x_i, a_i)|x_h = x, a_h = a\right].$$

Note that since $J_g^\pi \geq b$, the natural choice of the constraint in the episodic CMDP would be $V_{g,1}^\pi \geq Hb$. However, in order to compare the best policy in the episodic setup with the optimal policy in the infinite horizon setup we need to ensure that the optimal stationary policy for the infinite horizon problem is feasible in the episodic setup. If we have $V_{g,1}^\pi \geq Hb - \mathrm{sp}(v_g^*)$ we can conclude that the optimal stationary policy $\pi^*$ is also feasible for the episodic setup using the following lemma:

**Lemma 3.** *If $J_\diamond^\pi(s) = J_\diamond^\pi$ for $\diamond = r, g$, and any state $s$, stationary policy $\pi$, then*

$$|V_{\diamond,h}^\pi - (H - h + 1)J_\diamond^\pi| \leq \mathrm{sp}(v_\diamond^\pi) \quad (9)$$

Further, in order to apply the primal-dual algorithm we also need to show that there exists a strictly feasible policy for the episodic setup. Again using Lemma 3 and from Assumption 3, we conclude that there exists a strictly feasible policy for the episodic setup if $V_{g,1}^\pi \geq Hb + \gamma - \text{sp}(v_g^{\bar{\pi}})$. Thus, we consider the following constraint $V_{g,1}^\pi \geq Hb - \kappa$ where $\kappa = \max\{\text{sp}(v_g^*), \text{sp}(v_g^{\bar{\pi}}) - \gamma\}$.

**Lagrangian**: We consider the composite value function as $V_h^{\pi,Y}(x) = V_{r,h}^\pi(x) + Y V_{g,h}^\pi(x)$.

The following is proved in Ding et al. (2021).

**Lemma 4** (Boundedness of $Y^*$). *The optimal dual-variable* $Y^* \leq \dfrac{V_r^{\tilde{\pi}^*}(x_1) - V_r^{\bar{\pi}}(x_1)}{H\gamma} \leq \dfrac{H}{H\gamma}$

*where $\bar{\pi}^*$ is the optimal solution of the episodic CMDP.*

**Definition 2.** *We set $\xi = 2/\gamma$. $\xi$ is used to truncate the dual variable.*

### 3.2.2 ALGORITHM

We now describe our proposed Algorithm 2 (Appendix D). First, note that the finite-horizon state-action value function for linear MDP can be represented as the following linear form

$$Q_{\diamond,h}^\pi(x,a) = \phi(x,a)^T w_{\diamond,h}^\pi = \diamond_h(x,a) + \int_\mathcal{X} V_{\diamond,h+1}(x') p(dx'|x,a) \tag{10}$$

Hence, the natural estimates for the parameters $w_{r,h}^k, w_{g,h}^k$ are obtained by solving the following regularized least square problem

$$w_{\diamond,h}^k \leftarrow \arg\min_{w \in \mathbb{R}^d} \sum_{\tau=1}^{k-1} \sum_{h'} [\diamond(x_{h'}^\tau, a_{h'}^\tau) + V_{\diamond,h'+1}^k(x_{h'+1}^\tau) - w^T \phi(x_{h'}^\tau, a_{h'}^\tau)]^2 + \lambda ||w||_2^2, \tag{11}$$

$Q_{\diamond,h}^k(x,a) = \langle w_{\diamond,h}^k, \phi(x,a)\rangle + \beta ||\phi(x,a)||_{\Lambda_k^{-1}}$ where $\beta ||\phi(x,a)||_{\Lambda_k^{-1}}$ is the bonus term. Since we are considering a finite-horizon, hence, we can recursively set $V_{\diamond,h+1}$ starting from $V_{\diamond,H+1} = 0$. Thus, we do not need to solve the fixed-point equation unlike in Algorithm 1. Another difference with Algorithm 1 is that we introduce a pointwise bonus term for optimism. Finally, we use a primal-dual adaptation unlike in Algorithm 1.

The policy is based on the soft-max. In particular, at step $h$, $\pi_{h,k}(a|x)$ is computed based on the soft-max policy on the composite $Q$-function vector $\{Q_{r,h}^k(x,a) + Y_k Q_{g,h}^k(x,a)\}_{a \in \mathcal{A}}$ where $Y_k$ is the lagrangian multiplier. The computation of policy is also efficient unlike in Algorithm 1 as we do not need to search over the policy space. Note that the greedy policy with respect to the composite $Q$-function vector fails to show that log $\epsilon$-covering number for the individual value function class is at most $\log(T)$ (Ghosh et al., 2022). Note that Ghosh et al. (2022) maintains different covariance matrices at each step $h$, $\Lambda_h^k$. But we maintain a single $\Lambda^k$ for all $h$ since the transition probability and reward are the same across the steps. The last part (Step 15) of Algorithm 2 includes a dual-update step. The dual variable increases if the estimated utility value function does not satisfy the constraint.

### 3.2.3 MAIN RESULTS

**Theorem 2.** *Under Assumptions 1, 2, and 3, with probability $1 - 5\delta$, the regret and constraint violations are*

$$\text{Regret}(T) \leq \mathcal{O}((1 + \text{sp}(v_r^*))(dT)^{3/4}\iota), \quad \text{Violation}(T) \leq \frac{2(1+\xi)}{\xi}\tilde{\mathcal{O}}((1 + \kappa)(dT)^{3/4}\iota)$$

*where $\iota = \log(\log(|\mathcal{A}|)\xi 2dT/\delta)$.*

Unlike in Theorem 1, here, the optimal policy does not need to belong to $\Pi$. Wei et al. (2022) proved that the regret bound is $\tilde{\mathcal{O}}(T^{5/6})$ for the model-free RL algorithm in the tabular regime under a much stronger Assumption compared to ours. Since linear CMDP contains a tabular setup, our approach improves the upper bound for the tabular case as well. We show that it is possible to achieve zero violation while maintaining the same order on regret under an additional Assumption (Remark 1).

Note that the regret is also $\tilde{\mathcal{O}}((dT)^{3/4})$ for the unconstrained linear MDP (Wei et al., 2021a) for the efficient implementation variant of the algorithm.

*Analysis*: The overall regret can be decomposed in

$$\text{Regret}(T) = \sum_{k=1}^{T/H} (HJ_r^* - V_{r,1}^{\pi_k^*}(x_1^k)) + \sum_{k=1}^{T/H} (V_{r,1}^{\pi_k^*}(x_1^k) - V_{r,1}^{\pi_k}(x_1^k)) + \sum_{k=1}^{T/H} (V_{r,1}^{\pi_k}(x_1^k) - \sum_{t=(k-1)H+1}^{kH} r(x_t, a_t))$$

The third term on the right-hand side can be bounded using the Azuma-Hoeffding inequality. Hence, we focus on the first two terms of the right-hand side. The second part is exactly the same as the regret for the episodic scenario. *However, there is a subtle difference.* We need to show that optimal policy for the original infinite horizon is also feasible (see Appendix G.1) which we show using the fact that we have relaxed the constraint by $\kappa$. We show that the above is upper bounded by $\tilde{\mathcal{O}}(\sqrt{d^3 T H^2})$. Note that compared to Ghosh et al. (2022), we have $\sqrt{H}$ improvement as in our case the transition probability is independent of the time $h$ unlike in Ghosh et al. (2022).

One may wonder why not set the length $H$ too small. However, since our original problem is about average-reward over infinite horizon, we need to argue that the best finite-horizon policy also performs well under the infinite-horizon criteria. The first term characterizes the gap. We show that the sub-optimality gap to the best finite horizon policy is bounded by $T\text{sp}(v_r^*)/H$ using Lemma 3. Hence, if we set too small $H$, the gap would increase. By setting $H = T^{1/4}/d^{3/4}$ we balance the upper bounds on the first two terms of the right-hand side. As $T$ increases, $H$ also increases.

**Constraint Violation**: The violation term can also be decomposed as follows

$$\text{Violation}(T) = \sum_{k=1}^{T/H} (Hb - \kappa - V_{g,1}^{\pi_k}(x_1^k)) + (V_{g,1}^{\pi_k}(x_1^k) - \sum_{t=(k-1)H+1}^{kH} g(x_t, a_t)) + (T/H)\kappa$$

We obtain the upper bound of the first term as $\tilde{\mathcal{O}}(\sqrt{d^3 H^2 T})$. The second term can be bounded by Azuma-Hoeffding inequality. The third term denotes the error incurred due to the introduction of $\kappa$. We obtain the final constraint violation bound by replacing $H$ with $T^{1/4}/d^{3/4}$.

## 4 POLICY-BASED ALGORITHM

Even though Algorithm 2 is computationally efficient, still one needs to compute $V_{\diamond,h+1}^k$ for every encountered state and needs to evaluate $(\Lambda_k)^{-1}$. We now show that it is possible to achieve $\tilde{\mathcal{O}}(\sqrt{T})$ regret and constraint violation under a different and relaxed set of assumptions using a computationally less intensive algorithm. We propose a policy-based algorithm towards this end. First, we state the assumptions and then we describe the algorithm and the results.

**Assumption 4.** *There exists a constant $t_{mix} \geq 1$ such that for any policy $\pi$, and any distribution $\nu_1, \nu_2 \in \Delta_{\mathcal{X}}$ over the state-space $||\mathbb{P}^\pi \nu_1 - \mathbb{P}^\pi \nu_2||_{TV} \leq e^{-1/t_{mix}}||\nu_1 - \nu_2||_{TV}$ where $\mathbb{P}^\pi \nu(x') = \int_{\mathcal{X}} \sum_a \pi(a|x)\mathbb{P}(x'|x,a)d\nu(x)$, $TV$ is the total variation distance.*

Ergodic MDP satisfies the above assumption Hao et al. (2021). With the above assumption, we can conclude the following, as shown in the unconstrained setup (Wei et al., 2021a)).

**Lemma 5.** *Under Assumption 4, any stationary policy $\pi$ satisfies (4).*

Hence, under Assumption 4, the gain is constant, and any stationary policy indeed satisfies (4).

**Assumption 5.** *Let $\lambda_{min}(\cdot)$ be minimum eigen value of a matrix. There exists $\sigma > 0$ such that for any $\pi$, $\lambda_{min}\left(\int_{\mathcal{X}}(\sum_a \pi(a|x)\phi(x,a)\phi(x,a)^T)d\nu^\pi(x)\right) \geq \sigma$*

The assumption intuitively guarantees that every policy is explorative in the feature space. similar assumptions are also made (Wei et al., 2021a; Abbasi-Yadkori et al., 2019) in the unconstrained case. Similar to Wei et al. (2021a) we can relax this assumption to the setting where the above holds for *one* known policy instead of all policies.

**Lagrangian**: $J_r^\pi + Y J_g^\pi$ is the Lagrangian for the stationary policy $\pi$ where $Y$ is the dual variable. Under Assumption 3, optimal dual-variable $Y^* \leq \dfrac{J_r^{\pi^*}(x_1) - J_r^{\bar{\pi}}(x_1)}{\gamma} \leq \dfrac{1}{\gamma} \leq \xi/2.$

### 4.1 ALGORITHM

We consider a primal-dual adaptation of MDP-EXP2 in Algorithm 3 (Appendix E). The algorithm proceeds in epochs where each epoch consists of $B = \tilde{\mathcal{O}}(dt_{mix}/\sigma)$ steps. Within each epoch, we collect $B/2N$ trajectories where $N = \tilde{\mathcal{O}}(t_{mix})$. The update of policy $\pi_k$ for $k$-th epoch is equivalent to the online mirror descent (OMD), i.e.,

$$\pi_k(\cdot|\cdot) = \arg\max_{\pi}\langle\pi, q_r^{k-1}(\cdot,\cdot) + Y_{k-1}q_g^{k-1}(\cdot,\cdot)\rangle - \frac{1}{\alpha}D(\pi||\pi_{k-1}) \tag{12}$$

where $q_\diamond^{k-1}(x,a) = (w_\diamond^{k-1})^T\phi(x,a)$. We consider a composite $q$-function which is the sum of the $q_r$ and $q_g$ (scaled by the dual-variable). We estimate $w_\diamond^k$ from sample average reward $R_{k,m}$ and sample average utility $G_{k,m}$ by utilizing Lemma 5 and the fact that $q_\diamond^\pi$ is linear in feature space.

Since we consider primal-dual adaptation of MDP-EXP2 in Wei et al. (2021a), we maintain two different types of sample averages for the utility. $G_{k,m}$ is used to estimate $q_g^{\pi_k} + NJ_g^{\pi_k}$; on the other hand, $\hat{J}^k$ is used to estimate $J_g^{\pi_k}$. The dual update is based on the value of $\hat{J}^k$. Naturally, the analysis also differs significantly (see Appendix H.1).

### 4.2 MAIN RESULTS

**Theorem 3.** *Under Assumptions 2, 4,5, and 3, Algorithm 2 ensures that with probability $1 - 4\delta$,*

$$\text{Regret}(T) \leq (1 + \xi)\tilde{\mathcal{O}}(1/\sigma\sqrt{t_{mix}^3 T}), \quad \text{Violation}(T) \leq \frac{2(1+\xi)}{\xi}\tilde{\mathcal{O}}(1/\sigma\sqrt{t_{mix}^3 T}) \tag{13}$$

Wei et al. (2021a) shows that $\tilde{\mathcal{O}}(\sqrt{T})$ regret is achieved in the *expectation* under Assumptions 4, and 5 in the *unconstrained* setup. We show that even with a high probability, such a regret can be achieved along with $\tilde{\mathcal{O}}(\sqrt{T})$ of violation. Assumption 2 can be relaxed to the scenarios where $q_\diamond^\pi$ is linear (even though the underlying MDP is not linear) similar to unconstrained setup (Assumption 4 in Wei et al. (2021a)). Hence, $\tilde{\mathcal{O}}(\sqrt{T})$ regret and violation can be achieved for any linear $q^\pi$ even if the underlying MDP is not linear. In the model-based tabular CMDP, Chen et al. (2022) shows that the regret also scales with $\tilde{\mathcal{O}}(\sqrt{T})$ in the ergodic MDP. Also note that the regret scales with $t_{mix}$ and $\xi$ there as well (order of $t_{mix}$ is worst there). Note that the dependence on $d$ is implicit as $1/\sigma = \Omega(d)$.

We need to know $t_{mix}$, and $\sigma$ in Algorithm 3. Similar to the unconstrained setup Wei et al. (2021a), we can modify our algorithm to incorporate unknown $t_{mix}$ and $\sigma$ with slightly worsened regret.

**Remark 1.** *We can reduce the violation to $0$ with sacrificing the regret a little bit (the order is the same) under the Assumptions in Section 4 (Appendix H.5). The idea is the following, we consider a tighter optimization problem where we add $\epsilon$ in the right-hand side of the constraint in (2). We then bound the difference of the optimal gains in the tighter problem and the original problem as a function of $\epsilon$. The bound on regret and violation can be attained following our analysis for the tighter optimization problem since it is a CMDP with $b + \epsilon$ in place of $b$. Hence, by carefully choosing $\epsilon$, we can show that it is strictly feasible, and achieves zero violation for large enough $T$ (Lemma 36).*

*For the assumption in Section 3.2 and Section 3.1, we need an additional assumption (any stationary policy satisfies Bellman's equation (cf.(4)) to obtain zero constraint violation (Appendix H.5). The idea is similar to the one as described in earlier paragraph. The additional assumption would help us to bound the difference between the tighter and the original problem.*

## 5 EXPERIMENTAL RESULTS

We conduct numerical experiments to validate our theory on an environment similar to Chen et al. (2022). We implement Algorithm 3 and observe that regret and violation indeed grow sub-linearly even without the knowledge of $t_{mix}$, and $\sigma$. Please see Appendix I for details.

## 6 CONCLUSION AND FUTURE WORK

In this work, we provide model-free RL algorithms under two different sets of Assumptions. To the best of our knowledge, ours is the first work on the average reward-constrained linear MDP. Our results improve the regret and violation bound even for the finite horizon tabular model-free constrained RL setup. Whether we can develop a computationally efficient algorithm that achieves optimal regret under Assumption 1 constitutes a future research question. Whether we can extend the analysis to the non-linear function approximation is also an open question.

ACKNOWLEDGMENTS

This work has been partly supported by NSF grants NSF AI Institute (AI-EDGE) 2112471, CNS-2106933, 2007231, CNS-1955535, and CNS-1901057, and in part by Army Research Office under Grant W911NF-21-1-0244.

Xingyu Zhou is supported in part by NSF CNS-2153220.

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

**Organization of Appendix** We provide related work in Appendix A. In Appendix B, we provide some motivating examples. We summarize our results in Appendix C under various sets of Assumptions. We describe Algorithm 2 in Appendix D. We formally describe Algorithm 3 in Appendix E. We prove the results of Section 3.1 in Appendix F. We prove the results of Section 3.2 in Appendix G. We prove the results of Section 4 in Appendix H. We show how Algorithm 3 can achieve zero constraint violation while maintaining the same order on the regret bound (with respect to $T$) in Appendix H.5. In Appendix H.5 we also show how Algorithms 1 and 2 achieve zero violation while maintaining the same order on regret (with respect to $T$) with an additional assumption. We show numerical experiments in Appendix I.

## A  RELATED WORK

Model-free RL algorithms have been proposed (Xu et al., 2021; Ding et al., 2020; Bai et al., 2021) to solve CMDP in the discounted infinite horizon setting. All of these works consider an easier setting compared to standard RL in that they assume access to a simulator (Koenig and Simmons, 1993) (a.k.a. a generative model (Azar et al., 2012)), which is a strong oracle that allows the agent to query arbitrary state-action pairs and return the reward and the next state, hence greatly alleviating the intrinsic difficulty of exploration in RL. Wei et al. (2021b) proposed a 'triple-Q' algorithm that does not require a 'simulator'. However, it only considered the tabular setting and the episodic set-up.

To develop online sample-efficient algorithms for CMDPs, prior works have largely resorted to the finite horizon setup (Efroni et al., 2020; Brantley et al., 2020; Kalagarla et al., 2020; Liu et al., 2021). Model-based RL in the episodic linear kernel CMDP is considered in Ding et al. (2021). Model-based infinite horizon average reward CMDP for tabular setup is also considered in Zheng and Ratliff (2020); Singh et al. (2020).

## B  MOTIVATING EXAMPLES

First, we provide some examples of constrained MDP.

- Consider an intelligent agent taking an action to optimize the power consumption for a household. It would seek to minimize the overall cost while trying to maintain a minimum level of satisfaction (e.g., maintaining a certain temp., maintaining the charge of the electric vehicles). Here the reward can be cast as the negative of the cost for power consumption, while utility can be modeled as the satisfaction the user gets.

- As another example, consider a sensor network where sensor nodes sample and send information to a server (fusion center) for processing. However, these sensor nodes also need to satisfy the energy constraint as they have limited battery capacity. Such a decision process can be modeled as CMDP where (i) the reward depends on the nature of the information and whether the information is successfully received or not and (ii) the cost corresponds to the cost for sampling and transmitting information.

We now provide some examples of CMDP that might run for a long-time where the average reward CMDP would be the ideal candidate for modeling.

- Consider the sensor network example provided earlier. Here, the sensor node continually takes decisions, and thus, an infinite horizon average reward CMDP is a natural choice.

- As another example, consider that a server is scheduling jobs to different machines. The scheduler seeks to minimize the job completion time. The server wants to maintain a uniform queue length (i.e, the number of jobs waiting to be processed) across the machines which can only process jobs sequentially. The server is continually taking decisions, thus, it is maximizing the average reward while maintaining an average queue length below a certain threshold. This can be cast as the average reward CMDP. Another example is a controller in an autonomous system that takes a decision in order to maximize the average reward (objective) while trying to maintain system stability. We can model the system as an average reward CMDP where there is a utility of 1 if the system remains in the safe region at every step. The goal of the controller is to maximize the average reward while the system will be in one of the safe states for at least $1 - \epsilon$ fraction of times for the desired choice of $\epsilon > 0$. This can be cast as average reward CMDP where the controller seeks to maximize the average reward while the average utility is at least

Finally, we argue why we choose the function approximation setup. In many examples, provided above, the state space is continuous or at least very large. For example, the state of the battery for a sensor is continuous. Similarly, the length of the queue can be very large. Function approximation is generally used to approximate the Q-function or policy in such a large state space. We consider linear function approximation in our setup as a first step to handle the large state space.

## C SUMMARY OF OUR RESULTS IN A TABLE

We summarize our results in Table 1.

Table 1: Regret and Constraint Violations on Linear MDP for our proposed algorithms

| ALGORITHM | REGRET | VIOLATIONS | ASSUMPTIONS |
|---|---|---|---|
| ALGORITHM 1 | $\tilde{\mathcal{O}}(\sqrt{d^3 T})$ | $\tilde{\mathcal{O}}(\sqrt{d^3 T})^+$ | ASSUMPTION 1 |
| ALGORITHM 2 | $\tilde{\mathcal{O}}((dT)^{3/4})$ | $\tilde{\mathcal{O}}((dT)^{3/4})^+$ | ASSUMPTIONS 1, AND 3 |
| ALGORITHM 3 | $\tilde{\mathcal{O}}(\sqrt{T})$ | $\tilde{\mathcal{O}}(\sqrt{T})^*$ | ASSUMPTIONS 3,4, AND 5 |

$^*$ WE CAN REDUCE THE VIOLATION TO 0 FOR LARGE ENOUGH $T$ (AP-PENDIX H.5) WHILE MAINTAINING THE SAME ORDER OF REGRET WITH RESPECT TO $T$.
$^+$ WE CAN REDUCE THE VIOLATION TO 0 FOR ALGORITHMS 1 AND 2 IF WE ASSUME THAT ANY STATIONARY POLICY SATISFIES (4) (APPENDIX H.5).

## D ALGORITHM 2

Here, we state Algorithm 2.

---

**Algorithm 2** Model Free Primal-Dual Algorithm for Long-term Average Reward and Constraint

---

1: **Initialization:** $Y_1 = 0$, $w_{\diamond,h} = 0$, $H = \dfrac{T^{1/4}}{d^{3/4}}$, $K = T/H$, $\alpha = \dfrac{\log(|\mathcal{A}|)\sqrt{KH}}{2(1+\xi+H)}$, $\eta = \xi/\sqrt{KH^2}$, $\beta = C_1 dH\sqrt{\log(2\log|\mathcal{A}|\xi dT/\delta)}$
2: **for** episodes $k = 1, \ldots, K$ **do**
3:     Receive the initial state $x_1^k$.
4:     $\Lambda^k \leftarrow \sum_{\tau=1}^{k-1}\sum_h \phi(x_h^\tau, a_h^\tau)\phi(x_h^\tau, a_h^\tau)^T + \lambda\mathbf{I}$
5:     **for** step $h = H, H-1, \ldots, 1$ **do**
6:         $w_{r,h}^k \leftarrow (\Lambda^k)^{-1}[\sum_{\tau=1}^{k-1}\sum_{h'=1}^H \phi(x_{h'}^\tau, a_h^\tau)[r(x_{h'}^\tau, a_{h'}^\tau) + V_{r,h+1}^k(x_{h'+1}^\tau)]]$
7:         $w_{g,h}^k \leftarrow (\Lambda^k)^{-1}[\sum_{\tau=1}^{k-1}\sum_{h'=1}^H \phi(x_{h'}^\tau, a_{h'}^\tau)[g(x_{h'}^\tau, a_{h'}^\tau) + V_{g,h'+1}^k(x_{h'+1}^\tau)]]$
8:         $Q_{r,h}^k(\cdot,\cdot) \leftarrow \min\{\langle w_{r,h}^k, \phi(\cdot,\cdot)\rangle + \beta(\phi(\cdot,\cdot)^T(\Lambda^k)^{-1}\phi(\cdot,\cdot))^{1/2}, H\}$
9:         $Q_{g,h}^k(\cdot,\cdot) \leftarrow \min\{\langle w_{g,h}^k, \phi(\cdot,\cdot)\rangle + \beta(\phi(\cdot,\cdot)^T(\Lambda^k)^{-1}\phi(\cdot,\cdot))^{1/2}, H\}$
10:         $\pi_{h,k}(a|\cdot) = \dfrac{\exp(\alpha(Q_{r,h}^k(\cdot,a) + Y_k Q_{g,h}^k(\cdot,a)))}{\sum_a \exp(\alpha(Q_{r,h}^k(\cdot,a) + Y_k Q_{g,h}^k(\cdot,a)))}$
11:         $V_{r,h}^k(\cdot) = \sum_a \pi_{h,k}(a|\cdot)Q_{r,h}^k(\cdot,a)$, $V_{g,h}^k(\cdot) = \sum_a \pi_{h,k}(a|\cdot)Q_{g,h}^k(\cdot,a)$
12:     **for** step $h = 1, \ldots, H$ **do**
13:         Compute $Q_{r,h}^k(x_h^k, a)$, $Q_{g,h}^k(x_h^k, a)$, $\pi(a|x_h^k)$ for all $a$.
14:         Take action $a_h^k \sim \pi_{h,k}(\cdot|x_h^k)$ and observe $x_{h+1}^k$.
15:     $Y_{k+1} = \max\{\min\{Y_k + \eta(b - V_{g,1}^k(x_1^k)), \xi\}, 0\}$

---

## E ALGORITHM 3

Here, we describe Algorithm 3.

## F PROOF OF THE RESULT OF SECTION 3.1

**Notations**: Without loss of generality, we assume that the first element of $\phi(\cdot,\cdot)$ is 1. Hence $\phi(\cdot,\cdot)^T e_1 = 1$ where $e_1$ is a $d$-dimensional vector where the first element is 1 and the rest are 0. Also recall that $||w_\diamond^*|| \leq \sqrt{d}(2 + \mathrm{sp}(v_\diamond^*))$ from Lemma 1. As shown in Wei et al. (2021a) we

---

**Algorithm 3** Model Free Primal-Dual Algorithm for Long-term Average Reward in Linear MDP

1: **Initialization:** $Y_1 = 0$, $N = 8t_{mix}\log(T)$ $w_{\diamond,1} = 0$, $B = 32N(\log(dT))/\sigma$ $\alpha = \min\{1/(1 + \xi)\sqrt{Tt_{mix}}, \sigma/(24N(1 + \xi))\}$, $\eta = \xi/\sqrt{T/B}$.

2: **for** epochs $k = 1, \ldots, T/B$ **do**

3:     Define the policy $\pi_k(a|x)$ such that $\pi_k(a|x) \propto \exp\left(\alpha\phi(x,a)^T(\sum_{j=1}^{k-1} w_{r,j} + Y_j w_{g,j})\right)$, and initialize count $\bar{J}_k$.

4:     Execute $\pi_k$ in the entire epoch.

5:     **for** $t = (k-1)B + 1, \ldots, kB$ **do**

6:         Execute $a_t \sim \pi_k(\cdot|x_t)$, observes, $r_t(x_t, a_t)$, $g_t(x_t, a_t)$, and $x_{t+1}$.

7:         **if** $t \geq (k-1)B + N + 1$ **then**

8:             Sum the constraint $\bar{J}_k = \bar{J}_k + g(x_t, a_t)$.

9:     **for** step $m = 1, \ldots, B/2N$ **do**

10:       Define

$$\tau_{k,m} = (k-1)B + 2N(m-1) + N + 1$$

        as the start of the $m$-th trajectory.

11:       Compute

$$R_{k,m} = \sum_{t=\tau_{k,m}}^{\tau_{k,m}+N-1} r(x_t, a_t)$$

$$G_{k,m} = \sum_{t=\tau_{k,m}}^{\tau_{k,m}+N-1} g(x_t, a_t)$$

12:       Compute

$$M_k = \sum_{m=1}^{B/2N} \sum_a \pi(a|x_{\tau_{k,m}})\phi(x_{\tau_{k,m}}, a_{\tau_{k,m}})\phi(x_{\tau_{k,m}}, a_{\tau_{k,m}})^T$$

13:       **if** $\lambda_{min}(M_k) \geq B\sigma/(24N)$, **then**

14:           $w_{r,k} = M_k^{-1} \sum_{m=1}^{B/2N} \phi(x_{\tau_{k,m}}, a_{\tau_{k,m}})R_{k,m}$

15:           $w_{g,k} = M_k^{-1} \sum_{m=1}^{B/2N} \phi(x_{\tau_{k,m}}, a_{\tau_{k,m}})G_{k,m}$

16:       **else**

17:           $w_{r,k}, w_{g,k} = 0$.

18:     $\hat{J}_k = \bar{J}_k/(B - N)$

19:     $Y_{k+1} = \max\{\min\{Y_k + \eta(b - \hat{J}_k), \xi\}, 0\}$

---

have $\sup_x |v_\diamond^*(x)| \leq \frac{1}{2}\text{sp}(v_\diamond^*)$ without loss of generality. Without loss of generality, we assume $||\phi(x,a)||_2 \leq \sqrt{2}$ for all $(x,a) \in \mathcal{S} \times \mathcal{A}$, $||\mu(\mathcal{S})||_2 \leq \sqrt{d}$, $||\theta_j||_2 \leq \sqrt{d}$ for $\diamond = r, g$.

### F.1 PROOF-SKETCH OF THEOREM 1

Note from (3) that for $\diamond = r, g$

$$\sum_t (J_\diamond^* - \diamond(x_t, a_t)) = \sum_t [q_\diamond^*(x_t, a_t) - \mathbb{E}_{x' \sim p(\cdot|x_t, a_t)}[v_\diamond^*(x')]] \tag{14}$$

We then show that

**Lemma 6.** *With probability* $1 - \delta$,

$$\sum_t q_{s_t,\diamond}(x_t, a_t) - q_\diamond^*(x_t, a_t) \leq \sum_{t=1}^T (J_\diamond^* - J_{s_t,\diamond}) + \sum_{t=1}^T \mathbb{E}_{x' \sim p(\cdot|x_t, a_t)}[v_{s_t,\diamond}(x') - v_\diamond^*(x')]$$
$$+ \mathcal{O}(\beta_\diamond\sqrt{dT\log(T)}) \tag{15}$$

where $s_t$ is the last time the optimization problem is solved before time $t$. $(J_{s_t,\diamond}, b_{s_t,\diamond}, w_{s_t,\diamond}, \pi_{s_t})$ are the solutions which also specify $q_{s_t,\diamond}$ and $v_{s_t,\diamond}$.

The key step in proving the above lemma is that we need to show

$$\left\| \sum_{\tau=1}^{s_t-1} \phi(x_\tau, a_\tau)(v_{s_t,\diamond}(x') - \mathbb{E}_{x'\sim p(\cdot|x_t,a_t)} v_{s_t,\diamond}(x')) \right\|_{\Lambda_{s_t}^{-1}} \tag{16}$$

is upper bounded by $\mathcal{O}(\log T)$ with high probability. To this end, value-aware uniform concentration is required to handle the dependence between $v_{s_t,\diamond}$ and samples $\{x_\tau\}_{\tau=1}^{k-1}$, which renders the standard self-normalized inequality infeasible in the model-free setting. The general idea here is to fix a function class $\mathcal{V}_\diamond$ in advance and then show that each possible value function in our algorithm $v_{s_t,\diamond}$ is within this class which has polynomial log $\epsilon$-covering number.

The idea is to show that the function class $q_{s_t,\diamond}$ has log $\epsilon$ covering number which is at most $\mathcal{O}(d\log(1+\mathrm{sp}(v_\diamond^*)/\epsilon))$ (Lemma 7). Now, using the smoothness property of soft-max, we can also show that the policy class also has log $\epsilon$-covering number which is at most $d\log(1+16L/\epsilon)$. Combining the above we compute the $\epsilon$-covering number for the class $\mathcal{V}_\diamond$ and show that (16) is upper bounded by $\log(T)$ with high probability. The detailed proof is in Lemma 7.

Now, we divide the proof of Theorem 1 in two parts. First, we bound the regret and subsequently, we bound the violation.

*Proof of Regret*: From Lemma 2 with probability $1 - 2\delta$, $w_r^*, w_g^*, J_g^*, J_r^*, \pi^*$ are feasible solution for the optimization problem. Since we are maximizing $J_r$, thus, $J_{s_t,r} \geq J_r^*$ with probability $1 - 2\delta$. Further, from Lemma 6 with probability $1 - \delta$,

$$\sum_t q_{s_t,r}(x_t, a_t) - q_r^*(x_t, a_t) \leq \sum_t (J_r^* - J_{s_t,r}) + \sum_{t=1}^T \mathbb{E}_{x'\sim p(\cdot|x_t,a_t)}[v_{s_t,r}(x') - v_r^*(x')]$$
$$+ \mathcal{O}(\beta_r\sqrt{dT\log(T)})$$

Hence, from union bound, with probability $1 - 3\delta$,

$$\sum_t q_{s_t,r}(x_t, a_t) - q_r^*(x_t, a_t) \leq \sum_{t=1}^T \mathbb{E}_{x'\sim p(\cdot|x_t,a_t)}[v_{s_t,r}(x') - v_r^*(x')] + \mathcal{O}(\beta_r\sqrt{dT\log(T)}) \tag{17}$$

By rearranging

$$\sum_{t=1}^T (\mathbb{E}_{x'\sim p(\cdot|x_t,a_t)}[v_r^*(x')] - q_r^*(x_t, a_t))$$

$$\leq \sum_{t=1}^T (\mathbb{E}_{x'\sim p(\cdot|x_t,a_t)}[v_{s_t,r}(x')] - q_{s_t,r}(x_t, a_t)) + \mathcal{O}(\beta_r\sqrt{dT\log T})$$

$$= \sum_{t=1}^T (\mathbb{E}_{x'\sim p(\cdot|x_t,a_t)}[v_{s_t,r}(x')] - v_{s_t,r}(x_t)) + \sum_{t=1}^T (v_{s_t,r}(x_t) - q_{s_t,r}(x_t, a_t)) + \mathcal{O}(\beta_r\sqrt{dT\log T})$$

We define $\mathcal{F}_{t,2}$ as the $\sigma$-algebra generated by $[(x_\tau, a_\tau)]_{\tau=1}^t \cup x_{t+1}$. Then, $\mathbb{E}[v_{s_t,\diamond}(x_t) - q_{s_t,\diamond}(x_t, a_t)|\mathcal{F}_{t-1}] = 0$. Thus, $v_{s_t,\diamond}(x_t) - q_{s_t,\diamond}(x_t, a_t)$ is a Martingale difference. Since $||w_{s_t,r}|| \leq \sqrt{d}(2 + \mathrm{sp}(v_r^*))$ (from the optimization problem in Algorithm 1), hence, $v_{s_t,r}(x_t) - q_{s_t,r}(x_t, a_t)$ is upper bounded by $\beta_r\sqrt{T\log(T/\delta)}$ with probability $1 - \delta$. Thus, from union bound, with probability $1 - 4\delta$, we have

$$\sum_{t=1}^T (\mathbb{E}_{x'\sim p(\cdot|x_t,a_t)}[v_r^*(x')] - q_r^*(x_t, a_t))$$

$$\leq \sum_{t=1}^T (\mathbb{E}_{x'\sim p(\cdot|x_t,a_t)}[v_{s_t,r}(x')] - v_{s_t,r}(x_t)) + \mathcal{O}(\beta_r\sqrt{dT\log(T/\delta)}) \tag{18}$$

Notice that every time the algorithm updates (i.e., $s_t \neq s_{t-1}$) it holds that $\det(\Lambda_t) = \det(\Lambda_{s_t}) \geq 2\det(\Lambda_{s_{t-1}})$. Since $\det(\Lambda_{T+1})/\det(\Lambda_1) \leq (1 + T/\lambda)^d$, thus, it can not happen more than $d\log_2(1 + T) = \mathcal{O}(d \log T)$ times. Thus, with probability $1 - 4\delta$,

$$\sum_{t=1}^{T} (\mathbb{E}_{x' \sim p(\cdot|x_t, a_t)}[v_r^*(x')] - q_r^*(x_t, a_t))$$

$$\leq \sum_{t=1}^{T} (\mathbb{E}_{x' \sim p(\cdot|x_t, a_t)}[v_{s_{t+1}, r}(x')] - v_{s_{t+1}, r}(x_{t+1})) + \mathcal{O}(\beta_r \sqrt{dT \log(T/\delta)} + \beta_r d \log T) \quad (19)$$

The first term in the right-hand side of (19) can be bounded by $\mathcal{O}(\mathrm{sp}(v_r^*)\sqrt{dT \log(T/\delta)})$ with probability $1 - \delta$ by Azuma-Hoeffding inequality. Hence, with probability $1 - 5\delta$,

$$\sum_{t=1}^{T} (\mathbb{E}_{x' \sim p(\cdot|x_t, a_t)}[v_r^*(x')] - q_r^*(x_t, a_t))$$

$$\leq \mathcal{O}(\beta_r \sqrt{dT \log(T/\delta)} + \beta_r d \log T),$$

where the last step holds via Azuma-Hoeffding inequality.

Hence, the result follows from the definition of $\beta_r$, and (14).

*Proof of Constraint Violation*: From Lemma 6 with probability $1 - \delta$,

$$\sum_{t} q_{s_t, g}(x_t, a_t) - q_g^*(x_t, a_t)$$

$$\leq \mathcal{O}(\beta_g \sqrt{dT \log T}) + \sum_{t=1}^{T} (J_g^* - J_{s_t, g}) + \sum_{t=1}^{T} \mathbb{E}_{x' \sim p(\cdot|x_t, a_t)}[v_{s_t, g}(x') - v_g^*(x')]$$

Similar to (19) we obtain with probability $1 - 3\delta$

$$\sum_{t=1}^{T} (\mathbb{E}_{x' \sim p(\cdot|x_t, a_t)}[v_g^*(x')] - q_g^*(x_t, a_t))$$

$$\leq \mathcal{O}(\beta_g \sqrt{dT \log(T/\delta)} + \beta_g d \log T) + \sum_{t=1}^{T} (J_g^* - J_{s_t, g}) \quad (20)$$

Hence, rearranging (20) and from (14), we obtain

$$\sum_{t=1}^{T} (J_g^* - g(x_t, a_t)) - \sum_{t=1}^{T} (J_g^* - J_{s_t, g}) \leq \mathcal{O}(\beta_g \sqrt{dT \log T} + \beta_g d \log T)$$

$$\sum_{t=1}^{T} (J_{s_t, g} - g(x_t, a_t)) \leq \mathcal{O}(\beta_g \sqrt{dT \log(T/\delta)} + \beta_g d \log T) \quad (21)$$

Note that we can not conclude that $J_{s_t, g} \geq J_g^*$ unlike the reward term. Nevertheless, since $J_{s_t, g}$ is feasible, thus, from the optimization problem $\mathcal{P}_1$ in Algorithm 1, we have $J_{s_t, g} \geq b$. Hence, from (21) with probability $1 - 3\delta$,

$$\sum_{t=1}^{T} (b - g(x_t, a_t)) \leq \sum_{t=1}^{T} (J_{s_t, g} - g(x_t, a_t)) \leq \mathcal{O}(\beta_g \sqrt{dT \log(T/\delta)} + \beta_g d \log(T/\delta))$$

Hence, the result follows. $\qquad\square$

## F.2 PROOF OF LEMMA 2

For $\diamond = r, g$,

$$w_\diamond^* = (\Lambda_t)^{-1} \sum_{\tau=1}^{t-1} \phi(x_\tau, a_\tau) \phi(x_\tau, a_\tau)^T w_\diamond^* + \lambda \Lambda_t^{-1} w_\diamond^*$$

$$= (\Lambda_t)^{-1} \sum_{\tau=1}^{t-1} \phi(x_\tau, a_\tau)(\diamond(x_\tau, a_\tau) - J_\diamond^* + \mathbb{E}_{x' \sim p(\cdot|x_\tau, a_\tau)} v_\diamond^*(x')) + \lambda \Lambda_t^{-1} w_\diamond^*$$

$$= (\Lambda_t)^{-1} \sum_{\tau=1}^{t-1} \phi(x_\tau, a_\tau)(\diamond(x_\tau, a_\tau) - J_\diamond^* + v_\diamond^*(x_{\tau+1})) + \lambda \Lambda_t^{-1} w_\diamond^* + \epsilon_{\diamond,t}^* \tag{22}$$

where

$$\epsilon_{t,\diamond} = (\Lambda_t)^{-1} \sum_{\tau=1}^{t-1} \phi(x_\tau, a_\tau)(\mathbb{E}_{x' \sim p(\cdot|x_\tau, a_\tau)} v_\diamond^*(x') - v_\diamond^*(x_{\tau+1})) \tag{23}$$

Let us denote $\varepsilon_{\tau,\diamond}^* = \mathbb{E}_{x' \sim p(\cdot|x_\tau, a_\tau)} v_\diamond^*(x') - v_\diamond^*(x_{\tau+1})$ which is Martingale difference, then from the self-concentration result with probability $1 - 2\delta$ (Theorem 4),

$$\epsilon_{t,\diamond}^* \le \operatorname{sp}(v_\diamond^*) \sqrt{\log[\frac{det(\Lambda_t)^{1/2} det(\Lambda_0)^{-1/2}}{\delta}]} \tag{24}$$

Now, $det(\Lambda_t) \le (\lambda + t)^d$, $det(\Lambda_0) \ge \lambda^d$. Thus,

$$\epsilon_{t,\diamond}^* \le \operatorname{sp}(v_\diamond^*) \sqrt{d \log((1 + T)/\delta)} \tag{25}$$

On the other hand, note that $\lambda(\Lambda_t)^{-1} w_{\diamond,t}^* \le \beta_\diamond/2$, hence, by selecting $b_{\diamond,t} = \lambda(\Lambda_t)^{-1} w_{\diamond,t}^* + \epsilon_{\diamond,t}^*$ (which is upper bounded by $\beta_\diamond$) and $J_g^* \ge b$ we have the result.

## F.3 PROOF OF LEMMA 6

*Proof.* For notational simplicity, we denote $s_t = s$.

From the definition of $w_{s,\diamond}$

$$w_{s,\diamond} - w_\diamond^* = \Lambda_s^{-1} \sum_{\tau=1}^{s-1} \phi(x_\tau, a_\tau)(\diamond(x_\tau, a_\tau) - J_{s,\diamond} + v_{s,\diamond}(x_{\tau+1})) + b_{s,\diamond}$$

$$- \Lambda_s^{-1} \sum_{\tau=1}^{s-1} \phi(x_\tau, a_\tau)(\diamond(x_\tau, a_\tau) - J_\diamond^* + \mathbb{E}_{x' \sim p(\cdot|x_\tau, a_\tau)} v_\diamond^*(x')) - \lambda \Lambda_s^{-1} w_\diamond^*$$

$$= \Lambda_s^{-1} \sum_{\tau=1}^{s-1} \phi(x_\tau, a_\tau)(J_\diamond^* - J_{s,\diamond} + \mathbb{E}_{x' \sim p(\cdot|x_\tau, a_\tau)}[v_{s,\diamond}(x') - v_\diamond^*(x')])$$

$$+ \epsilon_{s,\diamond} + b_{s,\diamond} - \lambda \Lambda_s^{-1} w_\diamond^*$$

where $\epsilon_{s,\diamond}$ is defined in (23).

$$w_{s,\diamond} - w_\diamond^* = \Lambda_s^{-1} \sum_{\tau=1}^{s-1} \phi(x_\tau, a_\tau) \phi(x_\tau, a_\tau)^T (J_\diamond^* e_1 - J_{s,\diamond} e_1 + \int_\mathcal{X} (v_{s,\diamond}(x') - v_\diamond^*(x')) d\mu(x'))$$

$$+ \epsilon_{s,\diamond} + b_{s,\diamond} - \lambda \Lambda_s^{-1} w_\diamond^*$$

$$= J_\diamond^* e_1 - J_{s,\diamond}^* e_1 + \int_\mathcal{X} (v_{s,\diamond}(x') - v_\diamond^*(x')) d\mu(x') + \epsilon_{s,\diamond} + b_{s,\diamond}$$

$$- \lambda \Lambda_s^{-1}(J_\diamond^* e_1 - J_{s,\diamond} e_1 + \int_\mathcal{X} (v_{s,\diamond}(x') - v_\diamond^*(x')) d\mu(x')) - \lambda \Lambda_s^{-1} w_\diamond^* \tag{26}$$

Therefore,

$$q_{s,\diamond}(x_t, a_t) - q_\diamond^*(x_t, a_t) = \phi(x_t, a_t)^T(w_{s,\diamond} - w_\diamond^*)$$
$$\leq (J_\diamond^* - J_{s,\diamond}^*) + \mathbb{E}_{x'\sim p(\cdot|x_t,a_t)}[v_{s,\diamond}(x') - v_\diamond^*(x')] + \phi(x_t, a_t)^T(\epsilon_{s,\diamond} + b_{s,\diamond} + \lambda\Lambda_s^{-1}u_{s,\diamond}) \quad (27)$$

where

$$u_{s,\diamond} = -(J_\diamond^*e_1 - J_{s,\diamond}e_1 + \int_{\mathcal{X}}(v_{s,\diamond}(x') - v_\diamond^*(x'))d\mu(x')) - w_\diamond^* \quad (28)$$

We now bound the third term in the right hand side which will prove the lemma. From Lemma 21 with probability $1 - \delta$

$$||\epsilon_{s,\diamond}||_{\Lambda_s} = ||\sum_{\tau=1}^{s-1}\phi_\tau\varepsilon_{\diamond,\tau}||_{\Lambda_s^{-1}} \leq 4(2 + \mathrm{sp}(v_\diamond^*))\sqrt{d}\sqrt{d/2\log((s+\lambda)/\lambda) + \log(\mathcal{N}_{\epsilon,\diamond}/\delta)}$$
$$+ 4\sqrt{s^2\epsilon^2/\lambda} \quad (29)$$

We now compute $\mathcal{N}_{\epsilon,\diamond}$ which is the $\epsilon$-covering number for the value function class $v_{s,\diamond}$.

**Lemma 7.** *The $\epsilon$-covering number for the class $\mathcal{V}_\diamond = \{v_\diamond|v_\diamond(x) = \langle\pi(\cdot|x), q_\diamond(x, \cdot)\rangle_{\mathcal{A}}$ is upper bounded by*

$$\log N_{\epsilon,\diamond} \leq d\log\left(1 + \frac{2(2 + \mathrm{sp}(v_\diamond^*))\sqrt{2d}}{\epsilon'}\right) + d\log\left(1 + \frac{16L}{\epsilon'}\right). \quad (30)$$

*where $\epsilon' = \frac{\epsilon}{1 + (2 + \mathrm{sp}(v_\diamond^*))\sqrt{2d}}$, $\pi \in \Pi$, and $q_\diamond(\cdot, \cdot) = w_\diamond^T\phi(\cdot, \cdot)$, where $||w_\diamond|| \leq (2 + \mathrm{sp}(v_\diamond^*))\sqrt{d}$.*

*Proof.* Proof is in Appendix F.4. $\qquad\square$

Hence, from (29) and putting $\epsilon = \frac{1}{T}$, we obtain

$$||\epsilon_{s,\diamond}||_{\Lambda_s} \leq \mathcal{O}(\beta_\diamond) \quad (31)$$

with probability at least $1 - \delta$.

Now,

$$||\lambda\Lambda_s^{-1}u_{s,\diamond}||_{\Lambda_s} = \lambda||u_{s,\diamond}||_{\Lambda_s^{-1}} \leq \sqrt{\lambda}||u_{s,\diamond}|| \leq \mathcal{O}(1 + (2 + \mathrm{sp}(v_\diamond^*))d) = \mathcal{O}(\beta_\diamond). \quad (32)$$

Further, $||b_{s,\diamond}||_{\Lambda_s} \leq \beta_\diamond$.

Hence,

$$\phi(x_t, a_t)^T(\epsilon_{s,\diamond} + b_{s,\diamond} + \lambda\Lambda_s^{-1}u_{s,\diamond}) \leq ||\phi(x_t, a_t)||_{\Lambda_s^{-1}}||\epsilon_{s,\diamond} + b_{s,\diamond} + \lambda\Lambda_s^{-1}u_{s,\diamond}||_{\Lambda_s}$$
$$\leq 2||\phi(x_t, a_t)||_{\Lambda_t^{-1}}||\epsilon_{s,\diamond} + b_{s,\diamond} + \lambda\Lambda_s^{-1}u_{s,\diamond}||_{\Lambda_s}$$

Thus,

$$\sum_{t=1}^{T}(q_{s,\diamond}(x_t, a_t) - q_\diamond^*(x_t, a_t)) \leq \sum_{t=1}^{T}(J^* - J_{s_t,\diamond}) + \sum_{t=1}^{T}\mathbb{E}_{x'\sim p(\cdot|x_t,a_t)}[v_{s_t,\diamond}(x') - v_\diamond^*(x')]$$
$$+ \mathcal{O}(\beta_\diamond\sum_{t=1}^{T}||\phi(x_t, a_t)||_{\Lambda_t^{-1}})$$
$$\leq \sum_{t=1}^{T}(J_\diamond^* - J_{s_t,\diamond}) + \sum_{t=1}^{T}\mathbb{E}_{x'\sim p(\cdot|x_t,a_t)}[v_{s_t,\diamond}(x') - v_\diamond^*(x')]$$
$$+ \mathcal{O}\left(\beta_\diamond\sqrt{T}\sqrt{\sum_{t=1}^{T}||\phi(x_t, a_t)||_{\Lambda_t^{-1}}^2}\right)$$
$$\leq \sum_{t=1}^{T}(J_\diamond^* - J_{s_t,\diamond}) + \sum_{t=1}^{T}\mathbb{E}_{x'\sim p(\cdot|x_t,a_t)}[v_{s_t,\diamond}(x') - v_\diamond^*(x')]$$
$$+ \mathcal{O}(\beta_\diamond\sqrt{dT\log T}) \quad (33)$$

where the second inequality follows from Cauchy-Schwarz inequality. The last inequality follows from the fact that $\sqrt{\sum_{t=1}^{T}||\phi(x_t, a_t)||_{\Lambda_t^{-1}}^2} \leq \sqrt{2\log(\det(\Lambda_{T+1})/(\det(\Lambda_1)))}$, and $\log(\det(\Lambda_{T+1})) \leq d\log(\lambda + T)$ $\qquad\square$

### F.4 PROOF OF LEMMA 7

*Proof.* $||w_{s,\diamond}|| \leq (2 + \mathrm{sp}(v_\diamond^*))\sqrt{d}$. By Lemma 22, the covering number for $w_{s,\diamond}$ is

$$\left(1 + \frac{2(2 + \mathrm{sp}(v_\diamond^*))\sqrt{d}}{\epsilon}\right)^d$$

Since $q_{s,\diamond}(\cdot, \cdot) = \phi(\cdot, \cdot)^T w_{s,\diamond}$ and $||\phi|| \leq \sqrt{2}$, $\epsilon$-covering number for $w_{s,\diamond}$ is also the $\epsilon$-covering number for the class of state-action bias functions $q_{s,\diamond}$ is

$$\left(1 + \frac{2(2 + \mathrm{sp}(v_\diamond^*))\sqrt{2d}}{\epsilon}\right)^d \tag{34}$$

with respect to $\infty$-norm.

For the policy-parameter class, note that $||\zeta|| \leq L$. Hence, the $\epsilon$-covering number for the parameter class is $(1 + 2L/\epsilon)^d$ from Lemma 22. We also use the following result where we show that the 1-norm of the difference between two policies is close if the parameters are close from the property of soft-max.

**Lemma 8.** *If $\pi, \tilde{\pi} \in \Pi$ where $\pi$ is parameterized by $\zeta$ and $\tilde{\pi}$ is parameterized by $\tilde{\zeta}$, then for any $x \in \mathcal{X}$*

$$\sum_a |\pi(a|x) - \tilde{\pi}(a|x)| \leq 8||\zeta - \tilde{\zeta}|| \tag{35}$$

*if $||\zeta - \tilde{\zeta}|| \leq 1/2$.*

*Proof.* See Appendix F.5. $\qquad\square$

The value function class $v_{s,\diamond}$ is parameterized by $(\zeta, w_{s,\diamond})$. Consider $\tilde{\zeta}$ such that $||\tilde{\zeta} - \zeta|| \leq \epsilon/8$, then from Lemma 8 for every state $x$

$$\sum_a |\pi(a|x) - \tilde{\pi}(a|x)| \leq \epsilon \tag{36}$$

Further, consider $\tilde{w}_{s,\diamond}$ such that $||w_{s,\diamond} - \tilde{w}_{s,\diamond}|| \leq \epsilon$, then $\max_{x,a} |q_{s,\diamond}(x,a) - \tilde{q}_{s,\diamond}(x,a)| \leq \epsilon$ where $\tilde{q}_{s,\diamond}$ is parameterized by $\tilde{w}_{s,\diamond}$. Now, $\tilde{v}_{s,\diamond} = \langle \tilde{\pi}, \tilde{q}_{s,\diamond} \rangle$, and $v_{s,\diamond} = \langle \pi, q_{s,\diamond} \rangle$.

Then, for any $x$,

$$\begin{aligned}
\tilde{v}_{s,\diamond}(x) - v_{s,\diamond}(x) &= \langle \tilde{\pi}, \tilde{q}_{s,\diamond} \rangle - \langle \pi, q_{s,\diamond} \rangle \\
&= \langle \pi - \tilde{\pi}, q_{s,\diamond} \rangle - \langle \tilde{\pi}, (\tilde{q}_{s,\diamond} - q_{s,\diamond}) \rangle \\
&\leq \epsilon(2 + \mathrm{sp}(v_\diamond^*))\sqrt{2d} + \epsilon
\end{aligned} \tag{37}$$

Thus, consider $\epsilon' = \dfrac{\epsilon}{1 + (2 + \mathrm{sp}(v_\diamond^*))\sqrt{2d}}$, then,

$$\log N_{\epsilon,\diamond} \leq d\log\left(1 + \frac{2(2 + \mathrm{sp}(v_\diamond^*))\sqrt{2d}}{\epsilon'}\right) + d\log\left(1 + \frac{16L}{\epsilon'}\right). \tag{38}$$

$\qquad\square$

### F.5 PROOF OF LEMMA 8

*Proof.* Dividing $\pi(a|x)$ by $\tilde{\pi}(a|x)$ yields

$$
\begin{aligned}
\frac{\pi(a|x)}{\tilde{\pi}(a|x)} &= \frac{e^{\langle\phi(x,a),\zeta\rangle}}{e^{\langle\phi(x,a),\tilde{\zeta}\rangle}} \times \frac{\sum_{a'} e^{\langle\phi(x,a'),\tilde{\zeta}\rangle}}{\sum_{a'} e^{\langle\phi(x,a'),\zeta\rangle}} \\
&= e^{\langle\phi(x,a),\zeta-\tilde{\zeta}\rangle} \times \sum_{a'} (e^{\langle\phi(x,a'),\tilde{\zeta}-\zeta\rangle}) \frac{e^{\langle\phi(x,a'),\zeta\rangle}}{\sum_{\bar{a}} e^{\langle\phi(x,\bar{a}),\zeta\rangle}} \\
&= e^{\langle\phi(x,a),\zeta-\tilde{\zeta}\rangle} \sum_{a'} \pi(a'|x) e^{\langle\phi(x,a'),\tilde{\zeta}-\zeta\rangle}
\end{aligned}
\tag{39}
$$

Since $||\phi||\leq 1$, and exponential is a strictly increasing function, thus,

$$
\frac{\pi(a|x)}{\tilde{\pi}(a|x)} = e^{||\zeta-\tilde{\zeta}||} \sum_{a'} \pi(a'|x) e^{||\zeta-\tilde{\zeta}||} \leq e^{2||\zeta-\tilde{\zeta}||} \leq 1 + 4||\zeta - \tilde{\zeta}||
\tag{40}
$$

where the first inequality stems from the fact that $\sum_{a'} \pi(a'|x) = 1$. The second inequality stems from the fact that $e^x \leq 1 + 2x$ if $x \in (0, 0.5)$. Hence,

$$
\pi(a|x) - \tilde{\pi}(a|x) \leq 4\tilde{\pi}(a|x)||\zeta - \tilde{\zeta}||
\tag{41}
$$

We can apply the similar argument with the roles $\zeta$ and $\tilde{\zeta}$ reversed to conclude

$$
\tilde{\pi}(a|x) - \pi(a|x) \leq 4\pi(a|x)||\zeta - \tilde{\zeta}||
\tag{42}
$$

Hence,

$$
\begin{aligned}
\sum_a |\pi(a|x) - \tilde{\pi}(a|x)| &\leq 4||\zeta - \tilde{\zeta}|| \sum_a \max\{\pi(a|x), \tilde{\pi}(a|x)\} \\
&\leq 4||\zeta - \tilde{\zeta}|| \sum_a \{\pi(a|x) + \tilde{\pi}(a|x)\} \\
&= 8||\zeta - \tilde{\zeta}||
\end{aligned}
\tag{43}
$$

$\square$

### F.6 CLASS OF POLICIES

Note that we only use Lemma 8 to show Lemma 7 for the class of bias value function. Thus, our analysis can be extended to any policy parameterized by $\zeta$ such that

**Definition 3.** *For any $x$, $\sum_a |\pi(a|x) - \tilde{\pi}(a|x)| \leq R||\zeta - \tilde{\zeta}||$ holds for some $||\zeta - \tilde{\zeta}|| \leq L_1$ where $R$ is constant and $L_1 > 0$.*

Obviously, soft-max policy is one such policy which satisfies the condition of the Definition 3.

## G PROOF OF THE RESULTS OF SECTION 3.2

**Organization**: In Appendix G.1, we provide the proof sketch and subsequently we prove various parts in Appendix E.2 to Appendix E.9. We formally prove Theorem 2 in Appendix G.10.

**Notations for this Section**: Throughout this section, we denote $Q_{r,h}^k, Q_{g,h}^k, w_{r,h}^k, w_{g,h}^k, \Lambda_h^k$ as the $Q$-value and the parameter values estimated at the episode $k$. $V_{j,h}^k(\cdot) = \langle\pi_{h,k}(\cdot|\cdot), Q_{j,h}^k(\cdot,\cdot)\rangle_{\mathcal{A}}$. $\pi_{h,k}(\cdot|x)$ is the soft-max policy based on the composite $Q$-function at the $k$-th episode as $Q_{r,h}^k + Y_k Q_{g,h}^k$. To simplify the presentation, we denote $\phi_h^k = \phi(x_h^k, a_h^k)$.

Without loss of generality, we assume $||\phi(x,a)||_2 \leq 1$ for all $(x,a) \in \mathcal{S} \times \mathcal{A}$, $||\mu(\mathcal{S})||_2 \leq \sqrt{d}$, $||\theta_j||_2 \leq \sqrt{d}$ for $\diamond = r, g$.

### G.1 PROOF SKETCH OF THEOREM 2

We reiterate the decomposition of regret.

$$\text{Regret}(T) = \sum_{k=1}^{T/H}(HJ_r^* - V_{r,1}^{\pi^*}(x_1^k)) + \sum_{k=1}^{T/H}(V_{r,1}^{\pi^*}(x_1^k) - V_{r,1}^{\pi_k}(x_1^k))$$
$$+ \sum_{k=1}^{T/H}(V_{r,1}^{\pi_k}(x_1^k) - \sum_{t=(k-1)H+1}^{kH} r(x_t, a_t)) \tag{44}$$

The first term is bounded via Lemma 3 by $(T/H)\text{sp}(v_r^*)$ which we prove in Appendix G.3. The third-term is bounded using Azuma-Hoeffding inequality, and its upper bound is given by $\mathcal{O}(\sqrt{TH\iota})$ with probability $1 - \delta$. We bound the second term in Appendices G.4–G.8. In the following, we describe a rough idea to bound the second term.

Note that in order to bound the second term we follow Ghosh et al. (2022) which bounds the regret over an episodic setup. **However, there are subtle differences.** Ghosh et al. (2022) bounds the regret with respect to the optimal solution for the episodic case, i.e., Ghosh et al. (2022) bounds the following

$$\sum_{k=1}^{K}(V_{r,1}^{\bar{\pi}^*}(x_1^k) - V_{r,1}^{\pi_k}(x_1^k))$$

where $\bar{\pi}^*$ is the optimal solution for the episodic case (cf.(46)). Instead, we are interested in bounding

$$\sum_{k}(V_{r,1}^{\pi^*}(x_1^k) - V_{r,1}^{\pi_k}(x_1^k))$$

*Note that the optimal solution for episodic case, $\bar{\pi}^*$, can be different from $\pi^*$, the optimal solution for the infinite horizon average reward CMDP.* Since we relax the constraint by subtracting $\kappa$ from $Hb$, it would guarantee that $\pi^*$ is also feasible over the episodic case for any initial state $x_1^k$ by Lemma 3. Hence,

$$V^{\bar{\pi}^*}(x_1^k) - V^{\pi^*}(x_1^k) \geq 0$$

Thus,

$$\sum_{k}(V_{r,1}^{\pi^*}(x_1^k) - V_{r,1}^{\pi_k}(x_1^k)) \leq \sum_{k}(V_{r,1}^{\bar{\pi}^*}(x_1^k) - V_{r,1}^{\pi_k}(x_1^k)) \tag{45}$$

Now, we invoke the analysis of Ghosh et al. (2022) to bound the right-hand side of (45).

Since we relax the constraint there will be an additional term $\kappa$ in the violation at each episode. By summing over all the episodes the term will grow as $(T/H)\kappa$. Since $H = \mathcal{O}(T^{1/4})$, we can bound that additional term by $\mathcal{O}(T^{3/4})$.

Now, we bound the right-hand side of (45). We first state the episodic CMDP on which the Algorithm is learning for completeness.

$$\text{maximize }_\pi V_{r,1}^\pi(x_1^k) \quad \text{subject to } V_{g,1}^\pi(x_1^k) \geq Hb - \kappa \tag{46}$$

Similar to Ghosh et al. (2022), we first establish the following decomposition, which upper bounds the sum of regret and violation (Lemma 9).

**Lemma 9.** *For any $Y \in [0, \xi]$, we have*

$$\sum_{k=1}^{K}(V_{r,1}^{\bar{\pi}^*}(x_1^k) - V_{r,1}^{\pi_k}(x_1^k)) + Y(Hb - \kappa - V_{g,1}^{\pi_k}(x_1^k)) \leq \frac{1}{2\eta}Y^2 + \frac{\eta H^2 K}{2} +$$

$$\underbrace{\sum_{k=1}^{K}\left(V_{r,1}^{\bar{\pi}^*}(x_1^k) + Y_k V_{g,1}^{\bar{\pi}^*}(x_1^k)\right) - \left(V_{r,1}^k(x_1^k) + Y_k V_{g,1}^k(x_1^k)\right) +}_{\mathcal{T}_1}$$

$$\underbrace{\sum_{k=1}^{K}\left(V_{r,1}^k(x_1^k) - V_{r,1}^{\pi_k}(x_1^k)\right) + Y\sum_{k=1}^{K}\left(V_{g,1}^k(x_1^k) - V_{g,1}^{\pi_k}(x_1^k)\right)}_{\mathcal{T}_2} \tag{47}$$

*Proof.* See Appendix G.4. ☐

This will serve as the basis when applying optimization tools to bound the violation as well. Note that by making $Y = 0$, we can bound the regret.

Note that $\mathcal{T}_1$ is similar to the term related to optimism in the unconstrained case with the difference being that we now have two value functions weighted by the dual variable $Y_k$. Similarly, $\mathcal{T}_2$ is similar to prediction error term with the additional weight by $Y$. Since the first term in the above inequality can be easily bounded with a proper choice of $\eta$, we are only left to bound $\mathcal{T}_1$ and $\mathcal{T}_2$, respectively.

In order to bound $\mathcal{T}_1$ and $\mathcal{T}_2$, we use the following result

**Lemma 10.** *There exists a constant $C_2$ such that for any fixed $p \in (0, 1)$, if we let $\mathcal{E}$ be the event that*

$$\left\| \sum_{\tau=1}^{k-1} \sum_{h'=1}^{H} \phi_{j,h'}^{\tau} [V_{j,h+1}^k(x_{h'+1}^\tau) - \mathbb{P}V_{j,h+1}^k(x_{h'}^\tau, a_{h'}^\tau)] \right\|_{(\Lambda^k)^{-1}} \le C_2 dH \sqrt{\chi} \tag{48}$$

*for all $j \in \{r, g\}$, $\chi = \log[2(C_1+1)\xi \log(|\mathcal{A}|)dT/\delta]$, for some constant $C_2$, then $\Pr(\mathcal{E}) = 1 - 2\delta$.*

*Proof.* Please see the proof of Lemma 8 in Ghosh et al. (2022). ☐

The above lemma entails that for all $(k, h) \in [K] \times [H]$ with high probability

$$\left\| \sum_{\tau=1}^{k-1} \sum_{h} \phi(x_h^\tau, a_h^\tau) \left[ V_{\diamond,h+1}^k(x_{h+1}^\tau) - \mathbb{P}V_{\diamond,h+1}^k(x_h^\tau, a_h^\tau) \right] \right\|_{(\Lambda^k)^{-1}}$$

is upper bounded by lower order term (e.g., $\mathcal{O}(d\sqrt{\log T})$). Similar to Algorithm 1 the general idea here is to fix a function class $\mathcal{V}_{\diamond,h}$ in advance and then show that each possible value function in our algorithm $V_{\diamond,h}^k$ is within this class which has log-covering number which scales with $\log T$.

Similar to Ghosh et al. (2022) we introduce soft-max policy and define the following corresponding function classes to show that the value function class has log $\epsilon$-covering number which scales with $\log T$. We first define the following class for $Q$-function for $\diamond = r, g$. $\mathcal{Q}_{\diamond} = \{Q_{\diamond}|Q_{\diamond}(\cdot, \cdot) = \min\{\langle w_{\diamond}, \phi(\cdot, \cdot) \rangle + \beta\sqrt{\phi(\cdot,\cdot)^T(\Lambda_h)^{-1}\phi(\cdot,\cdot)}, H\}$. Then, we define the following value function class $\mathcal{V}_{\diamond}$. $\mathcal{V}_{\diamond} = \{V_{\diamond}|V_{\diamond}(\cdot) = \sum_a \pi(a|\cdot)Q_{\diamond}(\cdot, a); Q_{\diamond} \in \mathcal{Q}_{\diamond}, \pi \in \Pi_e\}$, where $\Pi_e$ is given by the following class $\Pi_e = \{\pi|\pi(a|\cdot) = \text{SOFT-MAX}_\alpha^a((Q_r(\cdot, \cdot) + YQ_g(\cdot, \cdot)); \forall a \in \mathcal{A}, Q_r \in \mathcal{Q}_r, Q_g \in \mathcal{Q}_g, Y \in [0, \xi]\}$. As described in Ghosh et al. (2022) greedy algorithm with respect to the composite $Q$-function fails to show that the log $\epsilon$-covering number for individual $\mathcal{V}_{\diamond}$ scales at most $\mathcal{O}(\log(T))$ as the greedy policy is not Lipschitz.

Using the above we show the following

**Lemma 11.** *There exists an absolute constant $\beta = C_1 dH\sqrt{\iota}$, $\iota = \log(\log(|\mathcal{A}|)\xi 2dT/\delta)$, and for any fixed policy $\pi$, on the event $\mathcal{E}$ defined in Lemma 10, we have*

$$\langle \phi(x, a), w_{\diamond,h}^k \rangle - Q_{\diamond,h}^\pi(x, a) = \mathbb{P}(V_{\diamond,h+1}^k - V_{\diamond,h+1}^\pi)(x, a) + \Delta^k(x, a) \tag{49}$$

*for some $\Delta^k(x, a)$ that satisfies $|\Delta^k(x, a)| \le \beta\sqrt{\phi(x, a)^T(\Lambda^k)^{-1}\phi(x, a)}$.*

*Proof.* See Appendix G.5. ☐

The above lemma bounds the difference between the value function maintained in Algorithm 2 (without the bonus term) and the value function for any policy for both the reward and utility value functions. We bound this using the expected difference at the next step plus an error term. The result shows that this error term can be upper bounded by the bonus term with a high-probability.

Using the above, we can easily show the following–

**Lemma 12.** *With prob. $1 - 2\delta$, (for the event in $\mathcal{E}$)*

$$Q_{r,h}^\pi(x, a) + Y_k Q_{g,h}^\pi(x, a) \le Q_{r,h}^k(x, a) + Y_k Q_{g,h}^k(x, a) + \mathbb{P}(V_{h+1}^{\pi,Y_k} - V_{h+1}^k)(x, a) \tag{50}$$

*Proof.* See Appendix G.6. □

Using the above, and the property of the soft-max (Lemma 19), we obtain

**Lemma 13.** *With probability* $1 - 2\delta$, $\mathcal{T}_1 \leq \dfrac{H \log(|\mathcal{A}|)}{\alpha}$

*Proof.* See Appendix G.7. □

Further, from Lemma 12, we obtain the following

**Lemma 14.** *On the event defined in* $\mathcal{E}$ *in Lemma 10, we have*

$$V_{\diamond,1}^k(x_1) - V_{\diamond,1}^{\pi_k}(x_1) \leq \sum_{h=1}^{H}(D_{\diamond,h,1}^k + D_{\diamond,h,2}^k) + \sum_{h=1}^{H} 2\beta\sqrt{\phi(x_h^k, a_h^k)^T(\Lambda_h^k)^{-1}\phi(x_h^k, a_h^k)} \quad (51)$$

where

$$D_{\diamond,h,1}^k = \langle (Q_{\diamond,h}^k(x_h^k, \cdot) - Q_{\diamond,h}^{\pi_k}(x_h^k, \cdot)), \pi_{h,k}(\cdot|x_h^k) \rangle - (Q_{\diamond,h}^k(x_h^k, a_h^k) - Q_{\diamond,h}^{\pi_k}(x_h^k, a_h^k))$$
$$D_{\diamond,h,2}^k = \mathbb{P}_h(V_{\diamond,h+1}^k - V_{\diamond,h+1}^{\pi_k})(x_h^k, a_h^k) - [V_{\diamond,h+1}^k - V_{\diamond,h+1}^{\pi_k}](x_{h+1}^k) \quad (52)$$

*Proof.* See Appendix G.8. □

Using the above result, Azuma-Hoeffding inequality, and plugging the value of $\beta$ we obtain the following

**Lemma 15.** *With probability* $1 - 4\delta$, $\mathcal{T}_2 \leq (Y+1)\mathcal{O}(\sqrt{d^3TH^2\iota^2})$

*Proof.* See Appendix G.9. □

**Combining all the pieces:** In (47), we replace $Y$ with 0, and $\eta = \xi/\sqrt{KH^2}$. Then, by combining (45),Lemmas 13 and 15 we bound the second term in (44) with probability $1 - 4\delta$. Thus, the final result follows from the union bound ( Appendix G.10).

**Constraint Violation**: The violation term can be decomposed as the following

$$\sum_{k=1}^{K}(Hb - \kappa - V_{g,1}^{\pi_k}(x_1^k)) + \sum_{k=1}^{K}(V_{g,1}^{\pi_k}(x_1^k) - \sum_{t=(k-1)H+1}^{kH} g(x_t, a_t)) + \sum_{k=1}^{K}\kappa \quad (53)$$

where $K = T/H$. Note that the third term is equivalent to $(T/H)\kappa$. The second term can be bounded by Azuma Hoeffding inequality $\tilde{\mathcal{O}}(\sqrt{TH\iota})$ with probability $1 - \delta$. We now describe how to bound the first term.

Note from Lemma 9

$$\sum_{k=1}^{K}(V_{r,1}^{\bar{\pi}^*}(x_1^k) - V_{r,1}^{\pi_k}(x_1^k)) + Y(Hb - \kappa - V_{g,1}^{\pi_k}(x_1^k)) \leq \frac{1}{2\eta}Y^2 + \frac{\eta H^2 K}{2} +$$

$$\underbrace{\sum_{k=1}^{K}\left(V_{r,1}^{\bar{\pi}^*}(x_1^k) + Y_k V_{g,1}^{\bar{\pi}^*}(x_1^k)\right) - \left(V_{r,1}^k(x_1^k) + Y_k V_{g,1}^k(x_1^k)\right) +}_{\mathcal{T}_1}$$

$$\underbrace{\sum_{k=1}^{K}\left(V_{r,1}^k(x_1^k) - V_{r,1}^{\pi_k}(x_1^k)\right) + Y\sum_{k=1}^{K}\left(V_{g,1}^k(x_1^k) - V_{g,1}^{\pi_k}(x_1^k)\right)}_{\mathcal{T}_2} \quad (54)$$

Now, using $\eta = \dfrac{\xi}{\sqrt{KH^2}}, Y \le \xi$, Lemma 13 and 15 we obtain with probability $1 - 4\delta$

$$\sum_{k=1}^{K}(V_{r,1}^{\pi^*}(x_1^k) - V_{r,1}^{\pi_k}(x_1^k)) + Y(Hb - \kappa - V_{g,1}^{\pi_k}(x_1^k)) \le \mathcal{O}(\sqrt{d^3\xi^2 T\iota^2}) \qquad (55)$$

Now, from the result of convex optimization (Corollary 1) we obtain that with probability $1 - 3\delta$,

$$\sum_{k=1}^{K}(Hb - \kappa - V_{g,1}^{\pi_k}(x_1^k)) \le \dfrac{2(1+\xi)}{\xi}\mathcal{O}(\sqrt{d^3 T\iota^2})$$

Thus, we obtain bound for the second term. The final result is obtained from the bounds of each individual term in (53).

### G.2 PRELIMINARY RESULTS

**Lemma 16.** *Under Assumption 2, for any fixed policy $\pi$, let $w_h^\pi$ be the corresponding weights such that $Q_{j,h}^\pi = \langle \phi(x,a), w_{j,h}^\pi \rangle$, for $j \in \{r,g\}$, then we have for all $h \in [H]$,*

$$||w_{j,h}^\pi|| \le 2H\sqrt{d} \qquad (56)$$

*Proof.* From the linearity of the action-value function, we have

$$\begin{aligned}Q_{j,h}^\pi(x,a) &= j_h(x,a) + \mathbb{P}V_{j,h}^\pi(x,a) \\ &= \langle \phi(x,a), \theta_{j,h}\rangle + \int_{\mathcal{S}} V_{j,h+1}^\pi(x')\langle \phi(x,a), d\mu(x')\rangle \\ &= \langle \phi(x,a), w_{j,h}^\pi \rangle \qquad (57)\end{aligned}$$

where $w_{j,h}^\pi = \theta_{j,h} + \int_{\mathcal{S}} V_{j,h+1}^\pi(x')d\mu(x')$.

Now, $||\theta_{j,h}|| \le \sqrt{d}$, and $||\int_{\mathcal{S}} V_{j,h+1}^\pi(x')d\mu_h(x')|| \le H\sqrt{d}$. Thus, the result follows. $\qquad \square$

**Lemma 17.** *For any $(k,h)$, the weight $w_{j,h}^k$ satisfies*

$$||w_{j,h}^k|| \le 2H\sqrt{dkH/\lambda} \qquad (58)$$

*Proof.* For any vector $v \in \mathcal{R}^d$ we have

$$|v^T w_{j,h}^k| = |v^T(\Lambda)_k^{-1}\sum_{\tau=1}^{k-1}\sum_{h'=1}^{H}\phi_h^\tau(x_{h'}^\tau, a_{h'}^\tau)(j(x_{h'}^\tau, a_{h'}^\tau) + \sum_a \pi_{h+1,k}(a|x_{h'+1}^\tau)Q_{j,h+1}^k(x_{h'+1}^\tau, a))| \qquad (59)$$

here $\pi_{h,k}(\cdot|x)$ is the Soft-max policy.

Note that $Q_{j,h+1}^k(x,a) \le H$ for any $(x,a)$. Hence, from (59) we have

$$\begin{aligned}|v^T w_{j,h}^k| &\le \sum_{\tau=1}^{k-1}\sum_{h'}|v^T(\Lambda^k)^{-1}\phi_{h'}^\tau|.2H \\ &\le \sqrt{\sum_{\tau=1}^{k-1}\sum_{h'}v^T(\Lambda^k)^{-1}v}\sqrt{\sum_{\tau=1}^{k-1}\sum_{h'}\phi_{h'}^\tau(\Lambda^k)^{-1}\phi_{h'}^\tau}.2H \\ &\le 2H||v||\dfrac{\sqrt{dkH}}{\sqrt{\lambda}} \qquad (60)\end{aligned}$$

We use Observation 1 to bound the second term. Note that $||w_{j,h}^k|| = \max_{v:||v||=1}|v^T w_{j,h}^k|$. Hence, the result follows. $\qquad \square$

### G.3 PROOF OF LEMMA 3

*Proof.* We prove the result for $\diamond = r$. The proof for $\diamond = g$ is similar.

For any $x$, and $h \in [H]$,

$$V_{r,h}^{\pi}(x) - (H + h - 1)J_r^{\pi} = \mathbb{E}[\sum_{h'=h}^{H} r(x_h', a_h') - J_r^{\pi}|\pi, x_h = x]$$

$$= \mathbb{E}[\sum_{h'=h}^{H} q_r^{\pi}(x_h', a_h') - \mathbb{P}v_r^{\pi}(x_h', a_h')|\pi, x_h = x]$$

$$= \mathbb{E}[\sum_{h'=h}^{H} (v_r^{\pi}(x_{h'}) - v_r^{\pi}(x_{h'+1})|\pi, x_h = x]$$

$$= v_r^{\pi}(x) - \mathbb{E}[v_r^{\pi}(x_{H+1})]$$

Hence, $|V_{r,h}^{\pi}(x) - (H + h - 1)J_r^{\pi}| \le \mathrm{sp}(v_r^{\pi})$ since $|v_r^{\pi}| \le \frac{1}{2}\mathrm{sp}(v_r^{*})$. $\qquad\square$

### G.4 PROOF OF LEMMA 9

We first state and prove the following result which is similar to the one proved in Ding et al. (2021).

**Lemma 18.** *For $Y \in [0, \xi]$,*

$$\sum_{k=1}^{K}(Y - Y_k)(Hb - \kappa - V_{g,1}^k(x_1^k)) \le \frac{Y^2}{2\eta} + \frac{\eta H^2 K}{2} \tag{61}$$

*Proof.*

$$|Y_{k+1} - Y|^2 = |Proj_{[0,\xi]}(Y_k + \eta(b - V_{g,1}^k(x_1^k))) - Proj_{[0,\xi]}(Y)|^2$$

$$\le (Y_k + \eta(b - V_{g,1}^k(x_1)) - Y)^2$$

$$\le (Y_k - Y)^2 + \eta^2 H^2 + 2\eta(Y_k - Y)(b - V_{g,1}^k(x_1^k)) \tag{62}$$

Summing over $k$, we obtain

$$0 \le |Y_{K+1} - Y|^2 = |Y_1 - Y|^2 + 2\eta\sum_{k=1}^{K}(Hb - \kappa - V_{g,1}^k(x_1^k))(Y_k - Y) + \eta^2 H^2 K$$

$$\sum_{k=1}^{K}(Y - Y_k)(b - V_{g,1}^k(x_1^k)) \le \frac{|Y_1 - Y|^2}{2\eta} + \frac{\eta H^2 K}{2} \tag{63}$$

Since $Y_1 = 0$, we have the result. $\qquad\square$

Now, we prove Lemma 9.

*Proof.* Note that

$$Y\sum_{k=1}^{K}(Hb - \kappa - V_{g,1}^{\pi_k}(x_1^k)) = \sum_{k=1}^{K}(Y - Y_k)(Hb - \kappa - V_{g,1}^k(x_1^k)) + Y_k(Hb - \kappa) - Y_k V_{g,1}^k(x_1^k)$$

$$+ Y(V_{g,1}^k(x_1^k) - V_{g,1}^{\pi_k}(x_1^k))$$

$$\le \frac{1}{2\eta}Y^2 + \frac{\eta}{2}H^2 K + \sum_{k=1}^{K}(Y_k(Hb - \kappa) + (Y - Y_k)V_{g,1}^k(x_1^k) - YV_{g,1}^{\pi_k}(x_1^k))$$

$$\le \frac{1}{2\eta}Y^2 + \frac{\eta}{2}H^2 K + \sum_{k=1}^{K}(Y_k V_{g,1}^{\pi^*}(x_1^k) - Y_k V_{g,1}^k(x_1^k) + YV_{g,1}^k(x_1^k) - YV_{g,1}^{\pi_k}(x_1^k))$$

where the first inequality follows from Lemma 18, and the second inequality follows from the fact that $V_{g,1}^{\pi^*}(x_1^k) \ge Hb - \kappa$. Hence, the result simply follows from the above inequality. $\qquad\square$

### G.5 PROOF OF LEMMA 11

*Proof.* We only prove for $\diamond = r$, the proof for $\diamond = g$ is similar.

Note that $Q_{r,h}^\pi(x,a) = \langle \phi(x,a), w_{r,h}^\pi \rangle = r_h(x,a) + \mathbb{P}V_{r,h+1}^\pi(x,a)$.

We can write $w_{r,h}^\pi = (\Lambda^k)^{-1}\Lambda^k w_{r,h}^\pi$.

Now, $\Lambda^k = \lambda\mathbf{I} + \sum_{\tau=1}^{k-1}\sum_{h'}\phi(x_{h'}^\tau, a_{h'}^\tau)\phi(x_{h'}^\tau, a_{h'}^\tau)^T$ (See the algorithm 1, line 3). Hence,

$w_{r,h}^\pi = (\Lambda^k)^{-1}(\lambda\mathbf{I} + \sum_{\tau=1}^{k-1}\sum_{h'}\phi(x_{h'}^\tau, a_{h'}^\tau)\phi(x_{h'}^\tau, a_{h'}^\tau)^T)w_{r,h}^\pi$

Finally, from the definition of $w_{r,h}^\pi$, $\phi(x_h^\tau, a_h^\tau)^T w_{r,h}^\pi = r_h(x_h^\tau, a_h^\tau) + \mathbb{P}V_{r,h+1}(x_h^\tau, a_h^\tau)$ (from Lemma 4).

Hence, $w_{r,h}^\pi = (\Lambda^k)^{-1}\lambda\mathbf{I}w_{r,h}^\pi + (\Lambda^k)^{-1}\sum_{\tau=1}^{k-1}\sum_{h'}\phi(x_{h'}^\tau, a_{h'}^\tau)(r_h(x_{h'}^\tau, a_{h'}^\tau) + \mathbb{P}V_{r,h+1}^\pi(x_{h'}^\tau, a_{h'}^\tau))$

Hence, we have

$$
\begin{aligned}
w_{r,h}^k - w_{r,h}^\pi &= (\Lambda^k)^{-1}\sum_{\tau=1}^{k-1}\sum_{h'=1}^H \phi_{h'}^\tau[r(x_{h'}^\tau, a_{h'}^\tau) + V_{r,h+1}^k(x_{h'+1}^\tau)] - w_{r,h}^\pi \\
&= -\lambda(\Lambda^k)^{-1}(w_{r,h}^\pi) + (\Lambda^k)^{-1}\sum_{\tau=1}^{k-1}\sum_{h'}\phi_{h'}^\tau[V_{r,h+1}^k(x_{h'+1}^\tau) - \mathbb{P}V_{r,h+1}^k(x_{h'}^\tau, a_{h'}^\tau)] \\
&\quad + (\Lambda^k)^{-1}\sum_{\tau=1}^{k-1}\sum_{h'}\phi_{h'}^\tau[\mathbb{P}V_{r,h+1}^k(x_{h'}^\tau, a_{h'}^\tau) - \mathbb{P}V_{r,h+1}^\pi(x_{h'}^\tau, a_{h'}^\tau)]
\end{aligned}
\tag{64}
$$

Now, we bound each term in the right hand side of expression in (64). We call those terms as $\mathbf{q}_1$, $\mathbf{q}_2$, and $\mathbf{q}_3$ respectively.

First, note that

$$
\begin{aligned}
|\langle\phi(x,a), \mathbf{q}_1\rangle| &= |\lambda\langle\phi(x,a), (\Lambda^k)^{-1}(w_{r,h}^\pi)\rangle| \\
&\le \sqrt{\lambda}\|w_{r,h}^\pi\|\sqrt{\phi(x,a)^T(\Lambda^k)^{-1}\phi(x,a)}
\end{aligned}
\tag{65}
$$

Second, from Lemma 10, for the event in $\mathcal{E}$, we have

$$
|\langle\phi(x,a), \mathbf{q}_2\rangle| \le CdH\sqrt{\chi}\sqrt{\phi(x,a)^T(\Lambda^k)^{-1}\phi(x,a)}
\tag{66}
$$

where $\chi = \log(2(C_1+1)\log(|\mathcal{A}|)\xi dT/p)$. Third,

$$
\langle\phi(x,a), \mathbf{q}_3\rangle = \langle\phi(x,a), (\Lambda^k)^{-1}\sum_{\tau=1}^{k-1}\sum_{h'}\phi_{h'}^\tau[\mathbb{P}(V_{r,h+1}^k - V_{r,h+1}^\pi)(x_{h'}^\tau, a_{h'}^\tau)]\rangle
$$

$$
= \langle\phi(x,a), (\Lambda^k)^{-1}\sum_{\tau=1}^{k-1}\sum_{h'}\phi_{h'}^\tau(\phi_{h'}^\tau)^T\int(V_{r,h+1}^k - V_{r,h+1}^\pi)(x')d\mu(x')\rangle
$$

$$
= \langle\phi(x,a), \int(V_{r,h+1}^k - V_{r,h+1}^\pi)(x')d\mu(x')\rangle - \langle\phi(x,a), \lambda(\Lambda^k)^{-1}\int(V_{r,h+1}^k - V_{r,h+1}^\pi)(x')d\mu(x')\rangle
\tag{67}
$$

The last term in (67) can be bounded as the following

$$
|\langle\phi(x,a), \lambda(\Lambda^k)^{-1}\int(V_{r,h+1}^k - V_{r,h+1}^\pi)(x')d\mu(x')\rangle| \le 2H\sqrt{d\lambda}\sqrt{\phi(x,a)^T(\Lambda^k)^{-1}\phi(x,a)}
\tag{68}
$$

since $\|\int(V_{r,h+1}^k - V_{r,h+1}^\pi)(x')d\mu(x')\|_2 \le 2H\sqrt{d}$ as $\|\mu(\mathcal{S})\| \le \sqrt{d}$. The first term in (67) is equal to

$$
\mathbb{P}(V_{r,h+1}^k - V_{r,h+1}^\pi)(x,a)
\tag{69}
$$

Note that $\langle \phi(x,a), w_{r,h}^k \rangle - Q_{r,h}^\pi(x,a) = \langle \phi(x,a), w_{r,h}^k - w_{r,h}^\pi \rangle = \langle \phi(x,a), \mathbf{q_1} + \mathbf{q_2} + \mathbf{q_3} \rangle$. Since $\lambda = 1$, we have from (65), (66,(68), and (69)

$$|\langle \phi(x,a), w_{r,h}^k \rangle - Q_{r,h}^\pi(x,a) - \mathbb{P}(V_{r,h+1}^k - V_{r,h+1}^\pi)(x,a)| \leq C_3 dH \sqrt{\chi} \sqrt{\phi(x,a)^T (\Lambda^k)^{-1} \phi(x,a)} \tag{70}$$

for some constant $C_3$ which is independent of $C_1$. Finally, note that

$$\begin{aligned} C_3 \sqrt{\chi} &= \sqrt{\log(2(C_1+1)\log(|\mathcal{A}|)\xi dT/p)} \\ &= C_3 \sqrt{\iota + \log(C_1+1)} \\ &\leq C_1 \sqrt{\iota} \end{aligned} \tag{71}$$

where $\iota = \log(2\log(|\mathcal{A}|)\xi dT/p)$. The last inequality follows from the fact that $\iota \in [\log 2, \infty)$ as $|A| \geq 2$, and $C_3$ is independent of $C_1$. Hence, we can always pick $C_3 \sqrt{\log 2 + \log(C_1+1)} \leq C_1 \sqrt{\log 4}$ which satisfies (71) for all values of $\iota \in [\log 2, \infty)$. $\square$

## G.6 PROOF OF LEMMA 12

*Proof.* From Lemma 11, we have w.p. $1 - 2\delta$,

$$\begin{aligned} Q_{r,h}^\pi(x,a) + Y_k Q_{g,h}^\pi(x,a) &\leq \langle \phi(x,a), w_{r,h}^k \rangle + Y_k \langle \phi(x,a), w_{g,h}^k \rangle + (1+Y_k)\beta \sqrt{\phi(x,a)^T(\Lambda^k)^{-1}\phi(x,a)} \\ &\quad + \mathbb{P}(V_{r,h+1}^\pi + Y_k V_{g,h+1}^\pi - V_{r,h+1}^k - Y_k V_{g,h+1}^k)(x,a) \\ &= Q_{r,h}^k(x,a) + Y_k Q_{g,h}^k(x,a) + \mathbb{P}(V_{h+1}^{\pi,Y_k} - V_{h+1}^k)(x,a) \end{aligned}$$

$\square$

## G.7 PROOF OF LEMMA 13

First, we state and prove a supporting result which bounds the value functions corresponding to the greedy policy and the soft-max policy at a given step. We show that this gap can be controlled by the parameter $\alpha$.

**Lemma 19.** $\bar{V}_h^k(x) - V_h^k(x) \leq \dfrac{\log|\mathcal{A}|}{\alpha}$

where

**Definition 4.** $\bar{V}_h^k(\cdot) = \max_a [Q_{r,h}^k(\cdot,a) + Y_k Q_{g,h}^k(\cdot,a)]$.

$\bar{V}_h^k(\cdot)$ is the value function corresponds to the greedy-policy with respect to the composite $Q$-function.

*Proof.* Note that

$$V_h^k(x) = \sum_a \pi_{h,k}(a|x)[Q_{r,h}^k(x,a) + Y_k Q_{g,h}^k(x,a)] \tag{72}$$

where

$$\pi_{h,k}(a|x) = \frac{\exp(\alpha[Q_{r,h}^k(x,a) + Y_k Q_{g,h}^k(x,a)])}{\sum_a \exp(\alpha[Q_{r,h}^k(x,a) + Y_k Q_{g,h}^k(x,a)])} \tag{73}$$

Denote $a_x = \arg\max_a [Q_{r,h}^k(x,a) + Y_k Q_{g,h}^k(x,a)]$

Now, recall from Definition 4 that $\bar{V}_h^k(x) = [Q_{r,h}^k(x, a_x) + Y_k Q_{g,h}^k(x, a_x)]$. Then,

$$
\begin{aligned}
\bar{V}_h^k(x) - V_h^k(x) &= [Q_{r,h}^k(x, a_x) + Y_k Q_{g,h}^k(x, a_x)] \\
&\quad - \sum_a \pi_{h,k}(a|x)[Q_{r,h}^k(x, a) + Y_k Q_{g,h}^k(x, a)] \\
&\leq \left( \frac{\log(\sum_a \exp(\alpha(Q_{r,h}^k(x, a) + Y_k Q_{g,h}^k(x, a))))}{\alpha} \right) \\
&\quad - \sum_a \pi_{h,k}(a|x)[Q_{r,h}^k(x, a) + Y_k Q_{g,h}^k(x, a)] \\
&\leq \frac{\log(|\mathcal{A}|)}{\alpha}
\end{aligned}
\tag{74}
$$

where the last inequality follows from Proposition 1 in Pan et al. (2019). $\qquad\square$

We are now ready to show Lemma 13.

*Proof.* We prove the lemma by Induction.

First, we prove for the step $H$.

Note that $Q_{\diamond, H+1}^k = 0 = Q_{\diamond, H+1}^\pi$.

Under the event in $\mathcal{E}$ as described in Lemma 10 and from Lemma 11, we have for $j = r, g$,

$$
|\langle \phi(x, a), w_{\diamond, H}^k(x, a) \rangle - Q_{\diamond, H}^\pi(x, a)| \leq \beta \sqrt{\phi(x, a)^T (\Lambda^k)^{-1} \phi(x, a)}
$$

Hence, for any $(x, a)$,

$$
\begin{aligned}
Q_{\diamond, H}^\pi(x, a) &\leq \min\{\langle \phi(x, a), w_{\diamond, H}^k \rangle + \beta \sqrt{\phi(x, a)^T (\Lambda^k)^{-1} \phi(x, a)}, H\} \\
&= Q_{\diamond, H}^k(x, a)
\end{aligned}
\tag{75}
$$

Hence, from the definition of $\bar{V}_h^k$,

$$
\begin{aligned}
\bar{V}_H^k(x) &= \max_a [Q_{r,H}^k(x, a) + Y_k Q_{g,h}^k(x, a)] \geq \sum_a \pi(a|x)[Q_{r,H}^\pi(x, a) + Y_k Q_{g,H}^\pi(x, a)] \\
&= V_H^{\pi, Y_k}(x)
\end{aligned}
\tag{76}
$$

for any policy $\pi$. Thus, it also holds for $\pi^*$. Hence, from Lemma 19, we have

$$
V_H^{\bar{\pi}^*, Y_k}(x) - V_H^k(x) \leq \frac{\log(|\mathcal{A}|)}{\alpha}
$$

Now, suppose that it is true till the step $h + 1$ and consider the step $h$.

Since, it is true till step $h + 1$, thus, for any policy $\pi$,

$$
\mathbb{P}(V_{h+1}^{\pi, Y_k} - V_{h+1}^k)(x, a) \leq \frac{(H - h)\log(|\mathcal{A}|)}{\alpha}
\tag{77}
$$

From (50) in Lemma 12 and the above result, we have for any $(x, a)$

$$
Q_{r,h}^\pi(x, a) + Y_k Q_{g,h}^\pi(x, a) \leq Q_{r,h}^k(x, a) + Y_k Q_{g,h}^k(x, a) + \frac{(H - h)\log(|\mathcal{A}|)}{\alpha}
\tag{78}
$$

Hence,

$$
V_h^{\pi, Y_k}(x) \leq \bar{V}_h^k(x) + \frac{(H - h)\log(|\mathcal{A}|)}{\alpha}
\tag{79}
$$

Now, again from Lemma 19, we have $\bar{V}_h^k(x) - V_h^k(x) \leq \frac{\log(|\mathcal{A}|)}{\alpha}$. Thus,

$$V_h^{\pi, Y_k}(x) - V_h^k(x) \leq \frac{(H - h + 1)\log(|\mathcal{A}|)}{\alpha} \tag{80}$$

Now, since it is true for any policy $\pi$, it will be true for $\pi^*$. From the definition of $V^{\pi, Y_k}$, we have

$$\left(V_{r,h}^{\bar{\pi}^*}(x) + Y_k V_{g,h}^{\bar{\pi}^*}(x)\right) - \left(V_{r,h}^k(x) + Y_k V_{g,h}^k(x)\right) \leq \frac{(H - h + 1)\log(|\mathcal{A}|)}{\alpha} \tag{81}$$

Hence, the result follows by summing over $K$ and considering $h = 1$. $\qquad\square$

## G.8 Proof of Lemma 14

*Proof.* By Lemma 11, for any $x, h, a, k$

$$\langle w_{\diamond,h}^k(x,a), \phi(x,a)\rangle + \beta\sqrt{\phi(x,a)^T(\Lambda^k)^{-1}\phi(x,a)} - Q_{\diamond,h}^{\pi_k}$$
$$\leq \mathbb{P}(V_{\diamond,h+1}^k - V_{\diamond,h+1}^{\pi_k})(x,a) + 2\beta\sqrt{\phi(x,a)^T(\Lambda^k)^{-1}\phi(x,a)} \tag{82}$$

Thus,

$$Q_{\diamond,h}^k(x,a) - Q_{\diamond,h}^{\pi_k}(x,a) \leq \mathbb{P}(V_{\diamond,h+1}^k - V_{\diamond,h+1}^{\pi_k})(x,a) + 2\beta\sqrt{\phi(x,a)^T(\Lambda^k)^{-1}\phi(x,a)}$$
$$\mathbb{P}(V_{\diamond,h+1}^k - V_{\diamond,h+1}^{\pi_k})(x,a) + 2\beta\sqrt{\phi(x,a)^T(\Lambda^k)^{-1}\phi(x,a)} - (Q_{\diamond,h}^k(x,a) - Q_{\diamond,h}^{\pi_k}(x,a)) \geq 0 \tag{83}$$

Since $V_{\diamond,h}^k(x) = \sum_a \pi_{h,k}(a|x)Q_{\diamond,h}^k(x,a)$ and $V_{\diamond,h}^{\pi_k}(x) = \sum_a \pi_{h,k}(a|x)Q_{\diamond,h}^{\pi_k}(x,a)$ where $\pi_{h,k}(a|\cdot) = \text{SOFT-MAX}_\alpha^a(Q_{r,h}^k + Y_k Q_{g,h}^k)\,\forall a$.

Thus, from (83),

$$V_{\diamond,h}^k(x_h^k) - V_{\diamond,h}^{\pi_k}(x_h^k) = \sum_a \pi_{h,k}(a|x_h^k)[Q_{\diamond,h}^k(x_h^k,a) - Q_{\diamond,h}^{\pi_k}(x_h^k,a)]$$

$$\leq \sum_a \pi_{h,k}(a|x_h^k)[Q_{\diamond,h}^k(x_h^k,a) - Q_{\diamond,h}^{\pi_k}(x_h^k,a)]$$

$$+ 2\beta\sqrt{\phi(x_h^k,a_h^k)^T(\Lambda^k)^{-1}\phi(x_h^k,a_h^k)} + \mathbb{P}(V_{\diamond,h+1}^k - V_{\diamond,h+1}^{\pi_k})(x_h^k,a_h^k) - (Q_{\diamond,h}^k(x_h^k,a_h^k) - Q_{\diamond,h}^{\pi_k}(x_h^k,a_h^k)) \tag{84}$$

Thus, from (84), we have

$$V_{\diamond,h}^k(x_h^k) - V_{\diamond,h}^{\pi_k}(x_h^k) \leq D_{\diamond,h,1}^k + D_{\diamond,h,2}^k + [V_{\diamond,h+1}^k - V_{\diamond,h+1}^{\pi_k}](x_{h+1}^k) + 2\beta\sqrt{\phi(x_h^k,a_h^k)^T(\Lambda^k)^{-1}\phi(x_h^k,a_h^k)} \tag{85}$$

Hence, by iterating recursively, we have

$$V_{\diamond,1}^k(x_1) - V_{\diamond,1}^{\pi_k} \leq \sum_{h=1}^H (D_{\diamond,h,1}^k + D_{\diamond,h,2}^k) + \sum_{h=1}^H 2\beta\sqrt{\phi(x_h^k,a_h^k)^T(\Lambda^k)^{-1}\phi(x_h^k,a_h^k)} \tag{86}$$

The result follows. $\qquad\square$

## G.9 Proof of Lemma 15

*Proof.* Note from Lemma 14, we have with probability $1 - 2\delta$,

$$\sum_{k=1}^K V_{\diamond,1}^k(x_1) - V_{\diamond,1}^{\pi_k}(x_1) \leq \sum_{k=1}^K \sum_{h=1}^H (D_{\diamond,h,1}^k + D_{\diamond,h,2}^k) + \sum_{k=1}^K \sum_{h=1}^H 2\beta\sqrt{\phi(x_h^k,a_h^k)^T(\Lambda^k)^{-1}\phi(x_h^k,a_h^k)} \tag{87}$$

We, now, bound the individual terms. First, we show that the first term corresponds to a Martingale difference.

For any $(k, h) \in [K] \times [H]$, we define $\mathcal{F}_{h,1}^k$ as $\sigma$-algebra generated by the state-action sequences, reward, and constraint values, $\{(x_i^\tau, a_i^\tau)\}_{(\tau,i) \in [k-1] \times [H]} \cup \{(x_i^k, a_i^k)\}_{i \in [h]}$.

Similarly, we define the $\mathcal{F}_{h,2}^k$ as the $\sigma$-algebra generated by $\{(x_i^\tau, a_i^\tau)\}_{(\tau,i) \in [k-1] \times [H]} \cup \{(x_i^k, a_i^k)\}_{i \in [h]} \cup \{x_{h+1}^k\}$. $x_{H+1}^k$ is a null state for any $k \in [K]$.

A filtration is a sequence of $\sigma$-algebras $\{\mathcal{F}_{h,m}^k\}_{(k,h,m) \in [K] \times [H] \times [2]}$ in terms of time index

$$t(k, h, m) = 2(k-1)H + 2(h-1) + m \tag{88}$$

which holds that $\mathcal{F}_{h,m}^k \subset \mathcal{F}_{h',m'}^{k'}$ for any $t \leq t'$.

Note from the definitions in (52) that $D_{\diamond,h,1}^k \in \mathcal{F}_{h,1}^k$ and $D_{\diamond,h,2}^k \in \mathcal{F}_{h,2}^k$. Thus, for any $(k, h) \in [K] \times [H]$,

$$\mathbb{E}[D_{\diamond,h,1}^k | \mathcal{F}_{h-1,2}^k] = 0, \quad \mathbb{E}[D_{\diamond,h,2}^k | \mathcal{F}_{h,1}^k] = 0 \tag{89}$$

Notice that $t(k, 0, 2) = t(k-1, H, 2) = 2(H-1)k$. Clearly, $\mathcal{F}_{0,2}^k = \mathcal{F}_{H,2}^{k-1}$ for any $k \geq 2$. Let $\mathcal{F}_{0,2}^1$ be empty. We define a Martingale sequence

$$M_{\diamond,h,m}^k = \sum_{\tau=1}^{k-1} \sum_{i=1}^{H} (D_{\diamond,i,1}^\tau + D_{j,i,2}^\tau) + \sum_{i=1}^{h-1} (D_{\diamond,i,1}^k + D_{\diamond,i,2}^k) + \sum_{l=1}^{m} D_{j,h,l}^k$$

$$= \sum_{(\tau,i,l) \in [K] \times [H] \times [2], t(\tau,i,l) \leq t(k,h,m)} D_{\diamond,i,l}^\tau \tag{90}$$

where $t(k, h, m) = 2(k-1)H + 2(h-1) + m$ is the time index. Clearly, this martingale is adopted to the filtration $\{\mathcal{F}_{h,m}^k\}_{(k,h,m) \in [K] \times [H] \times [2]}$, and particularly

$$\sum_{k=1}^{K} \sum_{h=1}^{H} (D_{j,h,1}^k + D_{j,h,2}^k) = M_{\diamond,H,2}^K \tag{91}$$

Thus, $M_{\diamond,H,2}^K$ is a Martingale difference satisfying $|M_{\diamond,H,2}^K| \leq 4H$ since $|D_{\diamond,h,1}^k|, |D_{\diamond,h,2}^k| \leq 2H$ From the Azuma-Hoeffding inequality, we have

$$\Pr(M_{\diamond,H,2}^K > s) \leq 2 \exp(-\frac{s^2}{16TH^2}) \tag{92}$$

With probability $1 - 2\delta$ at least for any $\diamond = r, g$,

$$\sum_k \sum_h M_{j,H,2}^K \leq \sqrt{16TH^2 \log(2/\delta)} \tag{93}$$

Now, we bound the second term. Note that the minimum eigen value of $\Lambda_h^k$ is at least $\lambda = 1$ for all $(k, h) \in [K] \times [H]$. By Lemma 20,

$$\sum_{k=1}^{K} \sum_h (\phi_h^k)^T (\Lambda_h^k)^{-1} \phi_h^k \leq 2 \log \left[ \frac{\det(\Lambda^{K+1})}{\det(\Lambda_0)} \right] \tag{94}$$

Moreover, note that $||\Lambda^{K+1}|| = ||\sum_{\tau=1}^{k} \sum_h \phi_h^k (\phi_h^k)^T + \lambda \mathbf{I}|| \leq \lambda + T$, hence,

$$\sum_{k=1}^{K} \sum_h (\phi_h^k)^T (\Lambda^k)^{-1} \phi_h^k \leq 2d \log \left[ \frac{\lambda + T}{\lambda} \right] \leq 2d\iota \tag{95}$$

Now, by Cauchy-Schwartz inequality, we have

$$\sum_{k=1}^{K} \sum_{h=1}^{H} \sqrt{(\phi_h^k)^T (\Lambda^k)^{-1} \phi_h^k} \leq \sqrt{KH} [\sum_{k=1}^{K} \sum_{h=1}^{H} (\phi_h^k)^T (\Lambda^k)^{-1} \phi_h^k]^{1/2}$$

$$\leq \sqrt{2dT\iota} \tag{96}$$

Note that $\beta = C_1 dH\sqrt{\iota}$.

Thus, we have with probability $1 - 4\delta$,

$$\sum_{k=1}^{K} V_{r,1}^k(x_1^k) - V_{r,1}^{\pi_k}(x_1^k) + Y\sum_{k=1}^{K}(V_{g,1}^k(x_1^k) - V_{g,1}^{\pi_k}(x_1^k))$$
$$\leq (Y+1)[\sqrt{2TH^2\log(4/p)} + C_4\sqrt{d^3H^2T\iota^2}] \qquad (97)$$

Hence, the result follows. $\qquad\square$

### G.10 PROOF OF THEOREM 2

Note from Lemma 9, Lemma 13, and Lemma 15, we have w.p. $1 - 4\delta$,

$$\sum_{k=1}^{K}(V_{r,1}^{\bar{\pi}^*}(x_1) - V_{r,1}^{\pi_k}(x_1)) + Y\sum_{k=1}^{K}(Hb - \kappa - V_{g,1}^{\pi_k}(x_1^k))$$
$$\leq \frac{Y^2}{2\eta} + \frac{\eta}{2}H^2K + \frac{HK\log|\mathcal{A}|}{\alpha} + \tilde{\mathcal{O}}((Y+1)\sqrt{d^3H^2T\iota^2}) \qquad (98)$$

Replacing $Y$ with 0 in (98), we have

$$\sum_{k=1}^{K}(V_{r,1}^{\bar{\pi}^*}(x_1^k) - V_{r,1}^{\pi_k}(x_1^k) \leq \frac{\eta}{2}H^2K + \frac{HK\log|\mathcal{A}|}{\alpha} + \mathcal{O}(\sqrt{d^3H^2T\iota^2})$$

By noting that $\eta = \frac{\xi}{\sqrt{KH^2}}$, and $\alpha = \frac{\log|\mathcal{A}|\sqrt{KH}}{2(1+\xi+H)}$, we have

$$\sum_{k=1}^{K}(V_{r,1}^{\bar{\pi}^*}(x_1^k) - V_{r,1}^{\pi_k}(x_1^k)) \leq \frac{\xi\sqrt{KH^2}}{2} + \sqrt{KH}(2(1+\xi+H)) + \mathcal{O}(\sqrt{d^3TH^2\iota^2})$$
$$= \tilde{\mathcal{O}}(\sqrt{d^3TH^2}) \qquad (99)$$

where the last equality follows from the fact that $KH = T$.

Now, observe the third term in (47). Note from Azuma-Hoeffding inequality (since $\mathbb{E}[V_{r,1}(\pi^k)(x_1^k) - \sum_{t=(k-1)H+1}^{kH}(r(x_h, a_h))] = 0$ with respect to the Filtration $\mathcal{F}_{k-1}$). Hence, with probability $1 - \delta$

$$\sum_{k=1}^{T/H}(V_{r,1}^{\pi_k}(x_1^k) - \sum_{t=(k-1)H+1}^{kH}(r(x_h, a_h))) \leq H\sqrt{T/H\log(2/\delta)}$$
$$\leq \mathcal{O}(\sqrt{TH\iota}) \qquad (100)$$

Now, using Lemma 3, (45), (99), and (100), in (44) we obtain with probability $1 - 5\delta$,

$$\text{Regret}(T) \leq (T/H)\text{sp}(v_r^*) + \tilde{\mathcal{O}}(\sqrt{d^3TH^2}) + \tilde{\mathcal{O}}(\sqrt{TH}) \qquad (101)$$

Since $H = T^{1/4}/d^{3/4}$, we obtain with probability $1 - 5\delta$,

$$\text{Regret}(T) \leq T^{3/4}d^{3/4}\text{sp}(v_r^*) + \tilde{\mathcal{O}}((dT)^{3/4}) = (1 + \text{sp}(v_r^*))\tilde{\mathcal{O}}((dT)^{3/4}) \qquad (102)$$

Now, we bound Violation. By noting that $\eta = \frac{\xi}{\sqrt{KH^2}}$, and $\alpha = \frac{\log|\mathcal{A}|\sqrt{KH}}{2(1+\xi+H)}$, we have from (98),

$$\sum_{k=1}^{K}(V_{r,1}^{\bar{\pi}^*}(x_1) - V_{r,1}^{\pi_k}(x_1^k)) + Y\sum_{k=1}^{K}(Hb - \kappa - V_{g,1}^{\pi_k}(x_1^k))$$
$$\leq \xi\sqrt{KH^2} + \sqrt{KH}(2(1+\xi+H)) + \mathcal{O}((\xi+1)\sqrt{d^3H^2T\iota^2}) \qquad (103)$$

From the convex optimization result (Corollary 1), we obtain

$$\sum_{k=1}^{K}(Hb - \kappa - V_{g,1}^{\pi_k}(x_1^k)) \leq \frac{2(1+\xi)}{\xi}\tilde{\mathcal{O}}(\sqrt{d^3 H^2 T}) \tag{104}$$

Also note from Azuma-Hoeffding inequality, w.p. $1 - \delta$

$$\sum_{k=1}^{T/H}(V_{g,1}^{\pi_k}(x_1^k) - \sum_{t=(k-1)H+1}^{kH} g(x_t, a_t)) \leq H\sqrt{T/H \log(2/\delta)} \tag{105}$$

Hence, combining (104) and (105), we obtain w.p. $1 - 5\delta$,

$$\sum_{k=1}^{K}(Hb - \kappa - V_{g,1}^{\pi_k}(x_1^k)) + \sum_{k=1}^{T/H}(V_{g,1}^{\pi_k}(x_1^k) - \sum_{t=(k-1)H+1}^{kH} g(x_t, a_t)) \leq \tilde{\mathcal{O}}(\frac{(1+\xi)}{\xi}\sqrt{d^3 T H^2})$$

$$\text{Violation}(T) \leq (T/H)\kappa + \tilde{\mathcal{O}}(\frac{(1+\xi)}{\xi}\sqrt{d^3 T H^2})$$

$$= (1+\kappa)\tilde{\mathcal{O}}(\frac{(1+\xi)}{\xi}(dT)^{3/4}) \tag{106}$$

where the last equality follows from the fact that $H = T^{1/4}/d^{3/4}$.

## G.11 Differences in Analysis for Algorithms 1 and 2

Note that we also need to show that log $\epsilon$-covering number for $v$ function class is also $\log(T)$ in Algorithm 1. However, since Algorithm 1 and Algorithm 2 are different, the analysis is also different. In particular, in Algorithm 2 we consider a primal-dual type algorithm, hence, the policy is based on the combined state-action value function. Thus, we need to show that the log $\epsilon$-covering number for individual value function scales at most with $\log(T)$ even though the policy is based on the composite $Q$-function. On the other hand, in Algorithm 1, we directly search for the policy which solves a constrained optimization problem. In Algorithm 2 the policy is itself a function of $Q_\diamond$ and dual variable $Y_k$. Thus, we obtain the result only using the property of the function class of $Q_\diamond$ and the boundedness of $Y_k$. On the other hand a separate policy function class is defined for Algorithm 1 and we obtain the result by exploiting of the smoothness property of the function class. Thus, our analysis would work for any policy class with smoothness property (Appendix F.6) unlike Algorithm 2. We need that the optimal policy should also belong to $\Pi$ in Algorithm 1. In contrast, in Algorithm 2 we consider an unconstrained version of the episodic CMDP setup. Carefully crafted Soft-max policy enables us to bound the gap with the optimal policy in the episodic setup. Hence, we do not need to search over a feasible policy space unlike in Algorithm 1.

## G.12 Supporting Results

The following result is shown in Abbasi-Yadkori et al. (2011) and in Lemma D.2 in Jin et al. (2020).

**Lemma 20.** *Let $\{\phi_t\}_{t\geq 0}$ be a sequence in $\Re^d$ satisfying $\sup_{t\geq 0}||\phi_t|| \leq 1$. For any $t \geq 0$, we define $\Lambda_K = \Lambda_0 + \sum_{k=1}^{K}\sum_h \phi_h^k(\phi_h^k)^T$. Then if the smallest eigen value of $\Lambda_0$ be at least 1, we have*

$$\log\left[\frac{\det(\Lambda_K)}{\det(\Lambda_0)}\right] \leq \sum_{k=1}^{K}\sum_h (\phi_h^k)^T(\Lambda^{k-1})^{-1}\phi_h^k \leq 2\log\left[\frac{\det(\Lambda_K)}{\det(\Lambda_0)}\right] \tag{107}$$

**Theorem 4.** *[Concentration of Self-Normalized Process Abbasi-Yadkori et al. (2011)] Let $\{\epsilon_t\}_{t=1}^{\infty}$ be a real-valued stochastic process with corresponding filtration $\{\mathcal{F}_t\}_{t=0}^{\infty}$. Let $\epsilon_t|\mathcal{F}_{t-1}$ be a zero mean and $\sigma$ sub-Gaussian, i.e., $\mathbb{E}[\epsilon_t|\mathcal{F}_{t-1}] = 0$, and*

$$\forall \zeta \in \Re, \quad \mathbb{E}[e^{\zeta\epsilon_t}|\mathcal{F}_{t-1}] \leq e^{\zeta^2\sigma^2/2}. \tag{108}$$

*Let $\{\phi_t\}_{t=1}^{\infty}$ be a $\Re^d$-valued Stochastic process where $\phi_t \in \mathcal{F}_{t-1}$. Assume $\Lambda_0 \in \Re^{d\times d}$ is a positive-define matrix, let, $t = kH$, $\Lambda_t = \Lambda_0 + \sum_{j=1}^{k}\sum_h \phi_h^j(\phi_h^j)^T$. Then for any $\delta > 0$ with probability at least $1 - \delta$, we have*

$$||\sum_{s=1}^{t}\phi_s\epsilon_s||_{\Lambda_t^{-1}}^2 \leq 2\sigma^2 \log\left[\frac{\det(\Lambda_t)^{1/2}\det(\Lambda_0)^{-1/2}}{\delta}\right] \tag{109}$$

The next result characterizes the covering number of an Euclidean ball (Lemma 5.2 in Vershynin (2010)).

**Lemma 21.** *Let $\{x_t\}_{t=1}^{\infty}$ be a stochastic process on state space $\mathcal{X}$ with corresponding filtration $\{\mathcal{F}_t\}$, $\{\phi_t\}$ be a $\mathbb{R}^d$-values stochastic process where $\phi_t \in \mathcal{F}_{t-1}$ and $||\phi_t|| \leq 1$, $\Lambda_t = \lambda \mathbf{I} + \sum_{s=1}^{t-1} \phi_s \phi_s^T$, and $\mathcal{V}$ be an arbitrary set of functions defined on $\mathcal{X}$ with $N_\epsilon$ be its $\epsilon$-covering number with respect to $dist(V, V') = \sup_x |V(x) - V'(x)|$ for some fixed $\epsilon > 0$. Then for any $\delta > 0$ with probability $1 - \delta$ for any $t$, $v \in \mathcal{V}$ so that $\sup_x |v(x)| \leq H$, we have*

$$\left\| \sum_{s=1}^{t-1} \phi_s(v(x_s) - \mathbb{E}[v(x_s)|\mathcal{F}_{t-1}]) \right\|_{\Lambda_t^{-1}}^2 \leq 4H^2 \left[ \frac{d}{2} \log\left(\frac{t+\lambda}{\lambda}\right) + \log \frac{N_\epsilon}{\delta} \right] + 8t^2 \epsilon^2 / \lambda^2 \quad (110)$$

**Observation 1.** *Let $\Lambda_k = \sum_{\tau=1}^{k-1} \sum_{h'} \phi_{h'}^\tau (\phi_{h'}^\tau)^T + \lambda \mathbf{I}$, where $\lambda > 0$, and $\phi_{h'}^\tau \in \mathbb{R}^d$, then*

$$\sum_{\tau=1}^{k-1} \sum_{h'} (\phi_{h'}^\tau)^T (\Lambda^k)^{-1} \phi_{h'}^\tau \leq d$$

*Proof.* We have $\sum_{\tau=1}^{k-1} \sum_{h'} (\phi_{h'}^\tau)^T (\Lambda^k)^{-1} \phi_{h'}^\tau = \sum_{\tau=1}^{k-1} \sum_{h'} \text{tr}((\phi_{h'}^\tau)^T (\Lambda^k)^{-1} \phi_{h'}^\tau)$. Further, $\sum_{\tau=1}^{k-1} \sum_{h'} \text{tr}((\phi_{h'}^\tau)^T (\Lambda^k)^{-1} \phi_{h'}^\tau) = \text{tr}((\Lambda_k)^{-1} \sum_{\tau=1}^{k-1} \sum_{h'} \phi_{h'}^\tau (\phi_{h'}^\tau)^T)$. Given the eigen value decomposition, $\sum_{\tau=1}^{k-1} \sum_{h'} \phi_{h'}^\tau (\phi_{h'}^\tau)^T = \mathbf{U} \text{diag}(\lambda_1, \ldots, \lambda_d) \mathbf{U}^T$. Thus, $\Lambda_k = \mathbf{U} \text{diag}(\lambda_1 + \lambda, \ldots, \lambda_d + \lambda) \mathbf{U}^T$.

Hence, $\text{tr}((\Lambda_k)^{-1} \sum_{\tau=1}^{k-1} \sum_{h'} \phi_{h'}^\tau (\phi_{h'}^\tau)^T) = \sum_{l=1}^d \lambda_l / (\lambda_l + d) \leq d$. $\qquad \square$

**Lemma 22.** *[Covering Number of Euclidean Ball] For any $\epsilon > 0$, the $\epsilon$-covering number of the Euclidean ball in $\mathbb{R}^d$ with radius $R$ is upper bounded by $(1 + 2R/\epsilon)^d$.*

We have used the following result from the optimization which is proved in Lemma 9 in Ding et al. (2021).

**Lemma 23.** *Let $Y^*$ be the optimal dual variable of the episodic CMDP (46), and $C \geq 2Y^*$, then, if*

$$V_{r,1}^{\bar{\pi}^*}(x_1^k) - V_{r,1}^{\pi_k}(x_1^k) + C(b - V_{g,1}^{\pi_k}(x_1^k)) \leq \delta_k \quad (111)$$

*then*

$$[b - V_{g,1}^{\pi_k}(x_1^k)]_+ \leq \frac{2\delta_k}{C}. \quad (112)$$

**Corollary 1.** *Let $Y^*$ be the optimal dual variable, and $C \geq 2Y^*$, if*

$$\sum_{k=1}^K V_{r,1}^{\bar{\pi}^*}(x_1^k) - V_{r,1}^{\pi_k}(x_1^k) + C(b - V_{g,1}^{\pi_k}(x_1^k)) \leq \delta \quad (113)$$

*then*

$$\sum_{k=1}^K (b - V_{g,1}^{\pi_k}(x_1^k)) \leq \frac{2\delta}{C} \quad (114)$$

*Proof.* Let $\delta_k = V_{r,1}^{\bar{\pi}^*}(x_1^k) - V_{r,1}^{\pi_k}(x_1^k) + C(b - V_{g,1}^{\pi_k}(x_1^k))$, then from Lemma 23,

$$(b - V_{g,1}^{\pi_k}(x_1^k))_+ \leq (b - V_{g,1}^{\pi_k}(x_1^k)) \leq \frac{2\delta_k}{C} \quad (115)$$

By summing over $k$, we obtain

$$\sum_{k=1}^K (b - V_{g,1}^{\pi_k}(x_1^k)) \leq \sum_k \frac{2\delta_k}{C} \leq \frac{2\delta}{C} \quad (116)$$

where the last inequality follows from the fact that $\sum_k \delta_k \leq \delta$. $\qquad \square$

# H PROOF OF RESULTS IN SECTION 4

**Organization**: We, first, provide a proof sketch of Theorem 3 in (Appendix H.1). Subsequently, we provide a detailed proof. We show that how we can achieve zero constraint violation via tweaking the algorithm in Appendix H.5.

**Notations for this Section**: As shown in Appendix A of Wei et al. (2021a), we assume that $||\phi(x,a)||_2 \leq 1$. Without loss of generality, we assume that we represent $\phi(x,a)^T e_1 = 1$ for some vector $e_1$ for every $(x,a)$. $||\mu(\mathcal{S})||_2 \leq \sqrt{d}$, $||\theta_j||_2 \leq \sqrt{d}$ for $j = r, g$. Also, for every policy $\pi$, $q^\pi(x,a)$ can be written as $\phi(x,a)^T w^\pi$. $||w^\pi|| \leq 6 t_{mix}\sqrt{d}$ (by Lemma 34). We denote $q_j^k(x,a)$ as $(w_j^k)^T \phi(x,a)$.

## H.1 PROOF SKETCH OF THEOREM 3

Regret can be decomposed as the following

$$\text{Regret}(T) = \sum_{k=1}^{K} B[J_r^* - J_r^{\pi_k}] + [\sum_k \sum_{t=(k-1)B+1}^{kB} (J_r^{\pi_k} - r(x_t, a_t))] \tag{117}$$

In the following, we bound each of the two terms at the right hand side. Unlike the unconstrained setup Wei et al. (2021a), here the decision depends on both $q_r^k$ and $q_g^k$, thus, the analysis will also be different. In particular, in order to bound the first term, we bound $\sum_{k=1}^{T/B} B[J_r^* - J_r^{\pi_k}] + BY(b - J_g^{\pi_k})$. Specifically, since $J_g^* \geq b$, we obtain for any $Y \in [0, \xi]$

$$\sum_{k=1}^{T/B} B[J_r^* - J_r^{\pi_k}] + BY(b - J_g^{\pi_k}) \leq$$

$$B \underbrace{\sum_{k=1}^{T/B}[J_r^* + Y_k J_g^* - J_r^{\pi_k} - Y_k J_g^{\pi_k}]}_{\mathcal{T}_1} + \underbrace{(Y - Y_k)B(\hat{J}^k - J_g^{\pi_k})}_{\mathcal{T}_2} + B\underbrace{(Y - Y_k)(b - \hat{J}^k)}_{\mathcal{T}_3}$$

Note that since the above expression is true for any $Y \in [0, \xi]$, we can recover the bound on regret by setting $Y = 0$.

For $\mathcal{T}_1$, we employ the value-difference lemma. *This step may seem similar to the one considered for the unconstrained version (Wei et al., 2021a) which bounds $\sum_k (J_r^* - J_r^{\pi_k})$. However, there are two important differences. We have an additional constraint term scaled by the dual variable $Y_k$ which also changes over time. Second, we provide high probability bound instead of expectation bound in Wei et al. (2021a). We utilize the one-step decent result of OMD from Lemma 3.3 in Cai et al. (2020), Azuma-Hoeffding inequality, and the boundedness of $||w_\diamond^k||$ using Assumption 6 to bound the term. The final result is shown in Lemma 24.*

**Lemma 24.** *With probability $1 - \delta$,*

$$\mathcal{T}_1 \leq BK\mathcal{O}(\alpha(1+\xi)^2 N^2/\sigma^2) + \frac{B}{\alpha}\log(|\mathcal{A}|) \tag{118}$$

*Proof.* See Appendix H.1.1. □

Note that in the unconstrained setup the term $\mathcal{T}_2$ does not arise. For $\mathcal{T}_2$, we rely on the fact $\mathbb{E}_k(\hat{J}^k - J_g^{\pi_k})$ are close since the way we collected samples $\hat{J}^k$ is close to an unbiased estimator for $J_g^{\pi_k}$ (Lemma 33). Using the fact that $|Y - Y_k| \leq \xi$, we bound $\mathcal{T}_2$ as the following

**Lemma 25.** *With probability $1 - \delta$,*

$$\mathcal{T}_2 \leq \frac{\xi}{T} + \xi\sqrt{TB\log(2/\delta)} \tag{119}$$

*Proof.* See Appendix H.1.2. □

In order to bound $\mathcal{T}_3$, we utilize the result from the dual-domain analysis (Corollary 3). We show that the upper bound of $\mathcal{T}_3 \leq \xi\sqrt{TB}$.

Hence, we have with probability $1 - 2\delta$,

$$\sum_{k=1}^{T/B} B[J_r^* - J_r^{\pi_k}] + BY(b - J_g^{\pi_k}) \leq$$

$$BK\mathcal{O}(\alpha(1+\xi)^2 N^2/\sigma^2) + \frac{B\log(|\mathcal{A}|)}{\alpha} + \mathcal{O}(\xi\sqrt{TB\log(2/\delta)}) \quad (120)$$

In order to bound the second term in (117), we need to show that policy between two epochs change slowly. Here, we use the value of the hyper-parameter $\alpha$, and the upper bound of the dual variable to bound the term

**Lemma 26.** *With probability* $1 - 2\delta$,

$$\sum_{t=(k-1)B+1}^{kB} (J_r^{\pi_k} - r(x_t, a_t)) \leq \tilde{\mathcal{O}}((1+\xi)\sqrt{Tt_{mix}^3}) \quad (121)$$

*Proof.* See Appendix H.2 □

Hence, combining the above, setting $Y = 0$ and replacing (120) in (117) we obtain with probability $1 - 4\delta$

$$\text{Regret}(T) \leq K\tilde{\mathcal{O}}(\alpha(1+\xi)^2/\sigma^2) + \frac{B}{\alpha}\log(|\mathcal{A}|) + \tilde{\mathcal{O}}((1+\xi)\sqrt{Tt_{mix}^3}) \quad (122)$$

Now, replacing the value of $\alpha$, and $B$, we obtain

$$\text{Regret}(T) \leq \tilde{\mathcal{O}}((1+\xi)/\sigma\sqrt{Tt_{mix}^3}) \quad (123)$$

**Constraint Violation**: Similar to the regret step, we decompose the violation in the following form

$$\sum_k B(b - J_g^{\pi_k}) + \sum_k (J_g^{\pi_k} - \sum_{t=(k-1)B+1}^{B} g(x_t, a_t))$$

The second term can be upper bounded in the similar way as the second term in (117) with probability $1 - 2\delta$. For the first term, we observe

$$\sum_k B(b - J_g^{\pi_k}) \leq \sum_k B(J_g^* - J_g^{\pi_k}) \quad (124)$$

Now, from (120), we have with probability $1 - 2\delta$

$$\sum_k B[J_r^* - J_r^{\pi_k}] + BY(b - J_g^{\pi_k}) \leq \mathcal{O}((1+\xi)/\sigma\sqrt{TB\log(2/p)}) \quad (125)$$

Finally, by replacing $B$ and applying Corollary 4, we obtain with probability $1 - 2\delta$

$$\sum_k B(b - J_g^{\pi_k}) \leq \frac{2(1+\xi)}{\xi\sigma}\tilde{\mathcal{O}}(\sqrt{Tt_{mix}^3}) \quad (126)$$

Hence, the result follows.

### H.1.1 PROOF OF LEMMA 24

We, first, state Lemmas 27,28,29 which will be useful for proving Lemma 24.

**Lemma 27.** *Using the value difference lemma (Lemma 15) in Wei et al. (2020), we have*

$$J_r^* + Y_k J_g^* - J_r^{\pi_k} - Y_k J_g^{\pi_k} = \int_{\mathcal{S}} \sum_a (\pi^*(a|x) - \pi_k(a|x))(q_r^{\pi_k}(x,a) + Y_k q_g^{\pi_k}(x,a))d\nu^{\pi^*}(x) \quad (127)$$

*where* $\nu^{\pi^*}(x)$ *is the stationary distribution corresponding to the stationary policy* $\pi^*$.

Also note that $\sum_a(\pi^*(a|x) - \pi_k(a|x))N(J_r^{\pi_k} + Y_k J_g^{\pi_k}) = 0$ since $J_\diamond^{\pi_k}$ is constant.

From Lemma 15 in Wei et al. (2021a)

**Lemma 28.** *Let* $\mathbb{E}[\cdot|\tau_{k,m}]$ *denote the expectation conditioned on* $(x_{\tau_{k,m}}, a_{\tau_{k,m}})$ *and all the history before* $\tau_{k,m}$, *then*

$$|\mathbb{E}[R_{k,m}|\tau_{k,m}] - (q_r^{\pi_k}(x_{\tau_{k,m}}, a_{\tau_{k,m}}) + NJ_r^{\pi_k})| \leq \frac{1}{T^7}$$

$$|\mathbb{E}[G_{k,m}|\tau_{k,m}] - (q_g^{\pi_k}(x_{\tau_{k,m}}, a_{\tau_{k,m}}) + NJ_g^{\pi_k})| \leq \frac{1}{T^7} \quad (128)$$

Thus, $R_{k,m}$ and $G_{k,m}$ are close to the unbiased estimator of $q_r^{\pi_k} + NJ_r^{\pi_k}$ and $q_g^{\pi_k} + NK_g^{\pi_k}$ respectively. Also from Lemma 16 in Wei et al. (2021a)

**Lemma 29.** *For* $\diamond = r, g$,

$$||\mathbb{E}_k[w_\diamond^k] - (w_\diamond^{\pi_k} + NJ_\diamond^{\pi_k}e_1)|| \leq \frac{1}{T^2} \quad (129)$$

*where* $\mathbb{E}_k$ *is the expectation conditioned on all history before epoch* $k$.

Now, we are ready to prove Lemma 24.

*Proof* From Lemma 27, We express $\mathcal{T}_1$ in (117) as

$$B\sum_k\sum_a(\pi^*(a|x) - \pi_k(a|x))(q_r^{\pi_k}(x,a) + Y_k q_g^{\pi_k}(x,a) + NJ_r^{\pi_k} + Y_k NJ_g^{\pi_k}) =$$

$$B\underbrace{\sum_k\sum_a(\pi^*(a|x) - \pi_k(a|x))((q_r^{\pi_k}(x,a) + Y_k q_g^{\pi_k}(x,a) + NJ_r^{\pi_k} + Y_k NJ_g^{\pi_k}) - \mathbb{E}_k[q_r^k(x,a) + Y_k q_g^k(x,a)])}_{\mathcal{T}_4}$$

$$+ B\underbrace{\sum_k\sum_a(\pi^*(a|x) - \pi_k(a|x))(\mathbb{E}_k[q_r^k(x,a) + Y_k q_g^k(x,a)] - (q_r^k(x,a) + Y_k q_g^k(x,a)))}_{\mathcal{T}_5}$$

$$+ B\underbrace{\sum_k\sum_a(\pi^*(a|x) - \pi_k(a|x))(q_r^k(x,a) + Y_k q_g^k(x,a))}_{\mathcal{T}_6} \quad (130)$$

**Bounding $\mathcal{T}_4$:** The bound of $\mathcal{T}_4$ is similar to the unconstrained case Wei et al. (2021a). The only difference is that since we have extra term corresponding to $Y_k q_g^k$, there is an additional scaling factor of $(1 + \xi)$. The details are in the following–

Note that $q_\diamond^{\pi_k}(x,a) = (w_\diamond^{\pi_k})^T\phi(x,a)$, $J_\diamond^{\pi_k} = J_\diamond^{\pi_k}e_1^T\phi(x,a)$, and $q_\diamond^k(x,a) = (w_\diamond^k)^T\phi(x,a)$.

From Lemma 29, we obtain that

$$|\mathbb{E}_k[w_\diamond^k]^T\phi(x,a) - (w_\diamond^{\pi_k} + NJ_\diamond^{\pi_k}e_1)^T\phi(x,a)| \leq ||\mathbb{E}_k[w_\diamond^k] - (w_\diamond^{\pi_k} + NJ_\diamond^{\pi_k}e_1)^T||||\phi(x,a)||$$
$$\leq \frac{1}{T^2}$$

Hence,

$$(\pi^*(a|x) - \pi_k(a|x))((q_r^{\pi_k}(x,a) + Y_k q_g^{\pi_k}(x,a) + NJ_r^{\pi_k} + Y_k NJ_g^{\pi_k}) - \mathbb{E}_k[q_r^k(x,a) + Y_k q_g^k(x,a)]) \leq$$
$$||\pi^*(\cdot|x) - \pi_k(\cdot|x)||_1||(q_r^{\pi_k}(x,a) + Y_k q_g^{\pi_k}(x,a) + NJ_r^{\pi_k} + Y_k NJ_g^{\pi_k}) - \mathbb{E}_k[q_r^k(x,a) + Y_k q_g^k(x,a)]||_\infty$$
$$\leq 2\frac{1+\xi}{T^2}$$

Hence,

$$\mathcal{T}_4 \leq 2\frac{1+\xi}{T} \quad (131)$$

**Bounding $\mathcal{T}_5$: Since we are providing high probability bound instead unlike the expected regret bound in Wei et al. (2021a), we need to bound $\mathcal{T}_5$ whereas such a term did not arise in Wei et al. (2021a).**

$\mathcal{T}_5$ is Martingale difference with respect to the Filtration $\mathcal{F}_{k-1}$ which contains all the histories till epoch $k$. $|q_r^k(x,a) + Y_k q_g^k(x,a)| \leq (1+\xi)N/\sigma$ (From Lemma 31). Hence, from Azuma-Hoeffding inequality, we obtain w.p. $1 - \delta$,

$$\sum_k \sum_a (\pi^*(a|x) - \pi_k(a|x))(\mathbb{E}_k[q_r^k(x,a) + Y_k q_g^k(x,a)] - (q_r^k(x,a) + Y_k q_g^k(x,a)))$$

$$\leq \mathcal{O}(2(1+\xi)N/\sigma\sqrt{2T/B\log(2/\delta)}) \quad (132)$$

Thus, multiplying the above with $B$, we obtain

$$\mathcal{T}_5 \leq \mathcal{O}(2(1+\xi)N/\sigma\sqrt{TB\log(2/\delta)}) \quad (133)$$

with probability $1 - \delta$.

**Bounding $\mathcal{T}_6$:** In the unconstrained case Wei et al. (2021a) bounds $\mathbb{E}[\sum_k \sum_a (\pi^*(a|x) - \pi_k(a|x))q_r^k(x,a)]$ (Lemma 19 in Wei et al. (2021a)). $\mathcal{T}_6$ may seem similar to the above expression, however, there are subtle differences. First, we do not have expectation since we need to consider high-probability bound. Second, we have additional $Y_k q_g^k$ term where $Y_k$ is changing over $k$. **Thus, we need to use different techniques compared to Wei et al. (2021a).** In particular, we rely on the analysis of OMD and one-step decent result from Lemma 3.3 in Cai et al. (2020) to bound $\mathcal{T}_6$. In particular, we use the result from Lemma 3.3 in Cai et al. (2020), to obtain

$$\langle \pi^* - \pi_{k-1}, q_r^{k-1} + Y_{k-1}q_g^{k-1} \rangle \leq \frac{1}{2}\alpha||q_r^{k-1} + Y_k q_g^{k-1}||_\infty^2 + \frac{1}{\alpha}(D(\pi^*|\pi_{k-1}) - D(\pi^*|\pi_k))$$

Now, by the construction of $w_r^{k-1}$ and $w_g^{k-1}$, we have $||q_r^{k-1} + Y_k q_g^{k-1}||_\infty = \mathcal{O}((1+\xi)^2 N^2/\sigma^2)$ (Lemma 31). Finally, by telescope summing over $k$ and using the fact that $D(\cdot)$ is non-negative, we have the result. The details can be found in Lemma 30. Specifically, we use the result from Lemma 30 to bound $\mathcal{T}_6$−

$$\mathcal{T}_6 \leq T\mathcal{O}(\alpha(1+\xi)^2 N^2/\sigma^2) + \frac{B}{\alpha}\log(|\mathcal{A}|) \quad (134)$$

since $KB = T$.

Hence, we have with probability $1 - \delta$,

$$B\sum_{k=1}^K \sum_a (\pi^*(a|x) - \pi_k(a|x))(q_r^{\pi_k}(x,a) + Y_k q_g^{\pi_k}(x,a))$$

$$\leq \frac{2(1+\xi)}{T} + T\mathcal{O}(\alpha(1+\xi)^2 N^2/\sigma^2) + \frac{B}{\alpha}\log(|\mathcal{A}|) + \mathcal{O}((1+\xi)\frac{N}{\sigma}\sqrt{TB\log(2/\delta)}) \quad (135)$$

Since $\alpha = \min\{\frac{\sigma}{(1+\xi)\sqrt{Tt_{mix}}}, \frac{\sigma}{24(1+\xi)N}\}$, $N = 8t_{mix}\log(T)$, $B = 32N\log(dT)\sigma^{-1}$, then,

$$B\sum_k \sum_a (\pi^*(a|x) - \pi_k(a|x))(q_r^{\pi_k}(x,a) + Y_k q_g^{\pi_k}(x,a)) \leq \tilde{\mathcal{O}}((1+\xi)/\sigma\sqrt{Tt_{mix}^3})$$

$$\sum_k \int_\mathcal{X} B\sum_a (\pi^*(a|x) - \pi_k(a|x))(q_r^{\pi_k}(x,a) + Y_k q_g^{\pi_k}(x,a))d\nu^{\pi^*}(x) \leq \tilde{\mathcal{O}}((1+\xi)/\sigma\sqrt{Tt_{mix}^3})$$

where the last inequality follows from the dominated convergence theorem and $\int_\mathcal{X} d\nu^{\pi^*}(x) = 1$.

Hence, we have with probability $1 - \delta$,

$$\mathcal{T}_1 \leq \tilde{\mathcal{O}}((1+\xi)/\sigma\sqrt{Tt_{mix}^3})$$

### H.1.2 PROOF OF LEMMA 25

*Proof.* We can decompose $\mathcal{T}_2$ as

$$\sum_k B(Y - Y_k)(\mathbb{E}_k(\hat{J}^k) - J^{\pi_k}) + B(Y - Y_k)(\hat{J}^k - \mathbb{E}_k(\hat{J}^k)) \tag{136}$$

From Lemma 33, $|\mathbb{E}_k(\hat{J}_k) - J^{\pi_k}| \leq \dfrac{2}{T^7}$. Also note that $(Y - Y_k)(\hat{J}^k - \mathbb{E}_k(\hat{J}^k))$ is a Martingale sequence where $\hat{J}^k \leq 1$. Further, $|Y - Y_k| \leq \xi$. Thus, we have with probability $1 - \delta$,

$$\sum_{k=1}^{K} B(Y - Y_k)(\hat{J}_k - \mathbb{E}_k(\hat{J}^k)) \leq B\xi\sqrt{2T/B \log(2/\delta)} \tag{137}$$

Since $K = T/B$, hence, with probability $1 - \delta$

$$\mathcal{T}_2 \leq \frac{\xi}{T^6} + \xi\sqrt{TB \log(2/\delta)}$$

$\square$

From Corollary 3 and $\eta = \dfrac{\xi}{\sqrt{T/B}}$, we have

$$\mathcal{T}_3 \leq \xi\sqrt{TB} \tag{138}$$

where recall that $\mathcal{T}_3$ is the third term in (117).

### H.2 PROOF OF LEMMA 26

*Proof.* Note that

$$\sum_k \sum_{t=(k-1)B+1}^{kB} (J_r^{\pi_k} - r(x_t, a_t)) = \sum_k \sum_{t=(k-1)B+1} (q_r^{\pi_k}(x_t, a_t) - \mathbb{E}_{x_{t+1}\sim\mathbb{P}(\cdot|(x_t,a_t))}(v_r^{\pi_k}(x_{t+1})))$$

$$= \sum_k \sum_{t=(k-1)B+1} (q_r^{\pi_k}(x_t, a_t) - v_r^{\pi_k}(x_t)) + \sum_k \sum_{t=(k-1)B+1} (v_r^{\pi_k}(x_t) - v_r^{\pi_k}(x_{t+1}))$$

$$+ \sum_k \sum_{t=(k-1)B+1} v_r^{\pi_k}(x_{t+1}) - \mathbb{E}_{x_{t+1}\sim\mathbb{P}(\cdot|x_t,a_t)}(v_r^{\pi_k}(x_{t+1}))$$

$$= \sum_k \sum_{t=(k-1)B+1} (q_r^{\pi_k}(x_t, a_t) - v_r^{\pi_k}(x_t)) + \sum_{k=2} (v_r^{\pi_k}(x_1^k) - v_r^{\pi_{k-1}}(x_1^k)) + v_r^{\pi_1}(x_1^1) - v_r^{\pi_K}(x_{KB}^K)$$

$$+ \sum_k \sum_{t=(k-1)B+1} v_r^{\pi_k}(x_{t+1}) - \mathbb{E}_{x_{t+1}\sim\mathbb{P}(\cdot|x_t,a_t)}(v_r^{\pi_k}(x_{t+1}))$$

Note that $|v^{\pi_1}(x_1^1)| \leq \mathcal{O}(t_{mix})$, and $|v^{\pi_K}(x_{KB}^K)| \leq \mathcal{O}(t_{mix})$, thus, we have

$$\sum_k \sum_{t=(k-1)B+1}^{kB} (J_r^{\pi_k} - r(x_t, a_t))$$

$$= \sum_k \sum_{t=(k-1)B+1} (q_r^{\pi_k}(x_t, a_t) - v_r^{\pi_k}(x_t)) + \sum_{k=2} (v_r^{\pi_k}(x_1^k) - v_r^{\pi_{k-1}}(x_1^k)) + \mathcal{O}(t_{mix})$$

$$+ \sum_k \sum_{t=(k-1)B+1} v_r^{\pi_k}(x_{t+1}) - \mathbb{E}_{x_{t+1}\sim\mathbb{P}(\cdot|x_t,a_t)}(v_r^{\pi_k}(x_{t+1})) \tag{139}$$

The first term is a Martingale sequence. Hence, with prob. $1 - \delta$, we have

$$\sum_k \sum_{t=(k-1)B+1} (q_r^{\pi_k}(x_t, a_t) - v_r^{\pi_k}(x_t)) \leq 6t_{mix}\sqrt{2T \log(2/\delta)} \tag{140}$$

For the second term, since $|\pi_k(a|x) - \pi_{k-1}(a|x)| \leq \mathcal{O}(\alpha((1 + \xi)N/\sigma)\pi_{k-1}(a|x))$ (by Lemma 32), then, by Lemma 7 in Wei et al. (2020), we have

$$v_r^{\pi_k}(x) - v_r^{\pi_{k-1}}(x) = \mathcal{O}(\alpha(1 + \xi)N^3/\sigma) \tag{141}$$

The third term is again a Martingale sequence. Hence, we have with probability $1 - \delta$,

$$\sum_k \sum_{t=(k-1)B+1} v_r^{\pi_k}(x_{t+1}) - \mathbb{E}_{x_{t+1} \sim \mathbb{P}(\cdot|x_t,a_t)}(v_r^{\pi_k}(x_{t+1})) \leq 6t_{mix}\sqrt{2T\log(2/\delta)}$$

Hence, we have with probability $1 - 2\delta$,

$$\sum_k \sum_{t=(k-1)B+1}^{kB} (J_r^{\pi_k} - r(x_t, a_t)) \leq (T/B)\mathcal{O}(\alpha(1 + \xi)N^3/\sigma) + 12t_{mix}\sqrt{2T\log(2/\delta)} \tag{142}$$

Now, using the value of $\alpha$, we obtain with probability $1 - 2\delta$,

$$\sum_{t=(k-1)B+1}^{kB} (J_r^{\pi_k} - r(x_t, a_t)) \leq \tilde{\mathcal{O}}(\sqrt{Tt_{mix}^3}) \tag{143}$$

$\square$

### H.3   Constraint Violation Bound

From (120), and the value of $\alpha$, $N$, and $B$, we have with probability $1 - 2\delta$

$$\sum_k B(J_r^* - J_r^{\pi_k}) + YB(b - J_g^{\pi_k}) \leq \tilde{\mathcal{O}}((1 + \xi)/\sigma\sqrt{Tt_{mix}^3}) \tag{144}$$

Now, from Corollary 4

$$\sum_k B(b - J_g^{\pi_k}) \leq \tilde{\mathcal{O}}(\frac{2(1 + \xi)}{\xi\sigma}\sqrt{Tt_{mix}^3}) \tag{145}$$

On the other hand similar to Lemma 26, we can also show that

**Corollary 2.** *With prob.* $1 - 2\delta$,

$$\sum_{t=(k-1)B+1}^{kB} (J_g^{\pi_k} - g(x_t, a_t)) \leq \tilde{\mathcal{O}}(\sqrt{Tt_{mix}^3}) \tag{146}$$

Hence, we obtain with probability $1 - 4\delta$,

$$\text{Violation}(T) \leq \tilde{\mathcal{O}}(\frac{3(1 + \xi)}{\xi\sigma}\sqrt{Tt_{mix}^3}) \tag{147}$$

### H.4   Supporting Results

Similar to Lemma 18, we can show the following

**Corollary 3.** *For any* $Y \in [0, \xi]$

$$\sum_k (Y - Y_k)(b - \hat{J}^k) \leq \frac{Y^2}{2\eta} + \frac{\eta K}{2} \tag{148}$$

**Lemma 30.**

$$\sum_{k=1}^K \langle \pi^* - \pi_k, q_r^k + Y_k q_g^k \rangle \leq K\mathcal{O}(\alpha(1 + \xi)^2 N^2/\sigma^2) + \frac{1}{\alpha}\log(|\mathcal{A}|) \tag{149}$$

*Proof.* Now, from the composition of the algorithm (12), we obtain

$$\pi_k = \arg\max_\pi \langle \pi, q_r^{k-1} + Y_{k-1}q_g^{k-1}\rangle - \frac{1}{\alpha}D(\pi|\pi_{k-1}) \tag{150}$$

From Lemma 3.3 in Cai et al. (2020),

$$\langle \pi^* - \pi_{k-1}, q_r^{k-1} + Y_{k-1}q_g^{k-1}\rangle \le \frac{1}{2}\alpha||q_r^{k-1} + Y_k q_g^{k-1}||_\infty^2 + \frac{1}{\alpha}(D(\pi^*|\pi_{k-1}) - D(\pi^*|\pi_k)) \tag{151}$$

Now, from Lemma 31, $||q_r^{k-1} + Y_k q_g^{k-1}||_\infty^2 \le \mathcal{O}((1+\xi)^2 N^2/\sigma^2)$

Now, by summing from $k = 2$ to $K + 1$, and shifting $k - 1$ to $k$, we obtain from (151),

$$\sum_{k=1}^K \langle \pi^* - \pi_k, q_r^k + Y_k q_g^k\rangle \le K\mathcal{O}(\alpha(1+\xi)^2 N^2/\sigma^2) + \frac{1}{\alpha}(D(\pi^*|\pi_1) - D(\pi^*|\pi_{K+1})) \tag{152}$$

By Pinsker's inequality

$$-D(\pi^*|\pi_{K+1}) \le -||\pi^* - \pi_{K+1}||_1^2 \le 0 \tag{153}$$

On the other hand, since $\pi_1$ is uniformly random, thus,

$$D(\pi^*|\pi_1) \le \log(|\mathcal{A}|) \tag{154}$$

Hence, from (152),

$$\sum_{k=1}^K \langle \pi^* - \pi_k, q_r^k + Y_k q_g^k\rangle \le K\mathcal{O}(\alpha(1+\xi)^2 N^2/\sigma^2) + \frac{\log(|\mathcal{A}|)}{\alpha}$$

□

**Lemma 31.** *For any x, and a*

$$|\phi(x,a)^T(w_r^{k-1} + Y_{k-1}w_g^k)| \le \mathcal{O}((1+\xi)N/\sigma) \tag{155}$$

*Proof.*

$$\phi(x,a)^T(w_r^{k-1} + Y_{k-1}w_g^k) \le ||\phi(x,a)|| ||w_r^{k-1} + Y_{k-1}w_g^k|| \tag{156}$$

Note that from the construction of the parameter,

$$||w_\diamond^{k-1}|| \le \frac{24N}{B\sigma}\frac{B}{2N}\sqrt{d}N \tag{157}$$

Hence, we have

$$\phi(x,a)^T(w_r^{k-1} + Y_{k-1}w_g^k) \le \mathcal{O}((1+\xi)N/\sigma) \tag{158}$$

□

**Lemma 32.** *For any a, x*

$$|\pi_k(a|x) - \pi_{k-1}(a|x)| \le \pi_{k-1}(a|x)\mathcal{O}(\frac{\alpha N}{\sigma}(1+\xi)) \tag{159}$$

*Proof.*

$$\pi_k(a|x) - \pi_{k-1}(a|x) = \frac{\pi_{k-1}(a|x)\exp(\alpha\phi(x,a)^T(w_r^{k-1} + Y_{k-1}w_g^{k-1}))}{\sum_b \pi_{k-1}(b|x)\exp(\alpha\phi(x,b)^T(w_r^{k-1} + Y_{k-1}w_g^{k-1}))} - \pi_{k-1}(a|x)$$

$$\le \frac{\pi_{k-1}(a|x)\exp(\alpha\phi(x,a)^T(w_r^{k-1} + Y_{k-1}w_g^{k-1})}{\sum_b \pi_{k-1}(b|x)}\exp(-\alpha\min_b \phi(x,b)^T(w_r^{k-1} + Y_{k-1}w_g^{k-1})) - \pi_{k-1}(a|x)$$

$$\le \pi_{k-1}(a|x)(\exp(\max_b 2\alpha\phi(x,b)^T(w_r^{k-1} + Y_{k-1}w_g^{k-1})) - 1)$$

Note that $\alpha \max_b |\phi(x,b)^T(w_r^{k-1} + Y_{k-1}w_g^{k-1})| \leq 1$ as long as $\alpha \leq \sigma/(24(1+\xi)N)$. Combining with the fact that $\exp(2x) \leq 1 + 8x$ for $x \in [0,1]$.

$$\exp(\max_b 2\alpha\phi(x,b)^T(w_r^{k-1} + Y_{k-1}w_g^{k-1})) - 1 \leq 8\alpha \max_b |\phi(x,b)^T(w_r^{k-1} + Y_{k-1}w_g^{k-1})|$$

$$= \mathcal{O}(\alpha\frac{(1+\xi)N}{\sigma})$$

Hence, the result follows. $\qquad\square$

**Lemma 33.** *For any $k$,*

$$|\mathbb{E}_k[\hat{J}^k] - J_g^{\pi_k}| \leq \frac{1}{T^7} \tag{160}$$

*Proof.* Let $\mathbb{P}^{\pi_k}(x_t, x_1)$ denote the probability that the learner reaches the state $x_t^k$ at time $t$ within epoch $k$ when the epoch starts with the state $x_1^k$. Hence,

$$||\mathbb{P}^{\pi_k}(x_t, x_1)||_{TV} = ||\mathbb{P}^{\pi_k}\delta_{x_1}(x)||_{TV}^{t-1} \tag{161}$$

where $\delta_{x_1}(x)$ is 1 when $x = x_1$, and 0 otherwise.

$$|\mathbb{E}_k[\hat{J}^k] - J_g^{\pi^k}| =$$

$$|\frac{1}{B-N}\sum_{t=N+1}^{B}(\int_{\mathcal{X}}\sum_a \pi_k(a_t|x_t)g(x_t, a_t)d\mathbb{P}^{\pi_k}(x_t, x_1) - \int_{\mathcal{X}}\sum_a \pi_k(x_t|a_t)g(x_t, a_t)d\nu^{\pi_k}(x)|$$

$$= |\frac{1}{B-N}\sum_{t=N+1}^{B} 2||\mathbb{P}^{\pi_k}(x_t^k, x_1^k) - \mathbb{P}^{\pi_k}\nu^{\pi_k}||_{TV}^{t-1}|$$

$$\leq |\frac{1}{B-N}\sum_{t=N+1}^{B} 2e^{-N/t_{mix}}|$$

$$\leq \frac{2}{T^7} \tag{162}$$

where the second last inequality follows from Assumption 4. The last inequality follows from the definition of $N$. $\qquad\square$

Similar to Corollary 1, we can show the following

**Corollary 4.** *For $C \geq 2Y^*$,*

$$\sum_{k=1}^{K}(J_r^{\pi^*} - J_r^{\pi_k}) + C(b - J_g^{\pi_k}) \leq \delta \tag{163}$$

*then*

$$\sum_k(b - J_g^{\pi_k}) \leq \frac{2\delta}{C} \tag{164}$$

Finally, we state some results which are proved in Wei et al. (2021a) (Lemma 6 and 14, respectively).

**Lemma 34.** *For any stationary policy $\pi$, and $x$ $|v^\pi(x)| \leq 4t_{mix}$, and $|q^\pi(x,a)| \leq 6t_{mix}$ for any $(x,a)$.*

**Lemma 35.**

$$q^\pi(x,a) = (w^\pi + J^\pi e_1)^T\phi(x,a) \tag{165}$$

## H.5 ZERO CONSTRAINT VIOLATION

We show that by considering a $\epsilon$-tighter optimization problem, and by carefully choosing $\epsilon$, one can obtain zero constraint violation while maintaining the same order on the regret.

First, we introduce some notations. Consider the $\epsilon$-tighter optimization problem

$$\text{maximize } J_r^\pi \quad \text{subject to } J_g^\pi \geq b + \epsilon \tag{166}$$

where in the constraint $b + \epsilon$ replaces $b$. If $\epsilon \leq \gamma/2$, then Slater's condition is satisfied as $\bar{\pi}$ would still be strictly feasible for the $\epsilon$-tighter CMDP (Assumption 3). The optimal dual variable is bounded by $Y^* \leq 2/\gamma$ for this tighter optimization problem. Since we need $\xi \geq 2Y^*$, we make $\xi = 4/\gamma$. We denote the optimal policy of the $\epsilon$-tighter optimization problem as $\pi^{\epsilon,*}$. We denote $J^{\pi^{\epsilon,*}}$ as the optimal constant gain of the tighter optimization problem (166).

Now, we are ready to state the result for the set of Assumptions in Section 4.

**Lemma 36.** *In Algorithm 3 replace $b$ with $b + \epsilon$, and $\xi$ with $4/\gamma$, then under Assumptions 3,2, 4, and 5, with probability $1 - 4\delta$*

$$\text{Regret}(T) \leq \tilde{\mathcal{O}}((1+\xi)/\sigma\sqrt{Tt_{mix}^3}) + \frac{T\epsilon}{\gamma}$$

$$\text{Violation}(T) \leq \tilde{\mathcal{O}}(\frac{2(1+\xi)}{\sigma\xi}\sqrt{Tt_{mix}^3}) - T\epsilon. \tag{167}$$

*where $\epsilon = \min\{C_6(\frac{2(1+\xi)}{\sigma\xi}\frac{\sqrt{Tt_{mix}^3}\log(dT)^2/\delta}{T}), \gamma/2\}$ for some absolute constant $C_6$.*

Note that when $\tilde{\mathcal{O}}(\frac{2(1+\xi)}{\sigma\xi}\frac{\sqrt{Tt_{mix}^3}}{T}) \leq \gamma/2$, then violation becomes $0$. Hence, for some constant $C_6$, we obtain violation $0$ when $C_6(\frac{2(1+\xi)}{\sigma\xi}\frac{\sqrt{Tt_{mix}^3}\log(dT)^2/\delta}{T}) \leq \gamma/2$. Also note that by plugging in the value of $\epsilon$, the upper bound on regret is still $\tilde{\mathcal{O}}(\sqrt{T})$. Hence, even when for large enough $T$ (still finite), the violation is $0$, while the regret bound is still $\tilde{\mathcal{O}}(\sqrt{T})$.

*Proof*: First, we prove the upper bound on regret. The regret can be decomposed as the following -

$$\text{Regret}(T) = \sum_{t=1}^{T}(J_r^{\pi^*} - J_r^{\pi^{\epsilon,*}}) + \sum_{t=1}^{T}(J_r^{\pi^{\epsilon,*}} - r(x_t, a_t)) \tag{168}$$

The first term can be bounded with the help of the following lemma (the proof for finite state is in Wei et al. (2022) and for episodic setup is in Wei et al. (2021b), we provide the proof at the end of this section)–

**Lemma 37.** *If each stationary policy $\pi$ induces a stationary state-action distribution $\nu^\pi(x, a)$, every stationary policy satisfies (4), and $\pi^{\epsilon,*}$ is the optimal solution of (166), then*

$$J_r^{\pi^*} - J_r^{\pi^{\epsilon,*}} \leq \frac{\epsilon}{\gamma}. \tag{169}$$

Under uniform mixture assumption (Assumption 4, the conditions of the Lemma can be verified.

Since the tighter optimization problem is also CMDP, we note that the second term in the right hand side of (168) is essentially the regret of the tighter CMDP.

Hence, from Theorem 3 and Lemma 37 we obtain the expression of the regret bound in Lemma 36.

**Constraint Violation**: Again applying Theorem 3 to the tighter optimization problem (166), we obtain

$$\sum_{t=1}^{T}(b + \epsilon - g(x_t, a_t)) \leq \tilde{\mathcal{O}}(\frac{2(1+\xi)}{\sigma\xi}\sqrt{Tt_{mix}^3})$$

Thus,

$$\sum_{t=1}^{T}(b - g(x_t, a_t)) \leq \tilde{\mathcal{O}}(\frac{2(1+\xi)}{\sigma\xi}\sqrt{Tt_{mix}^3}) - T\epsilon$$

Thus, the result follows. □

Now, we show that zero violation can be attained under the set of Assumptions in Theorem 2 and one additional assumption:

**Assumption 6.** *Every stationary policy $\pi$ satisfies (4), and induces a state-action stationary distribution $\nu^\pi(x, a)$.*

For uniform mixture assumption (Assumption 4), one can show that Assumption 6 holds. Note that Wei et al. (2022) which studies the infinite horizon average reward CMDP for the finite state space setup assumes the above assumption inherently in their work. We are now ready to state and prove the main result.

**Lemma 38.** *Replace $b$ with $b + \epsilon$ in Algorithm 2, $\kappa = \max\{\text{sp}(v_r^{\bar{\pi}}, \text{sp}(v_r^{\pi^{\epsilon,*}})\}$, and $\xi = 4/\gamma$, then, under Assumptions 3,2, 1, and 6,*

$$\text{Regret}(T) \leq \tilde{\mathcal{O}}((1 + \text{sp}(v_r^*)(dT)^{3/4}) + T\frac{\epsilon}{\gamma}$$

$$\text{Violation}(T) \leq \tilde{\mathcal{O}}((1 + \kappa)(dT)^{3/4}) - T\epsilon \tag{170}$$

*where $\epsilon = \min\{C_7\frac{((1+\kappa)(dT)^{3/4})\iota}{T}, \gamma/2\}$ for absolute constant $C_7$, and $\iota = \log(2\log(|\mathcal{A}|)dT/\delta)$.*

Note that when $\frac{\tilde{\mathcal{O}}((1+\kappa)(dT)^{3/4})}{T} \leq \gamma/2$, the violation becomes 0. Hence, there exists some constant $C_7$ such that when $C_7\frac{((1+\kappa)(dT)^{3/4})\iota}{T} \leq \gamma/2$, then the violation becomes 0. Hence, for large enough $T$ (Still finite), the violation indeed becomes 0. On the other hand, by plugging in the value of $\epsilon$, we note that the regret bound is still $\tilde{\mathcal{O}}((dT)^{3/4})$ (in fact it is $\tilde{\mathcal{O}}((1 + \kappa + \text{sp}(v_r^*))(dT)^{3/4})$.

*Proof.* First, we prove the upper bound on regret. The regret can be decomposed as the following -

$$\text{Regret}(K) = \sum_{t=1}^{T}(J_r^{\pi^*} - J_r^{\pi^{\epsilon,*}}) + \sum_{t=1}^{T}(J_r^{\pi^{\epsilon,*}} - r(x_t, a_t)) \tag{171}$$

The first term is bounded from Lemma 37 since we assume that every stationary policy satisfies (4). Hence,

$$J_r^{\pi^*} - J_r^{\pi^{\epsilon,*}} \leq \frac{\epsilon}{\gamma}. \tag{172}$$

Since the tighter optimization problem is also CMDP, we note that the second term in the right hand side of (171) is essentially the regret of the tighter CMDP.

Hence, from Theorem 2 and Lemma 37 we obtain the expression of the regret bound in Lemma 38.

**Constraint Violation**: Again applying Theorem 2 to the tighter optimization problem (166), we obtain

$$\sum_{t=1}^{T}(b + \epsilon - g(x_t, a_t)) \leq \tilde{\mathcal{O}}((1 + \kappa)(dT)^{3/4})$$

Thus,

$$\sum_{t=1}^{T}(b - g(x_t, a_t)) \leq \tilde{\mathcal{O}}((1 + \kappa)(dT)^{3/4}) - T\epsilon$$

□

We can also obtain that zero violation for Theorem 1 in a similar way while maintaining same order of regret by picking $\epsilon = \mathcal{O}(1/\sqrt{T})$ in the tighter CMDP.

Here, we show the proof of Lemma 37.

*Proof.* Let $\nu^\pi(x, a)$ be the stationary state-action occupation distribution corresponding to the stationary policy $\pi$. According to Assumption, for every policy induces a stationary distribution. Hence, $J_r^\pi = \int_{x,a} r(x, a) d\nu^\pi(x, a)$, and $J_g^\pi = \int_{x,a} g(x, a) d\nu^\pi(x, a)$.

Consider the state-action distribution $\nu^\epsilon(\cdot, \cdot) = (1 - \epsilon/\gamma)\nu^{\pi^*}(\cdot, \cdot) + \epsilon/\gamma \nu^{\bar{\pi}}(\cdot, \cdot)$. There exists a policy $\tilde{\pi}$ which induces the state-action distribution $\nu^\epsilon$ Altman (1999). Now,

$$\int_{x,a} g(x, a) d\nu^\epsilon(x, a) = (1 - \epsilon/\gamma)\int_{x,a} g(x, a) d\nu^{\pi^*}(x, a) + \epsilon/\gamma \int_{x,a} g(x, a) d\nu^{\bar{\pi}}(x, a)$$
$$\geq (1 - \epsilon/\gamma)b + \epsilon/\gamma(b + \gamma) = b + \epsilon \tag{173}$$

Thus, such a state-action occupation measure is feasible for the tighter optimization problem (166).

Further,

$$\int_{x,a} r(x, a) d\nu^\epsilon(x, a) = (1 - \epsilon/\gamma)\int_{x,a} r(x, a) d\nu^{\pi^*}(x, a) + \epsilon/\gamma \int_{x,a} r(x, a) d\nu^{\bar{\pi}}(x, a)$$
$$\geq (1 - \epsilon/\gamma)J_r^* \tag{174}$$

Since the state-action measure $\nu^\epsilon(x, a)$ is feasible, then $J_r^{\pi^{\epsilon,*}} \geq \int_{x,a} r(x, a) d\nu^\epsilon(x, a)$. Hence, from (174), we have

$$J_r^{\pi^{\epsilon,*}} - J_r^* \geq -\epsilon/\gamma$$

Hence, the result follows. □

# I  EXPERIMENTS

We evaluate Algorithm 3 on a similar model described in Chen et al. (2022); Singh et al. (2020)– a wireless node is continuously transmitting packets. The node consists of a queue where the maximum capacity is 9. At time $t$, the node needs to choose a transmission power $a_t \in \{0.1, 0.9\}$ as action. Higher transmission power results in higher probability of successful transmission. The number of packets arriving at the node at time $t$ is given by $Y_t$. We assume $Y_t \in \{0, 1, 2, 3\}$, and the corresponding probabilities are $(0.65, 0.2, 0.1, 0.05)$. The channel reliability is $0.9$, i.e., each attempted transmission has success probability of $0.9$. The dynamics of the queue is the following $Q_{t+1} = \min\{9, \max\{Q_t + Y_t - D_t, 0\}\}$ where $D_t$ is 1 with probability $a_t p_r$, and 0 otherwise. The goal is to maintain a short queue length with small transmission power. At time $t$, the node gets a reward $1 - a_t$. Reward decreases as $a_t$ increases. On the other hand, it gets a utility if the state of the queue is low. Specially, the utility is $g(Q_t) = 1 - 0.1Q_t$. We seek to ensure that the average utility is kept at least $0.7$, i.e., the goal is to keep average queue length less than or equal to 3. Compared to Chen et al. (2022), we use a different utility function. First, we ensure that the utility is bounded between 0 and 1. Further, Chen et al. (2022) considers the constraint as a cost function where we consider it as utility function. Hence, we need to provide higher utility for lower state and vice versa unlike in Chen et al. (2022). Further, our algorithm is model-free unlike in Chen et al. (2022).

Note that the setup can be represented in a tabular form. Since linear MDP contains tabular form, the feature space representation becomes simple, in particular, $\phi(s, a) = \mathbf{e}_{s,a}$ where $\mathbf{e}_{s,a}$ is 1 for the state-action pair $(s, a)$, and 0 otherwise. The dimension of the feature space is $|\mathcal{S}||\mathcal{A}|$.

We run Algorithm 3 for $5 \times 10^5$ time steps. The parameter we used are the followings: $B = 200$, $N = 10$, $\alpha = 1/\sqrt{T}$, $\eta = 10/\sqrt{T/B}$. Note that we do not know $t_{mix}$ or $\sigma$, yet, we achieve sub-linear regret and a violation which approaches zero. We have used $B\sigma/(24N) = 0.05$ which we obtain by tuning. Hence, our algorithm can work well even if we do not know the MDP dependent parameters.

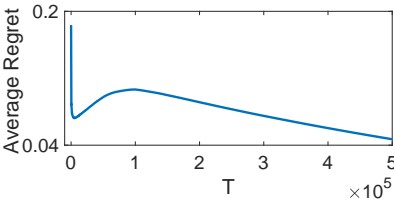 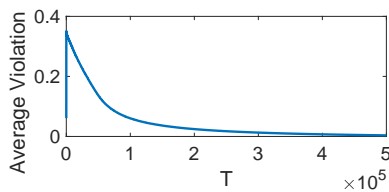

Figure 1: The plot for average regret (Regret divided by no. of steps) and average constraint violation (Violation$(T)/T$) as a function of $T$ (x-axis) for $\epsilon = 0.01$. The x-axis is in the order of $10^5$. Each plot is an average of $5$ trials.

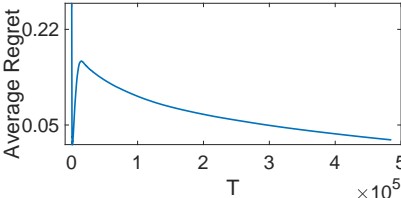 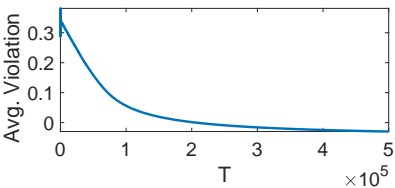

Figure 2: The plot for average regret (Regret$(T)/T$) and average constraint violation (Violation$(T)/T$) as a function of $T$ (x-axis) for $\epsilon = 0.1$. The x-axis is in the order of $10^5$. Each plot is an average of $5$ trials.

In Fig. 1 we plot the average regret and average violation as a function of $T$ for $\epsilon = 0.01$ (the tightness for constrained introduced in Remark 1). As predicted by our theory, our algorithm achieves sub-linear regret and sub-linear violation as both the average regret and average violation decrease to $0$ as $T$ increases. Initially, the regret decreases fast since we start with an infeasible policy (violation increases, Fig. 3). Subsequently, the regret increases as our algorithm seeks to reduce the violation by making the dual variable high (the policy would be sub-optimal). The average violation decreases rapidly (Fig. 2). Eventually, the regret again decreases when $T \approx 1 \times 10^5$ as our algorithm finds the dual variable which balances the regret and violation. In this regime, the average violation slowly approaches $0$ as $T$ grows.

In Fig. 2 we plot the average regret and average violation as a function of $T$ for $\epsilon = 0.1$. Again it is apparent that the regret and violation grow sub-linearly as the average regret and average violation decrease to $0$. Since $\epsilon$ is high, from Fig. 3 we observe that the violation indeed eventually becomes $0$. From Fig. 3, we observe that the violation starts decreasing after $T \approx 0.5 \times 10^5$ as opposed to $T \approx 10^5$ for smaller value of $\epsilon$. The variation of regret is similar to Fig. 1 as the regret first decreases rapidly and then increases till $T \approx 0.5 \times 10^5$. The regret eventually decreases steadily as $T$ increases further. Note that the regret starts to decrease much earlier (at $T \approx 0.5 \times 10^5$) in this scenario compared to the scenario where $\epsilon$ is smaller. Intuitively, since $\epsilon$ is higher, the dual variable increases at a higher rate which helps in finding the feasible solution at a faster rate. The regret then starts decreasing as once we find the dual variable which balances between the regret and violation.

We would like to point out the difference with Chen et al. (2022). The violation in Chen et al. (2022) oscillates, and never approaches $0$. On the other hand, in our evaluation, we observe that the violation eventually approaches $0$. This is due to the fact that we have a learning rate $\eta$ (which is $\mathcal{O}(1/\sqrt{T})$) in the dual-update step whereas in Chen et al. (2022) the learning rate is $1$, hence, in Chen et al. (2022) the dual variable oscillates more compared to our approach. Further, Chen et al. (2022) only guarantees constant violation, whereas we can achieve zero violation.

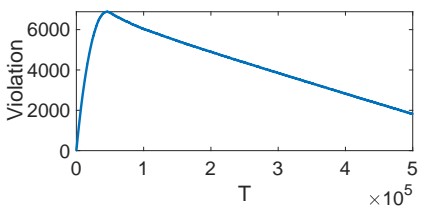
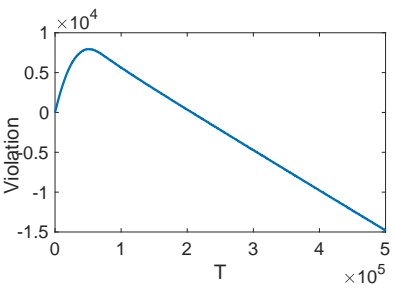

Figure 3: The plot for violation a function of $T$ (x-axis). The left-hand figure is for $\epsilon = 0.01$ and the right-hand figure is for $\epsilon = 0.1$.

