# OpenReview forum: "Achieving Sub-linear Regret in Infinite Horizon Average Reward Constrained MDP with Linear Function Approximation"
_ICLR.cc/2023/Conference — ICLR 2023 poster_

### Official Review · Reviewer_ob3W · 2022-10-21

**Confidence:** 3
**Correctness:** 4
**Technical Novelty And Significance:** 2
**Empirical Novelty And Significance:** 2
**Recommendation:** 6

**Clarity, Quality, Novelty And Reproducibility:**

The paper is written clearly and I believe it is technically correct. As mentioned in the weakness, the technical novelty can be questionable given the two papers Ghosh et al. (2022) and Wei et al. (2021a). The former deals with episodic CMDP; the latter deals with infinite-horizon average-reward linear MDP.

**Strength And Weaknesses:**

Strength:
1. The performance guarantee is significantly better than previous works. Compared with previous works for tabular CMDP, this paper works under weaker assumptions and has better upper bounds for regret and violation.
2. Linear function approximation is a highly noted topic in RL theory community. The constrained MDP problem is a meaningful extension and can be interesting to many.
3. The paper is clearly written and easy to follow.

Weakness:

1. Compared with Ghosh et al. (2022) and Wei et al. (2021a), the technical contribution is not significant. The regret analysis relies on first reducing the infinite-horizon average reward regret to the regret of finite horizons, the same as Wei et al. (2021a). Then, results for episodic CMDP are invoked from Ghosh et al. (2022) to bound the finite-horizon regret. This makes the novelty of Theorem 2 not significant enough. While the difference from Ghosh et al. (2022) is discussed it just remains superficial: the paper claims a $\sqrt{H}$ improvement, which is no surprise because it studies the homogeneous setting.



**Summary Of The Paper:**

This paper studied the Constrained MDP (CMDP) problem, under the infinite-horizon average-reward setup of linear MDP. The model-free algorithm proposed in this paper has better regret bound in terms of total steps $T$ compared with the previous algorithm for tabular CMDP, under weaker assumptions. Under the same assumption as the tabular CMDP algorithm, the authors show the algorithm can achieve smaller regret with no violation of the constraint. The second result in this paper is a policy-based algorithm that works under less realistic assumptions: uniform ergodicity and explorative policies. This algorithm takes a different approach and can achieve regret and violation of order $\sqrt{T}$.

**Summary Of The Review:**

This is a technically clear and rigorous paper. The novelty and significance are slightly decreased due to the existence of two previous works.

---

> ### Author Response · Authors · 2022-11-14
> **Response to Reviewer ob3W (Part 1)**
>
> We appreciate your time and thoughtful evaluation of our paper.  We recap your comment and present our detailed response as follows. We would be happy to provide further clarifications if suitable.
>
> >*Compared with Ghosh et al. (2022) and Wei et al. (2021a), the technical contribution is not significant*
>
> We address this question in two parts. First, let us highlight our overall contributions, and then we will address the question of technical novelties.
>
> ## Overall Contribution
>  We are the first (*model-free or model-based*) to provide **sub-linear regret and violation bound with linear function approximation**. Chen et al.'22 consider the tabular CMDP and provided sublinear regret and violation bound using **model-based approach**, instead we provide **model-free** approach and consider linear function approximation. In this way, *we extend the understanding of the infinite horizon average reward CMDP in a significant manner*. Now, we point out our technical contributions and the significance of our results in detail specifically over Ghosh et al.'22  and Wei et al.'21a.
>
> - **Significance of Algorithm 1**:  Algorithm 1 implies that $\tilde{O}(\sqrt{T})$ regret and violation bound are achievable under only Assumption 1 with linear function approximation setting for infinite horizon average reward CMDP.  *This result significantly advances our understanding of infinite horizon average reward CMDP*. It is well-known that model-free approach can achieve $\tilde{O}(\sqrt{T})$ regret under Assumption 1 for the **unconstrained setup**, however, whether it is possible to achieve $\tilde{O}(\sqrt{T})$ regret using *model-free* approach for **constrained setup was open even for tabular setup**. For example, Wei et al.’22 proposed a model-free algorithm that only achieves $\tilde{O}(T^{⅚})$ regret for tabular case (under a stronger set of Assumption compared to Assumption 1). **We have now answered that question in an affirmative way**. In this regard, we complement the results obtained by Chen et al.'22.  For a finite-state space (tabular), weakly communicating MDP satisfies Assumption 1, Chen et al.'22 showed that for weakly communicating MDP it is possible to achieve $\tilde{O}(\sqrt{T})$ regret and violation bound using *model-based* approach. Since linear CMDP contains tabular case,  we show that it is possible to achieve $\tilde{O}(\sqrt{T})$ regret and violation bound using **model-free** approach as well for weakly communicating CMDP.
>
> - Algorithm 1 is computationally inefficient. However, we observe that Algorithm 4 proposed by Chen et al.'22 which achieves $\tilde{O}(\sqrt{T})$ regret and violation under Assumption 1 is also **computationally inefficient**. Hence, we can conclude that achieving an efficient approach to achieve $\tilde{O}(\sqrt{T})$ regret and violation even for tabular case (model-based or model-free) under only Assumption 1 is **still an open question**.  In fact, Wei et al.'21a also proposed a **computationally inefficient** model-free algorithm that achieves $\tilde{O}(\sqrt{T})$ regret under only Assumption1. Thus, even for unconstrained linear MDP, how to achieve $\tilde{O}(\sqrt{T})$ regret using a **model-free computationally efficient algorithm under only Assumption 1** remains an open question.
>
>  ## Technical Novelty of Algorithm 1
>
> Now, we point out the technical novelty of Algorithm 1. In particular, we detail the novelties compared to Algorithm FOPO (which is also computationally inefficient) proposed in **Wei et al.'21a** for the unconstrained linear MDP.
>
> - First, we have an additional constraint where $J_{g,t}\geq b$ since we consider CMDP which we incorporate in our optimization problem solved in Algorithm 1.
>
> - Second and more importantly,  in the unconstrained case the policy was greedy with respect to $q_r$. Since the greedy policy may not be feasible, we need to search over all policies in the CMDP. **However, it is difficult to obtain sub-linear regret and violation for any arbitrary policy space.** Hence, we have to restrict the policy to smooth policy class $\Pi$ (Definition 1).
>   - The main reason for the introduction of $\Pi$ is that the uniform concentration bound (a key step for proving both regret and violation for model-free approach) for both reward and utility value function classes cannot be achieved unless the policy has smooth properties such as Lipschitz continuity. In particular, we need to show that the log $\epsilon$ covering number for each value function class scales at most $O(\log(T))$.  For the unconstrained setup, the greedy policy with respect to $q$ function was enough. However, this is not the case for the constrained setup as we need to show this result for both the reward and utility value functions. Thus, using the smoothness property of the soft-max, we show that log $\epsilon$-covering number for the class of value function scales at most $O(\log(T))$. Please see page 5 (marked in blue) and Appendix E for details.
>
> [..*contd.*]

---

> > ### Author Response · Authors · 2022-11-14
> > **Response to Reviewer ob3W (Part 2)**
> >
> > - **Significance of Algorithm 2**: In Algorithm 2, we propose an efficient variant of Algorithm 1 and show that $\tilde{O}((dT)^{3/4})$ regret and violation bound can be achieved using an efficient approach under Assumption 1. Even though the regret and violation bounds are worse, they are still better compared to $\tilde{O}(T^{⅚})$ bound achieved in Wei et al.’22 for the tabular case. *In the following, we detail the novelties of Algorithm 2 compared to Ghosh et al.’22 and Wei et al.’21a*.
> >
> > - **Difference between Ghosh et al.'22**: Our approach is to divide the entire time horizon $T$ into multiple episodes ($T/H$) where each episode consists of $H$ steps. We use the algorithm proposed by Ghosh et al.'22 to learn good policies in this episodic setup. We show that the regret and violation bound in the episodic case is $\tilde{O}(\sqrt{TH^2})$ using the analysis of Ghosh et al.'22. Now, for the next step in comparing with the infinite horizon set-up, we need to show that **the gap** between the total reward in the infinite-horizon setup and the value function corresponding to the optimal stationary policy (for the infinite-horizon problem) in this episodic setup can also be upper bounded. **Using Lemma 3**, we have shown that the upper bound is $O(T/H)$. Now, by choosing $H$ on the order of $T^{¼}$, we obtain the final result.  *In summary, we use Ghosh et al.'22 to learn a policy that will have low regret and violation in the episodic case. However, how to bound the gap with respect to the original infinite horizon average reward CMDP is non-trivial which will also be more apparent in the following.*
> >
> > - **Differences compared to Wei et al'21a**: The idea to divide the entire time horizon into multiple episodes and then invoking the algorithm designed for the episodic case is similar to the one proposed by Wei et al.'21a for the unconstrained setup. **However, there is a subtle difference due to the additional constraint in our case**, which we explain next.  In particular, we need to ensure that the optimal solution of the original infinite horizon average reward CMDP is also feasible for the episodic setup, otherwise, the learnt policy might be very far from the optimal one. **Such a requirement does not arise in the unconstrained case**. While the natural intuition would be to consider the following constraint $V_{g,1}^{\pi}(x_1^k)\geq Hb$, *this would not be sufficient as the optimal policy for the infinite horizon average reward may not be feasible.* Instead, we relax the constraint by subtracting a quantity $\kappa$ (independent of $T$) from $Hb$ which ensures that the optimal policy for the infinite-horizon average reward CMDP is feasible. However, this adds an extra $(T/H)\kappa$ term in the violation bound because we relaxed the constraint for the episodic case. Since $H=O(T^{¼})$, we show that the above term only grows as $O(T^{¾})$ (Please see pages 6 and 8, and pages 21 and 23, marked in blue).
> >
> > -  **Significance of Algorithm 3**: Algorithm 3 achieves **zero constraint violation** with $O(\sqrt{T})$ regret under a *stronger set of Assumptions* (similar to the ones for unconstrained setup considered in Wei et al.'21a). **This is the first computationally efficient algorithm that achieves $\tilde{O}(\sqrt{T})$ regret and zero violation using a model-free approach (even for tabular case).**  We propose a primal-dual adaptation of the MDP-EXP2. Extending the MDP-EXP2 (proposed in Wei et al.'21a) to the constrained case is non-trivial. Unlike the unconstrained setup, the decision is taken based on the joint state-action bias function corresponding to reward and utility. Thus, how to bound the regret and violation separately from this joint decision process is not trivial. Second, we provide a high-probability bound unlike the expectation bound in Wei et al.'21a, hence, the analysis also differs. In the following, we provide high-level ideas. Please see Appendix G.1 (page 34) and Appendix G.1.1. (page 36) for details.
> >
> >    - **Technical Novelty**: In the unconstrained setup, the key step is to bound the gap $\sum_k (J^*-J^{\pi_k})$. Instead, we seek to bound $\sum_{k}(J_r^*-J_r^{\pi_k})+Y(b-J_g^{\pi_k})$ which we show that upper bounded by $\sum_{k}(J^*_r+Y_kJ_g^*-J_r^{\pi_k}-Y_kJ_g^{\pi_k})+(Y-Y_k)(\hat{J}^k-J_g^{\pi_k})+(Y-Y_k)(b-\hat{J}_g^k)$. The main technical novelty comes from bounding each of the above terms. In particular, one can not rely on the results of MDP-EXP2 to bound the above terms. For example, for the first term,  since the dual variable $Y_k$ changes over time, we cannot use the results of MDP-EXP2 which bounds $\sum_k (J^*-J^{\pi_k})$. Instead, we use the results from the OMD analysis (one-step descent lemma) and how we obtain $w_r^k, w_g^k$ to bound the first term with high probability. For the second term, we design $\hat{J}^k$ in a way such that it would be close to an unbiased estimator for the average utility. Finally, to bound the third term, we rely on dual analysis.

---

> > > ### Author Response · Authors · 2022-11-22
> > > **Follow Up**
> > >
> > > Dear Reviewer ob3W,
> > >
> > > We just wanted to check in and ask if the rebuttal clarified and answered the questions raised in your review. We would be very happy to engage further if there are additional questions!
> > >
> > > Also, if you feel that we have answered your concerns, please consider increasing the score.

---

> > > > ### Author Response · Authors · 2022-12-02
> > > > **Clarification with respect to $\sqrt{H}$ improvement**
> > > >
> > > > We unintentionally overlook your comment on $\sqrt{H}$ improvement over Ghosh et al.'22. We respond to it in detail here. We would be happy to engage with you further if you have any more questions.
> > > >
> > > > >*While the difference from Ghosh et al. (2022) is discussed it just remains superficial: the paper claims a  $\sqrt{H}$ improvement, which is no surprise because it studies the homogeneous setting*
> > > >
> > > > Note that **we do not claim that**  $\sqrt{H}$ improvement is our main contribution. Instead, in our paper, we clarified that we obtain  $\sqrt{H}$ improvement over Ghosh et al.'22 since the infinite-horizon setup is a homogenous setup (please see page 8, paragraph after Regret (T) expression or paragraph just before the one marked in blue) so that it won't confuse the readers.
> > > >
> > > > Please see our earlier responses to you (both Part 1 and part 2) for our main contributions.

---

> > > > > ### Author Response · Authors · 2022-12-09
> > > > > **Follow up after clarifying the doubt regarding $\sqrt{H}$ improvement**
> > > > >
> > > > > Dear Reviewer ob3W,
> > > > >
> > > > > We just wanted to check in and ask if our clarification on $\sqrt{H}$ improvement has addressed your concern! We will be very happy to engage with you if you have any more concerns/doubts.
> > > > >
> > > > > Also, if you feel that we have answered your concerns, please consider increasing the score.
> > > > >
> > > > > Authors

---

### Official Review · Reviewer_3YVd · 2022-10-24

**Confidence:** 2
**Correctness:** 3
**Technical Novelty And Significance:** 3
**Empirical Novelty And Significance:** 3
**Recommendation:** 6

**Clarity, Quality, Novelty And Reproducibility:**

All statements are clear and well-supported. The whole paper is well-written. Main results are not very interesting but are the first time to be proposed and proved.

**Strength And Weaknesses:**

Strength.

This paper is sufficiently complete. It proposes new algorithms, builds a theoretically-improved convergence analysis on both regret and violation bounds, and empirically verifies the result on stochastic environments. And this work is purely model-free; it doesn't require any estimation of transition kernel.

Weaknesses.

I didn't find any major weaknesses.

**Summary Of The Paper:**

This work proposes two approach, optimism-based algorithm and policy-based algorithm for infinite-horizon average reward constrained linear MDP under different assumptions and achieve better convergence upper bounds.

**Summary Of The Review:**

I recommend to accept this paper since the author has presented a complete result, including new algorithm designs, theoretical analysis, and empirical studies. And these results are new.

---

> ### Author Response · Authors · 2022-11-14
> **Response to Reviewer 3YVd**
>
> We appreciate your time and thoughtful evaluation of our paper. We are glad that you have liked our paper and find that the paper is well-written and the statements are clear and well-supported. We would also be happy to provide further clarifications if you have any.

---

> > ### Author Response · Authors · 2022-11-29
> > **Regarding Lowering the correctness score**
> >
> > Dear Reviewer 3YVd,
> >
> > We noticed that you decreased the correctness score from 4 to 3 without any explanation. All the other reviewers maintained a score of 4. We believe that our result is correct. Can you please point out why do you feel that our results are not correct? Can you please give us a chance to convince you that the results are correct?

---

> > > ### Comment · Reviewer_3YVd · 2022-11-30
> > > **Comments**
> > >
> > > I am pasting my comments during the discussion below. In summary, I would agree with Reviwer ob3W's opinion that the complexity improvement is not as significant as it is expected since homogeneous setting is an easier case and the overall complexity improvement will rely on the improvement made in this step.
> > >
> > > ====================================================
> > > Thanks AC for raising the discussion. I have read the author's rebuttal and other reviewers' comments. I do agree with Reviewer 2G2e and Reviewer ob3W: both the algorithm design and the theoretical analysis are not significantly different from existing work.
> > >
> > > My main reason of voting for an accept is that this work has an order-wise improvement compared to existing research under the same setting, but Reviwer ob3W has pointed out that the homogeneous setting is the reason why this paper makes the sqrt{H} improvement, and it seems that the author didn't directly resolve this problem in the rebuttal. I would trust the judgement from Reviwer ob3W and reflect it by reduing my score from 8 to 6 (I still believe this work has high-quality and clear writing).

---

> > > > ### Author Response · Authors · 2022-12-02
> > > > **Response to your follow-up comment**
> > > >
> > > > Thank you for engaging with us and providing us with a chance to respond. Note that **we do not claim** that $\sqrt{H}$ improvement is our main contribution. Instead, in our paper, we clarified that we obtain $\sqrt{H}$ improvement over Ghosh et al.'22 since the infinite-horizon setup is a homogenous setup unlike the episodic setup (please see page 8, paragraph after Regret (T) expression or paragraph just before the one marked in blue) so that it won't confuse the readers.
> > > >
> > > > Our main contributions are the followings:
> > > > 1. We are the first (model-free or model-based) to provide sub-linear regret and violation bound with linear function approximation for infinite horizon CMDP.
> > > > 2. **Since the linear CMDP includes the tabular setup, we also improve the *existing* model-free results for tabular CMDP with respect to $T$**. For example, Wei et al.'22 proposed a model-free algorithm that only achieves $\tilde{\mathcal{O}}(T^{⅚})$ regret for the tabular case. Our algorithm 2 (computationally efficient) achieves $\tilde{\mathcal{O}}(T^{3/4})$ **regret** with zero violation under same assumption. Algorithm 1 (computationally inefficient) achieves $\tilde{\mathcal{O}}(T^{1/2})$ **regret** under only Assumption 1 which is the first such result for CMDP with linear function approximation using the model-free algorithm.
> > > > 3. We also achieve $\tilde{\mathcal{O}}(\sqrt{T})$ regret with zero violation using a computationally efficient algorithm (Algorithm 3) under a stronger set of assumptions.
> > > > 4. We conduct numerical experiments and empirically show that regret and violation indeed grow sub-linearly in a simulated setup (Appendix H).
> > > >
> > > > Please see the responses to Reviewers ob3W and 2G2e for more details. For technical novelties, also see highlighted parts in Section 3.1 (page 5), Section 3.2 (pages 6 and 8), Appendix E, Appendix F.1, and Appendix G.1.
> > > >
> > > > We would be happy to engage with you further if you need any more clarifications.

---

> > > > > ### Author Response · Authors · 2022-12-09
> > > > > **Follow-up**
> > > > >
> > > > > Dear Reviewer 3YVd,
> > > > >
> > > > > We are wondering whether our follow-up answer has addressed your doubt. We will be very happy if you provide us with more opportunities to engage if you have any more concerns/doubts.
> > > > >
> > > > > Also, if you feel that we have answered your concerns, please consider increasing the score.

---

### Official Review · Reviewer_2G2e · 2022-10-25

**Confidence:** 3
**Clarity, Quality, Novelty And Reproducibility:** Please check my comments above.
**Correctness:** 4
**Technical Novelty And Significance:** 2
**Empirical Novelty And Significance:** Not applicable
**Recommendation:** 3

**Strength And Weaknesses:**

Strength:
  1. The problem setup considered in this paper is infinite horizon average reward CMDP with linear function approximation, which is meaningful and interesting.
  2. The theoretical results in both regret and constraint violation have improvements over existing works in the above mentioned specific problem domain. For example, the efficient algorithm 2 has better bounds of \tilt{O}(T)^{3/4 compared with known one.
  3. The paper is well-written and easy to read.

Weakness:
  1. The algorithm 2 and 3 are direct extensions of the existing algorithms in the literature with incremental efforts on having improved results in both regret and constraint violation bounds.
  2. For the algorithm 2, the main difference is to show how to bound the gap between the best finite horizon policy and the infinite horizon case, which is obtained by Lemma 3 and its proof in the paper. The additional effort looks very incremental.


**Summary Of The Paper:**

The paper studies the infinite horizon average reward constrained Markov Decision Process (CMDP) for the model-free linear CMDP setup. The paper proposes 3 different algorithms under different assumptions or computation efficiency. The first algorithm is based on fixed-point optimization with optimism, which is to solve a very complicated optimism under uncertainty problem for every time step, which as the author/s pointed out, is very inefficient, but enjoys the \tilt{O}(\sqrt{T}) bound guarantee for both the regret and constraint violation. The second algorithm is computationally efficient but has worse bound of \tilt{O}(T)^{3/4}, which is an extension of the previous work LSVI-UCB algorithm from the finite-horizon setting to the infinite one with additional analysis on the horizon truncation impact on the regret and constraint violation. The third algorithm, a primal-dual adaptation of previous work's algorithm MDP-EXP2, has stronger assumptions than the first two algorithms, which is both efficient and having \tilt{O}(\sqrt{T}) bound guarantee for both the regret and constraint violation.

**Summary Of The Review:**

As I commented above, I think the paper tries to solve a meaningful problem in the RL area. But all the proposed efficient algorithms are heavily dependent on previous works and the theoretical results are incremental as well.

---

> ### Author Response · Authors · 2022-11-14
> **Response to Reviewer 2G2e Part 1**
>
> We appreciate your time and thoughtful evaluation of our paper. We recap your comment and present our detailed response as follows. We would be happy to provide further clarifications if suitable.
>
> >*The algorithm 2 and 3 are direct extensions of the existing algorithms in the literature with incremental efforts*
>
> We address this question in two parts. First, let us highlight our overall contributions, and then we will address the question of technical novelties of all our Algorithms including Algorithms 2 and 3.
>
> - **Overall Contribution**: We are the first (*model-free or model-based*) to provide **sub-linear regret and violation bound with linear function approximation**. Chen et al.'22 consider the tabular CMDP and provided sublinear regret and violation bound using **model-based approach**, instead we provide **model-free** approach and consider linear function approximation. **In this way, we extend the understanding of the infinite horizon average reward CMDP in a significant manner.** Now, we point out our technical contributions and the significance of our results in detail which will signify the novelties over existing algorithms.
>
> - **Significance of Algorithm 1**:  Algorithm 1 implies that $\tilde{O}(\sqrt{T})$ regret and violation bound are achievable under only Assumption 1 with linear function approximation setting for infinite horizon average reward CMDP.  *This result significantly advances our understanding of infinite horizon average reward CMDP*. It is well-known that model-free approach can achieve $\tilde{O}(\sqrt{T})$ regret under Assumption 1 for the **unconstrained setup**, however, whether it is possible to achieve $\tilde{O}(\sqrt{T})$ regret using *model-free* approach for **constrained setup was open even for tabular setup**. For example, Wei et al.’22 proposed a model-free algorithm that only achieves $\tilde{O}(T^{⅚})$ regret for tabular case (under a stronger set of Assumption compared to Assumption 1). **We have now answered that question in an affirmative way**. In this regard, we complement the results obtained by Chen et al.'22.  For a finite-state space (tabular), weakly communicating MDP satisfies Assumption 1, Chen et al.'22 showed that for weakly communicating MDP it is possible to achieve $\tilde{O}(\sqrt{T})$ regret and violation bound using *model-based* approach. Since linear CMDP contains tabular,  we show that it is possible to achieve $\tilde{O}(\sqrt{T})$ regret and violation bound using **model-free** approach as well for weakly communicating CMDP.
>
> - Algorithm 1 is computationally inefficient. However, we observe that Algorithm 4 proposed by Chen et al.'22 which achieves $\tilde{O}(\sqrt{T})$ regret and violation under Assumption 1 is also **computationally inefficient**. Hence, we can conclude that achieving an efficient approach to achieve $\tilde{O}(\sqrt{T})$ regret and violation even for tabular case (model-based or model-free) under only Assumption 1 is **an open question**.  In fact, Wei et al.'21a also proposed a **computationally inefficient** model-free algorithm that achieves $\tilde{O}(\sqrt{T})$ regret under only Assumption1. Thus, even for unconstrained linear MDP, how to achieve $\tilde{O}(\sqrt{T})$ regret using a **model-free computationally efficient algorithm** under only Assumption 1 remains an open question.
>
> - **Technical Novelty**: Now, we point out the technical novelty of Algorithm 1. In particular, we detail the novelties compared to Algorithm FOPO (which is also computationally inefficient) proposed in Wei et al.'21a for the unconstrained linear MDP. First, we have an additional constraint where $J_{g,t}\geq b$ since we consider CMDP which we incorporate in Algorithm 1. Second and more importantly,  in the unconstrained case the policy was greedy with respect to $q_r$. Since the greedy policy may not be feasible, we need to search over all policies in the CMDP. **However, it is difficult to obtain sub-linear regret and violation for any arbitrary policy space.** Hence, we have to restrict the policy to smooth policy class $\Pi$ (Definition 1, page 5). The main reason for the introduction of $\Pi$ is that the uniform concentration bound (a key step for proving both regret and violation for model-free approach) for both reward and utility value function classes cannot be achieved unless the policy has smooth properties such as Lipschitz continuity. In particular, we need to show that the log $\epsilon$ covering number for each value function class scales at most $O(\log(T))$.  For the unconstrained setup, the greedy policy with respect to $q$ function was enough to show the above. However, this is not the case for the constrained setup as we need to show this result for both the reward and utility value functions. Thus, using the smoothness property of the soft-max, we show that log $\epsilon$-covering number for the class of value function scales at most $O(\log(T))$. (Please see page 5 and Appendix E for details) [..*contd*]

---

> > ### Author Response · Authors · 2022-11-14
> > **Response to Reviewer 2G2e Part 2**
> >
> > - **Significance of Algorithm 2**: In Algorithm 2, we propose an efficient variant of Algorithm 1 and show that $\tilde{O}((dT)^{3/4})$ regret and violation bound can be achieved using an efficient approach under Assumption 1. Even though the regret and violation bounds are worse, they are still better compared to $\tilde{O}(T^{⅚})$ bound achieved in Wei et al.’22 for the tabular case. *In the following, we detail the novelties of Algorithm 2 compared to Ghosh et al.’22 and Wei et al.’21a*.
> >
> > - **Difference between Ghosh et al.'22**: Our approach is to divide the entire time horizon $T$ into multiple episodes ($T/H$) where each episode consists of $H$ steps. We use the algorithm proposed by Ghosh et al.'22 to learn good policies in this episodic setup. We show that the regret and violation bound in the episodic case is $\tilde{O}(\sqrt{TH^2})$ using the analysis of Ghosh et al.'22. Now, for the next step in comparing with the infinite horizon set-up, we need to show that **the gap** between the total reward in the infinite-horizon setup and the value function corresponding to the optimal stationary policy (for the infinite-horizon problem) in this episodic setup can also be upper bounded. Using Lemma 3, we have shown that the upper bound is $O(T/H)$. Now, by choosing $H$ on the order of $T^{¼}$, we obtain the final result.  *In summary, we use Ghosh et al.'22 to learn a policy that will have low regret and violation in the episodic case. However, how to bound the gap with respect to the original infinite horizon average reward CMDP is non-trivial which will also be more apparent in the following.*
> >
> > - **Differences compared to Wei et al'21a**: The idea to divide the entire time horizon into multiple episodes and then invoking the algorithm designed for the episodic case is similar to the one proposed by Wei et al.'21a for the unconstrained setup. **However, there is a subtle difference due to the additional constraint in our case**, which we explain next.  In particular, we need to ensure that the optimal solution of the original infinite horizon average reward CMDP is also feasible for the episodic setup, otherwise, the learnt policy might be very far from the optimal one. Such a requirement does not arise in the unconstrained case. While the natural intuition would be to consider the following constraint $V_{g,1}^{\pi}(x_1^k)\geq Hb$, this would not be sufficient as the optimal policy for the infinite horizon average reward may not be feasible. Instead, we relax the constraint by subtracting a quantity $\kappa$ (independent of $T$) from $Hb$ which ensures that the optimal policy for the infinite-horizon average reward CMDP is feasible. However, this adds an extra $(T/H)\kappa$ term in the violation bound because we relaxed the constraint for the episodic case. Since $H=O(T^{¼})$, we show that the above term only grows as $O(T^{¾})$ (Please see pages 6 and 8, and pages 21 and 23).
> >
> > - Now, we point out the implication of Algorithm 3, which achieves **zero constraint violation** with $O(\sqrt{T})$ regret under a *stronger set of Assumptions* (similar to the ones for unconstrained setup considered in Wei et al.'21a). **This is the first computationally efficient algorithm that achieves $\tilde{O}(\sqrt{T})$ regret and zero violation using a model-free approach.** Algorithm 3 can be implemented efficiently. We propose a primal-dual adaptation of the MDP-EXP2. Extending the MDP-EXP2 (proposed in Wei et al.'21a) to the constrained case is non-trivial. Unlike the unconstrained setup, the decision is taken based on the joint state-action bias function corresponding to reward and utility. Thus, how to bound the regret and violation separately from this joint decision process is not trivial. Second, we provide a high-probability bound unlike the expectation bound in Wei et al.'21a, hence, the analysis also differs. In the following, we provide high-level ideas. Please see Appendix G.1 (page 34) and Appendix G.1.1. (page 36) for details.
> >
> >   - In unconstrained setup, the key step is to bound the gap $\sum_k (J^*-J^{\pi_k})$. Instead, we seek to bound $\sum_{k}(J_r^*-J_r^{\pi_k})+Y(b-J_g^{\pi_k})$ which we show that upper bounded by $\sum_{k}(J^*_r+Y_kJ_g^*-J_r^{\pi_k}-Y_kJ_g^{\pi_k})+(Y-Y_k)(\hat{J}^k-J_g^{\pi_k})+(Y-Y_k)(b-\hat{J}_g^k)$. The main technical novelty comes from bounding each of the above terms. In particular, one can not rely on the results of MDP-EXP2 to bound the above terms. For example, for the first term,  since the dual variable $Y_k$ changes over time, we cannot use the results of MDP-EXP2 which bounds $\sum_k (J^*-J^{\pi_k})$. Instead, we use the results from the OMD analysis (one-step descent lemma) to bound the first term. For the second term, we design $\hat{J}^k$ in a way such that it would be close to an unbiased estimator for $J_g^{\pi_k}$. Finally, to bound the third term, we rely on dual analysis. We bound the constraint in a similar way and using the result from duality.

---

> > > ### Author Response · Authors · 2022-11-22
> > > **Follow Up**
> > >
> > > Dear Reviewer 2G2e,
> > >
> > > We just wanted to check in and ask if the rebuttal clarified and answered the questions raised in your review. We would be very happy to engage further if there are additional questions!
> > >
> > > Also, if you feel that we have answered your concerns, please consider increasing the score.

---

> > > > ### Author Response · Authors · 2022-12-14
> > > > **Follow up**
> > > >
> > > > Dear Reviewer 2G2e,
> > > >
> > > > We are sorry to bother you. However, we wanted to check whether in any way we can address your remaining concerns and convince you to recommend acceptance of our work before the discussion period ends.
> > > >
> > > > We have highlighted our contributions [here](https://openreview.net/forum?id=zZhX4eYNeeh&noteId=XA8J8N1ZqV) and [here](https://openreview.net/forum?id=zZhX4eYNeeh&noteId=FqgujWFzoZ) in addition to our detailed responses to your questions.
> > > > In particular, we are asking you to consider the *statistical significance* of Algorithm 1 and the *technical novelties* in proving the result. Besides, we have highlighted the technical novelties of Algorithms 2 and 3 and why those are not straightforward.
> > > >
> > > > Authors

---

### Official Review · Reviewer_65aT · 2022-11-03

**Confidence:** 3
**Correctness:** 4
**Technical Novelty And Significance:** 3
**Empirical Novelty And Significance:** Not applicable
**Recommendation:** 6

**Clarity, Quality, Novelty And Reproducibility:**

**Clarity:**
 The paper is well-organized and clearly written.

**Quality:**
The paper appears to be technically sound. The proofs appear to be correct, but I have
not carefully check all details.

**Novelty:**
This paper makes non-trivial advances over the current state-of-the-art.

**Reproducibility:**
The code is unavailable, which makes it difficult to reproduce the empirical results. However, sufficient details are given to reproduce the main theoretical results.

**Details Of Ethics Concerns:**

Since this work is a theoretical paper, I do not find any ethical concerns.

**Strength And Weaknesses:**

**Strengths of paper:**
1. Authors first propose an algorithm with $O(\sqrt{d^3T})$ regret and constraint violation upper bounds, which has the tightest regret upper bound, but the algorithm is computationally inefficient. So they propose a computationally efficient algorithm that uses the primal-duel adaption of the LSVI-UCB algorithm but has $O((dT)^{3/4})$ regret constraint violation upper bounds.

2. Under strong assumptions (Assumption 4 and Assumption 5), authors propose policy-based algorithms that are computationally efficient and have $O(\sqrt{T})$ regret constraint violation upper bounds (or even constant constraint violation bound).

3. No simulator (a common assumption) is needed to show the proposed algorithms' bounds.

**Weakness of paper:**
1. The assumptions for policy-based algorithms are too strong, which can limit their practical use.

2. To show any algorithm to be "provably sample-efficient," we need to have a lower bound. I didn't find any discussion or comparison with the lower bound in the main paper.

3. It is unclear when the primal-dual algorithm and fixed episode length (to deal with infinite horizon) are used; then, the learned policy is also optimal. If the learned policy is sub-optimal, it can lead to linear regret and constraint violations.

4. Empirical evaluations are weak. Even regret shown in Figure 1 (left) looks linear.



**Question and other comments.**

Please address the above weaknesses. I have a few more questions:
1. Would you give a few motivating examples (with what reward and utility functions) for your problem setting?
2. What is $\pi_1$ in Algorithm 1?


A minor comment:
1. Page 3, second last paragraph, second line:$j \rightarrow \diamond$.


I am open to changing my score based on the authors' responses.

**Summary Of The Paper:**

This paper studies the model-free linear constrained Markov Decision Process with infinite horizon average rewards. The goal is to learn a policy for selecting actions that minimize the regret (the difference between the maximum achievable average reward and the policy's total average reward) while keeping the total constraint violations as low as possible.

The authors propose four different algorithms (the last two are policy-based algorithms) and show their sub-linear regret and constraint violation upper bounds.

**Summary Of The Review:**

This paper has some overlap with my current work. My recent work was focused on closely related topics and I am knowledgeable about most of the topics covered by the paper.

---

> ### Author Response · Authors · 2022-11-14
> **Response to Reviewer 65aT (Part 1)**
>
> We appreciate your time and thoughtful evaluation of our paper. We recap your comment and present our detailed response as follows. Several of your questions raise challenging open problems that we have thought about, and which we believe might constitute interesting research directions with potentially great practical significance. We would also be happy to provide further clarifications if suitable.
>
> >*The assumptions for policy-based algorithms are too strong, which can limit their practical use*
>
> - First, we would like to point out that Assumptions 4 and 5 are required for analytical purposes only. **Our Algorithm 3 is still applicable when those assumptions are not satisfied**. In our empirical evaluations, we have not used such assumptions nor even the knowledge of $t_{mix}$ and $\sigma$ (in fact, we do not require the knowledge of $t_{mix}$ and $\sigma$ for analysis which we explain later).
>
> - Second, we would like to point out that both Assumptions 4 and 5 are fairly common assumptions in the literature, even for the unconstrained case.
>    - Assumption 4 (uniformly mixing) is assumed to show regret bounds for unconstrained linear MDP [A1] and also for the model-based approach for tabular CMDP [A2] (note that they consider an ergodic MDP, which implies uniform mixing).
>   - Assumption 5 is also a fairly common assumption for the papers which consider function approximation setup. For example, Assumption 5 is considered for the unconstrained MDP [A1] as well. Both Assumptions 4 and 5 are assumed in [A3, A4], which consider function approximation setup for unconstrained MDP.
> - **Regarding knowing $t_{mix}$ and $\sigma$**: We can relax this requirement in both Assumptions 4 and 5 using the idea of [A1] (please see Appendix E there) with a slightly worse regret and violation bound. The idea is to slowly increase epoch length and trajectory length with time, and make sure that they exceed the required amount ($t_{mix}$) in the long run.
> - We could also pursue the following directions to relax Assumption 5. For example, [A5] have proposed a method to relax Assumption 5 where they only require a uniform excitation feature for an exploratory policy $\pi_e$ rather than for all the policies. The idea is to use this policy first at each epoch $k$ for $N$ no. of time steps before gathering samples from the policy $\pi_k$ for the rest of the $N$ steps.  Using a similar idea, we can relax Assumption 5 so that the uniform excitation feature only needs to hold for a *known* exploratory policy rather than any policy.
>
> >*...Regarding Lower Bound...*
>
> Thank you. It is proved that even for tabular case $O(\sqrt{T})$ regret is unimprovable for the unconstrained setup. Hence, this would be the lower bound for the constrained setup as well. We have now explicitly mentioned the above (page 5, last para).
>
> > *..whether the learned policy can lead to linear regret and constraint violations when the primal-dual algorithm and fixed episode length are used*
>
> We want to clarify that although the learned policy in each epoch is sub-optimal (due to truncation),  by carefully designing the length of the episode ($H$) and the constraint of the episodic setup, **we can still show sub-linear regret and sub-linear constraint**. A more detailed explanation is provided below.
>
> - As mentioned earlier, we achieve the sub-linear regret and violation bound by carefully designing the length of the episode ($H$) and the constraint of the episodic setup. *First, we show that if the optimal policy for the infinite horizon average reward CMDP is feasible for the episodic setup with length $H$, then the gap between the value function corresponding to the optimal policy for the episodic CMDP and the original average reward CMDP grows as $O(T/H)$.* Thus, if we make $H$ on the order of $T^{¼}$ (it also increases with $T$), **it will ensure that such an optimality gap would grow as $O(T^{¾})$ only**.
> - Now, we just need to ensure that the optimal policy for the average reward CMDP is feasible for the episodic case. The natural intuition would be to consider the following constraint $V_{g,1}^{\pi}(x_1^k)\geq Hb$. **However, this would not be sufficient as the optimal policy for the average reward infinite horizon CMDP may not be feasible**, i.e, the policy we would learn may be far from the optimal policy for the infinite horizon average reward CMDP. Hence, we need to subtract a quantity $\kappa$ which would ensure that the optimal policy for the infinite horizon average reward CMDP is feasible. *Note that since we are subtracting $\kappa$ we have an additional $(T/H)\kappa$ term.* However, this is not a problem since $H=O(T^{¼})$, the violation bound would grow only as $O(T^{¾})$. The proof outline is found in Appendix F.1 (page-21 and 23) and the full proof details are given in Appendix F.2- Appendix F.10.

---

> > ### Author Response · Authors · 2022-11-14
> > **Response to Reviewer 65aT (Part 2)**
> >
> > >*Empirical evaluations are weak. Even regret shown in Figure 1 looks linear.*
> > - We have now plotted the average regret (regret divided by step) and average violation, **it is clear that both the regret and violations grow sub-linearly as the average regret and average violation decrease to 0** (Figures 1 and 2, Appendix H, page 45).
> > - We have now done two sets of experiments with two different values of $\epsilon$ which show the impact of the tightness of the constraint in achieving zero violation (Remark 1, page 9). Figure 3 (page 46) shows that **violation indeed goes to $0$ for a smaller value of $T$ when $\epsilon$ is larger**. Please see Appendix H (page 44) for details.
> >
> > >*Would you give a few motivating examples (with what reward and utility functions) for your problem setting?*
> > - First, we provide some examples of constrained MDP.
> >  1. Consider an intelligent agent taking an action to optimize the power consumption for a household. It would seek to minimize the overall cost while trying to maintain a minimum level of satisfaction (e.g., maintaining a certain temp., maintaining the charge of the electric vehicles). Here the reward can be cast as the negative of the cost for power consumption, while utility can be modeled as the satisfaction the user gets.
> > 2. As another example, consider a sensor network where sensor nodes sample and send information to a server (fusion center) for processing. However,  these sensor nodes also need to satisfy the energy constraint as they have limited battery capacity. Such a decision process can be modeled as CMDP where (i) the reward depends on the nature of the information and whether the information is successfully received or not and (ii) the cost corresponds to the cost for sampling and transmitting information.
> > - We now provide some examples of CMDP that might run for a long-time where the average reward CMDP would be the ideal candidate for modeling.
> > 1. Consider the sensor network example provided earlier. Here, the sensor node continually takes decisions, and thus, an infinite horizon average reward CMDP is a natural choice.
> > 2. As another example, consider that a server is scheduling jobs to different machines. The scheduler seeks to minimize the job completion time. The server wants to maintain a uniform queue length (i.e, the number of jobs waiting to be processed) across the machines which can only process jobs sequentially. The server is continually taking decisions, thus, it is maximizing the average reward while maintaining an average queue length below a certain threshold. This can be cast as the average reward CMDP.
> > 3. Another example is a controller in an autonomous system that takes a decision in order to maximize the average reward (objective) while trying to maintain system stability. We can model the system as an average reward CMDP where there is a utility of 1 if the system remains in the safe region at every step. The goal of the controller is to maximize the average reward while the system will be in safe states for at least $1-\epsilon$ fraction of times for desired choice of $\epsilon>0$. This can be cast as average reward CMDP where the controller seeks to maximize the average reward while the average utility is at least $1-\epsilon$.
> > - Finally, we argue why we choose the function approximation setup. In many examples, provided above, the state space is continuous or at least very large. For example, the state of the battery for a sensor is continuous. Similarly, the length of the queue can be very large. Function approximation is generally used to approximate the Q-function or policy in such a large state space. We consider linear function approximation in our setup as a first step to handle the large state space.
> >
> > >*Regarding $\pi_1$ in Algorithm 1*
> >
> > We believe that the reviewer meant $\pi_t$. $\pi_t$ is the policy which solves $P_1$ in Algorithm 1 (line 5). Note that, unlike the unconstrained case, the greedy policy may not be optimal for CMDP (Please see Page 5 second para after Definition 1).
> >
> > >*Regarding the typo*
> >
> > Thank you for pointing that out. We have rectified the typo (Page 3).
> >
> > References
> >
> > [A1]. Wei, Chen-Yu, et al. "Learning infinite-horizon average-reward mdps with linear function approximation." International Conference on Artificial Intelligence and Statistics. PMLR, 2021.
> >
> > [A2]. Chen, Liyu, Rahul Jain, and Haipeng Luo. "Learning Infinite-Horizon Average-Reward Markov Decision Processes with Constraints." arXiv preprint arXiv:2202.00150 (2022).
> >
> > [A3]. Abbasi-Yadkori, Yasin, et al. "Politex: Regret bounds for policy iteration using expert prediction." International Conference on Machine Learning. PMLR, 2019.
> >
> > [A4]. Hao, B., Lazic, N., Abbasi-Yadkori, Y., Joulani, P., & Szepesvári, C. ``Adaptive approximate policy iteration" In International Conference on Artificial Intelligence and Statistics, PMLR, 2021.
> >
> > [A5].  Abbasi-Yadkori, Yasin, et al. "Exploration-enhanced Politex." arXiv preprint arXiv:1908.10479 (2019).

---

> > > ### Author Response · Authors · 2022-11-22
> > > **Follow Up**
> > >
> > > Dear Reviewer 65aT,
> > >
> > > We just wanted to check in and ask if the rebuttal clarified and answered the questions raised in your review. We would be very happy to engage further if there are additional questions!
> > >
> > > Also, if you feel that we have answered your concerns, please consider increasing the score.

---

> > > > ### Author Response · Authors · 2022-12-12
> > > > **Follow Up**
> > > >
> > > > Dear Reviewer 65aT,
> > > >
> > > > Your feedback has greatly improved our paper. Since the discussion period is going to end very soon, we are wondering whether you have any more feedback or concerns regarding our work. We will be happy to address those concerns if you have any.
> > > >
> > > > Authors

---

> > > > > ### Comment · Reviewer_65aT · 2022-12-13
> > > > > **After Rebuttal**
> > > > >
> > > > > Dear Authors,
> > > > >
> > > > > Thank you for your detailed response to address my queries. After reading other reviews and your responses, I am increasing my score.

---

> > > > > > ### Author Response · Authors · 2022-12-13
> > > > > > **Thank you**
> > > > > >
> > > > > > Dear Reviewer 65aT,
> > > > > >
> > > > > > Thank you for increasing the score. We are glad that we have addressed your queries.
> > > > > >
> > > > > > Authors

---

### Author Response · Authors · 2022-11-17
**General Response**

We thank the reviewers for their time and constructive feedback. *All the reviewers have agreed that the paper is well-written and technically sound*.  We would like to highlight that the reviewers acknowledged several other strengths of our work.

**Reviewer 65aT**:
1. The authors first propose an algorithm with $\tilde{O}(\sqrt{d^3T})$ regret and constraint violation, which has the tightest regret upper bound, but the algorithm is computationally inefficient. So they propose a computationally efficient algorithm that uses the primal-duel adaption of the LSVI-UCB algorithm but has $\tilde{O}((dT)^{¾})$ regret constraint violation upper bounds.
2. Under strong assumptions (Assumption 4 and Assumption 5), authors propose policy-based algorithms that are computationally efficient and have $O(\sqrt{T})$ regret constraint violation upper bounds (or even constant constraint violation bound).
3. No simulator (a common assumption) is needed to show the proposed algorithms' bounds.
4. This paper makes non-trivial advances over the current state-of-the-art.

**Reviewer 2G2e**:
1. The problem setup considered in this paper is infinite horizon average reward CMDP with linear function approximation, which is meaningful and interesting.
2. The theoretical results in both regret and constraint violation have improvements over existing works in the above mentioned specific problem domain. For example, the efficient algorithm 2 has better bounds of $\tilde{O}(T^{¾})$ compared with known one.

**Reviewer 3YVd**:
1. This paper is sufficiently complete. It proposes new algorithms, builds a theoretically-improved convergence analysis on both regret and violation bounds, and empirically verifies the result on stochastic environments. And this work is purely model-free; it doesn't require any estimation of transition kernel.
2. All statements are clear and well-supported. Main results are not very interesting but are the first time to be proposed and proved.

**Reviewer ob3W**:
1. The performance guarantee is significantly better than previous works. Compared with previous works for tabular CMDP, this paper works under weaker assumptions and has better upper bounds for regret and violation.
2. Linear function approximation is a highly noted topic in RL theory community. The constrained MDP problem is a meaningful extension and can be interesting to many.

The reviewers also raised some comments. In our detailed answers below, we believe that we have fully answered them. We would be happy to engage more if the reviewers have any more comments.

We have modified the paper in light of the comments made by the reviewers. **All the changes are marked in blue** (the revised version has been uploaded). The major changes are summarized in the following:

- We have now highlighted the technical novelties of Algorithms 1, 2, and 3 over the existing approaches.
   - **Please see Page 5 for Algorithm 1 for high-level differences (Also please see Appendix E.1 on page 14 for the technical differences)**.
   - **Please see pages 6 and 8 for Algorithm 2 for high. Also, see Appendix F.1 (pages 21 and 23) where we have now modified the outline of the proof to reflect on the differences compared to the existing algorithms**.
  - **Please see pages 34 (Appendix G.1) and 36 (Appendix G.1.1) where we have now pointed out the technical differences**.
- We have added new experimental results and insights on pages 44-46 (Appendix H). **New results (Figures 1-3) clearly show that regret and violation indeed grow sub-linearly which validates our theoretical findings**. Further, by choosing $\epsilon$ slightly higher, the numerical results show that one can achieve zero violation (Figure 3).

As reviewer 65aT pointed out that we have provided sufficient details to reproduce the results. Nevertheless, we will make all our codes public for the camera-ready version.

---

### Author Response · Authors · 2022-12-06
**Highlighting Our Contributions (Part 1)**

There seems to be some confusion among the reviewers regarding our contributions. We would like to highlight our main contributions in the following. We would be very happy to engage with the reviewers if they have any doubt or confusion.

1. **We are the first (model-free or model-based) to provide sub-linear regret and violation bound with linear function approximation**. Chen et al.'22 consider the tabular CMDP and provided sublinear regret and violation bound using model-based approach, instead we provide model-free approach and consider linear function approximation. In this way, we extend the understanding of the infinite horizon average reward CMDP in a significant manner.
-------------

2. **Algorithm 1 implies that $O(\sqrt{T})$ regret and violation bound are achievable under only Assumption 1 with linear function approximation setting for infinite horizon average reward CMDP**.  *This result significantly advances our understanding of infinite horizon average reward CMDP.* It is known that model-free approach can achieve $O(\sqrt{T})$ regret under Assumption 1 for the **unconstrained setup**, however, **whether it is possible to achieve $O(\sqrt{T})$ regret using model-free approach for constrained setup was open even for tabular setup**. For example, *Wei et al.’22 proposed a model-free algorithm that only achieves $O(T^{⅚})$ regret for tabular case* (under a stronger set of Assumption compared to Assumption 1). **We have now answered that question in an affirmative way.** In this regard, we complement the results obtained by Chen et al.'22  For a finite-state space (tabular), weakly communicating MDP satisfies Assumption 1, Chen et al.'22 showed that for weakly communicating MDP it is possible to achieve $O(\sqrt{T})$ regret and violation bound using **model-based** approach. Since linear CMDP contains tabular case,  we show that it is possible to achieve $O(\sqrt{T})$ regret and violation bound using **model-free** approach for weakly communicating MDP.
-------------
3. **Algorithm 1 is computationally inefficient, however, the result has statistical significance as it shows that it is possible to achieve $O(\sqrt{T})$ regret only under Assumption 1 for constrained MDP using a model-free approach**. *We would like to point out that developing provably-efficient algorithms (even though they can be computationally inefficient) has been standard to understand the difficulty in decision-making in RL setup. For example, model-based algorithm (Algorithm 4) proposed by Chen et al.'22 for tabular CMDP setup which achieves $O(\sqrt{T})$ regret and violation under Assumption 1 is also computationally inefficient*. In fact, Wei et al.'21a also proposed a computationally inefficient model-free algorithm that achieves $O(\sqrt{T})$ regret under only Assumption1 for unconstrained infinite horizon average reward case. Even though the algorithms are computationally inefficient they augment our understanding. *Even for unconstrained nonlinear function approximation episodic setups algorithms that achieve optimal sample complexity and yet computationally inefficient are proposed [A1,A2]. Those results are significant in expanding our knowledge*. **In a similar manner, we are urging the reviewers to consider the statistical significance of Algorithm 1**. Achieving the result is not straightforward as we need to obtain $\epsilon$-covering number for individual value function class unlike the unconstrained case. In order to achieve, the $\epsilon$-covering number we use the smoothness properties of the policy function class $\Pi$. The technical novelties of Algorithm 1 are highlighted in Section 3 (page 5) and Appendix E.
--------

4. **In Algorithm 2, we propose an *efficient variant* of Algorithm 1 and show that $O((dT)^{3/4})$ regret and violation bound can be achieved using an efficient approach under only Assumption 1.**  **In fact, under same set of Assumptions in Wei et al.'22, we achieve $O(T^{3/4})$ regret and zero violation in an infinite horizon average reward linear CMDP using a *model-free* algorithm whereas Wei et al.'22 only achieves $O(T^{5/6})$ regret in tabular case.** *Thus,  our result significantly improves the existing result.* Our approach is to divide the entire time horizon $T$ into multiple episodes ($T/H$) where each episode consists of $H$ steps. We use the algorithm proposed by Ghosh et al.'22 to learn good policies in this episodic setup.  Now, for the next step in comparing with the infinite horizon set-up, we need to show that the gap between the total reward in the infinite-horizon setup and the value function corresponding to the optimal stationary policy (for the infinite-horizon problem) in this episodic setup can also be upper bounded. However, bounding the above gap is not trivial and in fact one of our main contributions. Please see our response to reviewers ob3W and 2G2e (also Section 3.2, pages 6 and 8,  and Appendix F.1).

---

> ### Author Response · Authors · 2022-12-06
> **Highlighting our Contributions (Part 2)**
>
> 5. **Now, we point out the significance of Algorithm 3, which achieves zero constraint violation with $O(\sqrt{T})$ regret under a stronger set of Assumptions**. *Algorithm 3 can be implemented efficiently.*  **Thus, this is the first result that shows that $O(\sqrt{T})$ regret and zero violation is achievable using a model-free approach for CMDP using linear function approximation.** We propose a primal-dual adaptation of the MDP-EXP2. Extending the MDP-EXP2 (proposed in Wei et al. '21) to the constrained case is non-trivial. Unlike the unconstrained setup, the decision is taken based on the joint state-action bias function corresponding to reward and utility. Thus, how to bound the regret and violation separately from this joint decision process is not trivial. Second, we provide a high-probability bound unlike the expectation bound in Wei et al.'21a, hence, the analysis also differs. We point out the technical novelties in response to reviewers 2G2e and ob3W (and also see  Appendix G.1., and Appendix G.1.1.).
> ------
>
> 6. We also perform numerical evaluations on a simulated environment. **Our evaluations indeed show sub-linear regret and zero violation.** Please see Appendix H and the response to reviewer 65aT (part 2).
> -------
>
> References:
>
> [A1]. Jin, Chi, Qinghua Liu, and Sobhan Miryoosefi. "Bellman eluder dimension: New rich classes of RL problems, and sample-efficient algorithms." *Advances in neural information processing systems 34 (2021)*: 13406-13418.
>
> [A2]. Dann, Christoph, et al. "A provably efficient model-free posterior sampling method for episodic reinforcement learning." *Advances in Neural Information Processing Systems 34 (2021)* : 12040-12051.

---

### Decision · Program_Chairs · 2023-01-20

**Decision:**

Accept: poster

**Justification For Why Not Higher Score:**

Some reviewers made the argument that many of the ideas used in the algorithms have appeared in similar form in prior work.

**Justification For Why Not Lower Score:**

Overall, the authors prove a number of new results in settings that have been studied and are considered interesting in the literature. Notably, the reviewer with the lower score did not engage in discussions during the rebuttal phase.

**Metareview: Summary, Strengths And Weaknesses:**

This paper gives algorithms with improved regret bounds for certain natural MDP settings. Significant discussion took place during the rebuttal process. Given this discussion and my own understanding, I believe this work should be accepted to ICLR.

**Note From Pc:**

if the above contains the word "oral" or "spotlight" please see: "oral" presentation means -> notable-top-5% and "spotlight" means -> notable-top-25%. As stated in our emails, we are disassociating presentation type from AC recommendations